**TOOLS**

# A live-cell marker to visualize the dynamics of stable microtubules throughout the cell cycle

Klara I. Jansen[1], Malina K. Iwanski[1], Mithila Burute[1], and Lukas C. Kapitein[1]

The microtubule (MT) cytoskeleton underlies processes such as intracellular transport and cell division. Immunolabeling for posttranslational modifications of tubulin has revealed the presence of different MT subsets, which are believed to differ in stability and function. Whereas dynamic MTs can readily be studied using live-cell plus-end markers, the dynamics of stable MTs have remained obscure due to a lack of tools to directly visualize these MTs in living cells. Here, we present StableMARK (Stable Microtubule-Associated Rigor-Kinesin), a live-cell marker to visualize stable MTs with high spatiotemporal resolution. We demonstrate that a rigor mutant of Kinesin-1 selectively binds to stable MTs without affecting MT organization and organelle transport. These MTs are long-lived, undergo continuous remodeling, and often do not depolymerize upon laser-based severing. Using this marker, we could visualize the spatiotemporal regulation of MT stability before, during, and after cell division. Thus, this live-cell marker enables the exploration of different MT subsets and how they contribute to cellular organization and transport.

## Introduction

Cells use microtubules (MTs) to establish intracellular organization. These stiff and polarized polymers span long distances through cells and serve as tracks for directional transport driven by motor proteins that move toward either the plus- or minus-end of MTs. Most MTs switch between phases of polymerization and depolymerization in a process called dynamic instability (Mitchison and Kirschner 1984), which is coupled to the nucleotide state of the newly incorporated tubulin subunits at the MT plus-end (Akhmanova and Steinmetz 2015). The exact dynamics of MTs can furthermore be tuned by a wide variety of MT-associated proteins (MAPs) that can stabilize or destabilize MTs, either by modulating their end dynamics or by affecting the MT lattice along its entire length. In addition, MTs can undergo posttranslational modifications (PTMs), and some of these are believed to also impact, directly or indirectly, MT stability (Janke and Magiera 2020; Roll-Mecak 2020).

Immunocytochemistry on fixed cells stained for different PTMs has revealed that the concerted action of MT-regulating factors can lead to the emergence of distinct, coexisting MT subpopulations that greatly differ in their chemical composition and stability (Verhey and Gaertig 2007; Burute and Kapitein 2019; Janke and Magiera 2020; Roll-Mecak 2020). Most MTs are dynamic, meaning that they undergo rapid cycles of growth and shrinkage and quickly depolymerize upon cold treatment or when cells are exposed to MT-destabilizing agents such as nocodazole. Dynamic MTs generally acquire few PTMs, perhaps because of their short lifetime (Webster et al., 1987; Piperno et al., 1987; Kreis 1987; Wehland and Weber 1987). In contrast to dynamic MTs, stable MTs survive such treatments and are strongly enriched in PTMs, such as acetylation (K40) or C-terminal detyrosination (Webster et al., 1987; Piperno et al., 1987; Kreis 1987; Wehland and Weber 1987; Cambray-Deakin and Burgoyne 1987). The existence of different MT subsets is believed to play an important role in the spatial organization of cells as different MT subsets have distinct organizations and are used by specific motor proteins as tracks for organelle transport (Burute and Kapitein 2019). Stable, long-lived MTs polarize toward the leading edge in migrating fibroblasts (Gundersen and Bulinski 1988) and are predominantly localized in the perinuclear area in common cell lines such as HeLa, COS-7, and U2OS, whereas dynamic MTs display a more homogenous distribution in these cell types. In dendrites of hippocampal neurons, stable MTs are enriched in the core of the dendritic shaft, while dynamic MTs mainly localize near the plasma membrane (Tas et al., 2017; Katrukha et al., 2021). In addition, stable and dynamic MTs often have opposite orientations in dendrites (Tas et al., 2017). Furthermore, it has been shown that some motor proteins bind selectively to one subset of MTs. For example, Kinesin-1 moves preferentially along stable MTs (Cai et al., 2009; Dunn et al., 2008; Guardia et al., 2016; Tas et al., 2017; Liao and Gundersen 1998; Kaul et al., 2014), whereas Kinesin-3 prefers dynamic MTs (Guardia et al., 2016; Tas et al., 2017; Lipka

[1]Department of Biology, Cell Biology, Neurobiology and Biophysics, Faculty of Science, Utrecht University, Utrecht, Netherlands.

Correspondence to Lukas C. Kapitein: l.kapitein@uu.nl.

et al., 2016). The guidance of motors by distinct MT subsets has been implied to underly organelle positioning in non-neuronal cells (Guardia et al., 2016; Mohan et al., 2019; Serra-Marques et al., 2020) and polarized transport in hippocampal neurons (Tas et al., 2017).

Although the functional relevance of MT subsets for the spatial organization of cells is becoming increasingly clear, little is known about the organization and establishment of the MT subsets themselves. How do certain MTs become stabilized and how do they obtain their specific spatial organization? Is the stability of an MT only modulated at its ends or also along the lattice? And how do PTMs and MAPs relate to MT stability? Addressing these questions would greatly benefit from tools that enable the direct visualization of MT subsets in live cells. Dynamic MTs can be readily visualized in live cells using plus-end markers or the recently developed sensor for tyrosinated tubulin (Kesarwani et al., 2020); however, this is not the case for stable MTs, which can currently only be visualized upon cold treatment or exposure to MT-depolymerizing drugs, or approximated using immunocytochemistry for different PTMs. To address this, we set out to develop a live-cell marker for stable MTs. We identify rigor Kinesin-1 as a faithful marker for stable MTs that can, at low expression levels, be used to visualize the behavior of stable MTs at high spatiotemporal resolution. We employ our live-cell marker to explore the dynamic properties of stable MTs and show how different MTs are selectively stabilized before, during, and after mitosis.

## Results

Stable MTs are characterized by their long lifetime and resistance to MT-depolymerizing drugs, yet the exact origin of their stability has remained largely unknown (Akhmanova and Kapitein 2022). Stable MTs are enriched in PTMs like acetylation and detyrosination, and immunolabeling for these modifications is often used as a proxy for stable MTs in experimental work. Given the intrinsic property of Kinesin-1 to move selectively along the subset of MTs enriched in acetylation and detyrosination (Cai et al., 2009; Dunn et al., 2008; Liao and Gundersen 1998; Kaul et al., 2014), we set out to test the application of rigor Kinesin-1 as a marker for stable MTs. The G234A point mutation in Kinesin-1 perturbs its ATPase activity (Rice et al., 1999), resulting in a motor protein that has a very low rate of MT unbinding. Similar to the active motor, the G234A rigor mutant of Kinesin-1 binds to and effectively decorates acetylated MTs (Tas et al., 2017; Guardia et al., 2016; Farías et al., 2017). To minimize the amount of fluorescent rigor-protein needed to allow proper visualization during live-cell imaging experiments, we fused rigor Kinesin-1 to a tandem of the very bright fluorophore mNeonGreen (Shaner et al., 2013) and named the resulting construct "StableMARK," for Stable Microtubule-Associated Rigor-Kinesin (Fig. 1, A and B). When we expressed StableMARK in U2OS cells it localized, as expected, specifically to the subset of acetylated MTs (Fig. 1, C–E).

To examine whether StableMARK truly labels stable MTs or is merely a marker for MTs that are acetylated, we first tested the relation between MT acetylation and MT stability. U2OS

cells were treated using different (combinations) of MT-impacting drugs and subsequently immunolabeled for acetylated tubulin and total tubulin. Upon treatment with DMSO as a vehicle control, acetylated MTs formed a clear subset within the total tubulin network. Upon treatment with the MT-stabilizing drug Taxol, most MTs became acetylated and the total intensity of acetylated tubulin increased 1.6 ± 0.5 (mean ± SD)-fold. Treatment with tubacin, an inhibitor of the MT-deacetylating enzyme Histone Deacetylase 6, also resulted in a clear increase in MT acetylation levels (fold change: 1.7 ± 0.5). Upon treatment with nocodazole, an MT-targeting drug that interferes with MT polymerization, the majority of MTs were lost and all MTs that survived were acetylated (fold change of total tubulin: 0.19 ± 0.07). In cells that were first treated with Taxol and subsequently treated with Taxol and nocodazole simultaneously, no MT mass was lost and the level of MT acetylation was high (fold change total tubulin: 1.2 ± 0.3; acetylated tubulin: 1.6 ± 0.5). However, in cells that were first treated with tubacin to induce over-acetylation of MTs and then treated with tubacin and nocodazole simultaneously, most MTs were lost and only a subset remained (fold change total tubulin: 0.12 ± 0.05). These data demonstrate that while nocodazole-resistant, stable MTs are acetylated, MT acetylation by itself does not confer nocodazole resistance (i.e., stability) to MTs, which is in agreement with earlier work (Palazzo et al., 2003; Fig. 2, A and B).

Next, we tested the localization of StableMARK in U2OS cells treated with DMSO, Taxol, or tubacin. In control conditions or upon treatment with DMSO, StableMARK localized with high specificity to the subset of acetylated MTs (Manders' colocalization coefficient of acetylated tubulin to StableMARK: 0.75 ± 0.15; colocalization of StableMARK with acetylated tubulin: 0.89 ± 0.06). When MTs were stabilized using Taxol (and also became more acetylated), StableMARK decorated most MTs (Manders' colocalization coefficient of acetylated tubulin to StableMARK: 0.86 ± 0.08; colocalization of StableMARK with acetylated tubulin: 0.93 ± 0.05). However, when MTs were over-acetylated (but not stabilized) using tubacin, StableMARK remained localized to a specific, mostly perinuclear subset of MTs and did not decorate all acetylated MTs (Manders' colocalization coefficient of acetylated tubulin to StableMARK: 0.45 ± 0.14; colocalization of StableMARK with acetylated tubulin: 0.96 ± 0.03; Fig. 2, C and D), indicating that StableMARK labeling of MTs is not directed by MT acetylation but by MT stability.

We found that detyrosinated MTs were largely absent in our U2OS cells and that StableMARK localized to more MTs than just the detyrosinated ones (Fig. S1, A and B). Nevertheless, as detyrosination has also been associated with stable MTs, we sought to test how StableMARK localization responded to elevated levels of detyrosinated MTs; we increased levels of detyrosinated tubulin in U2OS cells by overexpressing the tubulin carboxypeptidase vasohibin 1 (VSH1) and its interacting partner small vasohibin-binding protein (SVBP; Nieuwenhuis et al., 2017; Fig. S1, C and D). Upon coexpression of StableMARK and VSH1/SVBP, StableMARK remained localized to its specific subset and did not decorate all detyrosinated MTs (Manders' colocalization coefficient of detyrosinated tubulin to StableMARK: 0.34 ± 0.12; colocalization of StableMARK to detyrosinated

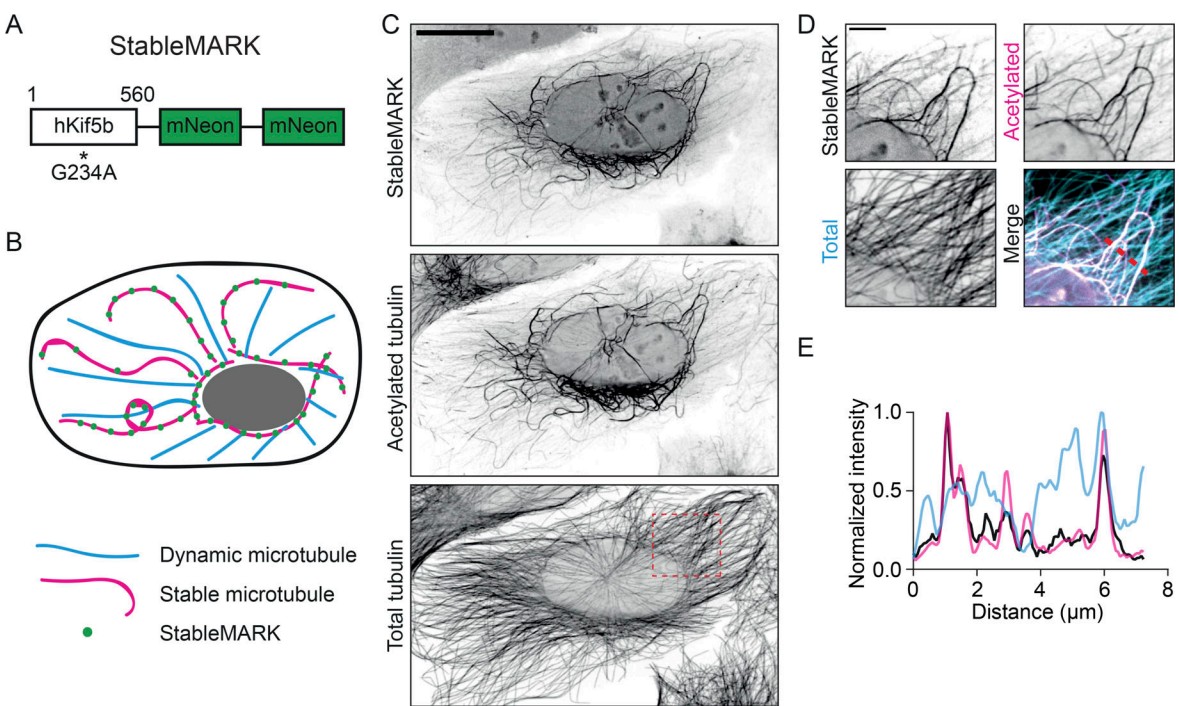

**Figure 1. StableMARK labels a subset of (acetylated) MTs. (A)** StableMARK: 1–560 truncation of human Kinesin-1 containing the G234A rigor mutation fused to a tandem of mNeonGreen separated by linker sequences. **(B)** Cartoon: Upon expression in cells, StableMARK decorates stable MTs in a speckle-like manner. **(C)** Fluorescence images of U2OS cells expressing StableMARK and immunolabeled for acetylated tubulin and α-tubulin are shown in inverted contrast. **(D)** Zooms of the region indicated by a dashed box in C. **(E)** Intensity profile across the region indicated with a dashed line in D. Scale bars, 20 µm (C), 5 µm (D).

tubulin: 0.78 ± 0.09; Fig. S1, E and F). Thus, StableMARK binding to MTs is not guided by MT detyrosination. In addition, we found that increasing MT detyrosination levels by overexpression of VSH1/SVBP does not confer stability to MTs as the majority of detyrosinated MTs were lost upon treatment with nocodazole (Fig. S1 G), in agreement with earlier work (Khawaja et al., 1988).

Given that neither acetylation nor detyrosination is directly recognized by StableMARK, we next tried to establish which feature of stable MTs StableMARK recognizes. Here, we were guided by three key findings. First, the addition of Taxol, the only treatment that altered the localization of StableMARK, is known to expand the MT lattice longitudinally, increasing the dimer rise by ∼2 Å (Alushin et al., 2014). Indeed, an expanded lattice is common to MTs stabilized by different means (Taxol, guanosine-5'-[(α,β)-methyleno]triphosphate [GMPCPP]), and hydrolysis-deficient tubulin; LaFrance et al., 2022; Alushin et al., 2014). Second, MTs bound by StableMARK have an expanded lattice in cells, as indicated by fluorescence-guided cryo-focused ion beam transmission electron microscopy data (de Jager et al., 2022 *Preprint*). Third, in vitro experiments have revealed that at high concentrations in the no-nucleotide state, the kinesin-1 motor domain can extend MT lattice spacing, suggesting that it prefers to bind to this expanded MT lattice conformation at lower concentrations (Peet et al., 2018; Shima et al., 2018). We, therefore, purified StableMARK and tested its preference for different MT lattices in vitro (Fig. S2 A). We prepared double-cycled GMPCPP-bound MT seeds with a high percentage of fluorescent tubulin (49%) and immobilized these in a flow cell.

Upon the addition of soluble tubulin (with 5% fluorescent tubulin) and GTP, MTs grew templated by these seeds. These MTs are expected to hydrolyze their GTP to produce GDP lattices. We then added Taxol to expand the MT lattices or DMSO to leave them compacted, followed by a brief with StableMARK. Importantly, all assays were performed paired to ensure that pipetting errors or any other variability could not account for any observed differences (Fig. S2 B). We then counted the number of StableMARK molecules along the compacted (GDP + DMSO) and expanded (GDP + Taxol) lattices and found that there were 1.47 ± 0.16 (mean ± SD) times more motors on the expanded MTs than on the paired compacted MTs. This suggests that the selectivity of StableMARK for stable MTs is at least in part mediated by its preference for expanded MT lattices.

Having established StableMARK as a specific marker for stable MTs whose selectivity appears to be (in part) due to the increased dimer rise of these MTs, we tested its potential as a live-cell marker. Earlier work has described the increased lifetime of stable MTs compared with dynamic MTs (Schulze and Kirschner 1986; Webster et al., 1987; Infante et al., 2000). To test if we could recapitulate this finding with our live-cell marker, we cotransfected U2OS cells with StableMARK and mCherry-tubulin as a total tubulin marker (see Video 1 for a live whole-cell timelapse) and picked cells with low levels of StableMARK expression (i.e., a speckled decoration of MTs). Using total internal reflection fluorescence (TIRF) microscopy, we observed that many MTs labeled exclusively by mCherry-tubulin displayed dynamic instability, whereas MTs decorated by StableMARK

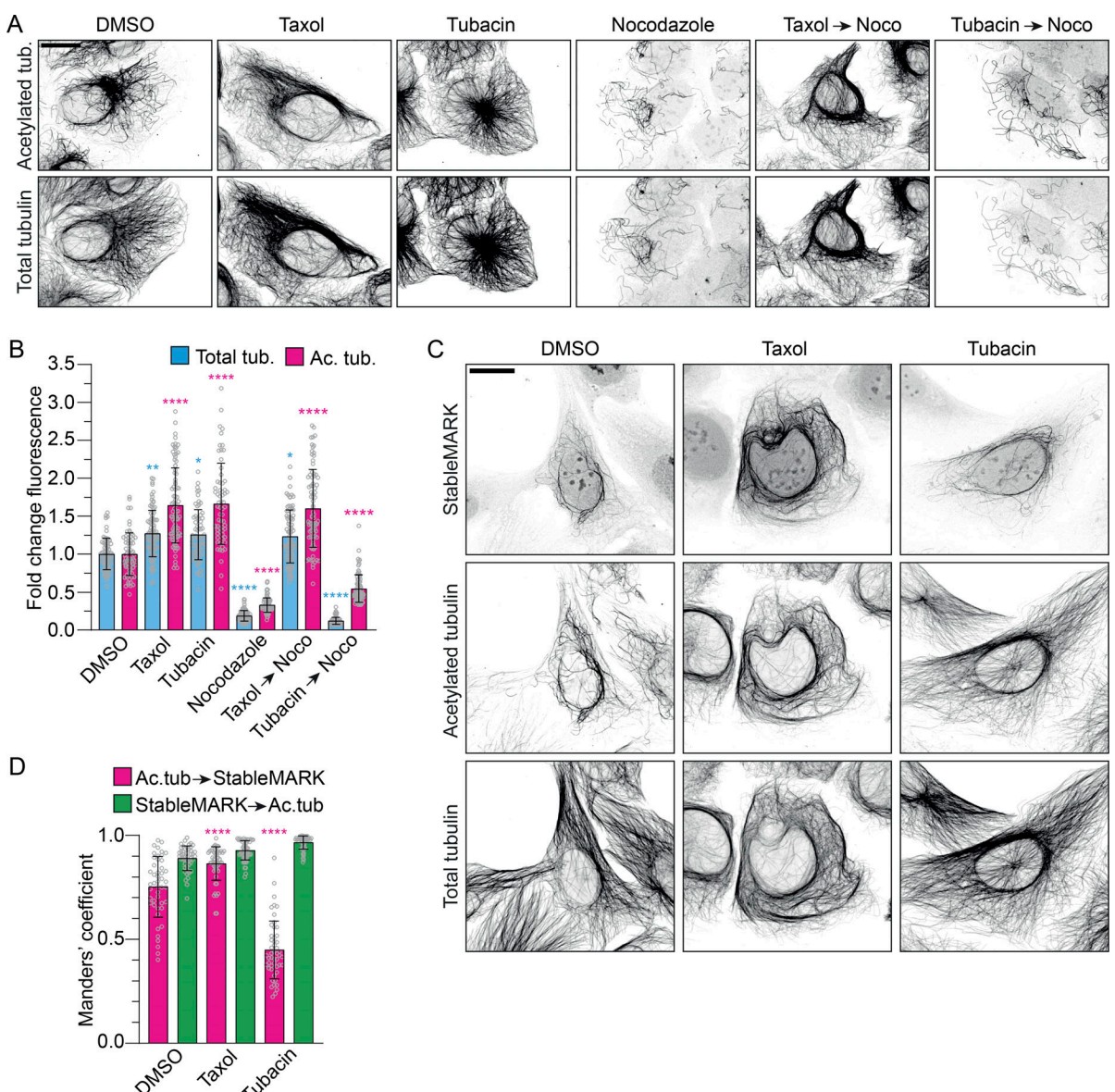

Figure 2. **StableMARK labels the subset of stable MTs. (A)** Fluorescence images of U2OS cells treated with 0.1% DMSO (3 h), 10 μM Taxol (2 h), 10 μM tubacin (2 h), 10 μM nocodazole (1 h), 10 μM Taxol (2 h) followed by 10 μM Taxol + 10 μM nocodazole (1 h), or 10 μM tubacin (2 h) followed by 10 μM tubacin + 10 μM nocodazole (1 h) and immunolabeled for acetylated tubulin and α-tubulin. **(B)** Bar graph showing the fold change in mean fluorescence intensity of acetylated tubulin and total tubulin normalized to the DMSO control for the different conditions described in A. Graph represents mean ± SD as well as individual values (gray circles) for 64–90 cells per condition from three independent experiments. Kruskal–Wallis test with Dunn's multiple comparisons test. *P ≤ 0.05, **P ≤ 0.01, ****P ≤ 0.0001. **(C)** Fluorescence images of U2OS cells expressing StableMARK treated with 0.1% DMSO (2 h), 10 μM Taxol (2 h), or 10 μM tubacin (2 h) and immunolabeled for acetylated tubulin and α-tubulin. **(D)** Bar graph showing thresholded Manders' coefficients measured from 12.5 × 12.5 μm ROIs per cell for the conditions described in C. Pink bars represent the colocalization coefficient for acetylated MTs to StableMARK-decorated MTs. Green bars represent the colocalization coefficient for StableMARK-decorated MTs to acetylated MTs. Graph represents mean ± SD as well as individual values (gray circles) for 47–52 cells per condition from three independent experiments. Kruskal–Wallis test with Dunn's multiple comparisons test. ****P ≤ 0.0001. Scale bars, 20 μm (A and C).

remained stable on the imaging timescale of several minutes (Fig. 3 A; Video 2). Next, we lowered our frame rate and increased the imaging duration and found that our live-cell marker can be used to track MTs that are stable on much longer time scales (>30 min; Fig. 3 B; Video 3). To further confirm the stability of the MTs recognized by StableMARK, we performed a nocodazole treatment. The addition of 10 μM nocodazole induced a rapid loss of most MTs labeled with mCherry-tubulin, whereas the subset

of StableMARK-decorated MTs remained largely unaffected by the treatment. Indeed, 30 min after the addition of nocodazole, a clear network of StableMARK-positive MTs could still be observed (Fig. S1 H). Thus, StableMARK labels the subset of stable MTs, which in these cells are highly acetylated in control conditions.

A characteristic feature of stable MTs is their curved appearance (Piperno et al., 1987; Friedman et al., 2010; Katrukha

Figure 3. **StableMARK as a live-cell marker for stable MTs. (A and B)** Stills from live-cell imaging of StableMARK and mCherry-tubulin on different time scales (see also Videos 2 and 3). Colored arrowheads in A show examples of MTs displaying dynamic instability. Colored arrowheads in B indicate StableMARK-

decorated MTs that are retained during the duration of imaging. Time, min:s. **(C and E)** Stills and schematic representations from live-cell imaging of StableMARK in U2OS cells depicting MT sliding (indicated by colored arrowheads; see also Video 4) and MT looping (see also Video 5). Time, s:ms. **(D)** Kymograph of sliding event in C. **(F)** Histogram showing the speeds of StableMARK-MT movements (115 events from 42 cells from two independent experiments). The green line shows Gaussian fit with a mean of 0.7 µm/s. Scale bars, 2.5 µm.

---

et al., 2017). As our live-cell marker decorates stable MTs in a speckled manner, each StableMARK molecule effectively acts as a fiducial marker, facilitating straightforward tracking of individual MT filaments. Using our live-cell marker, we could thus study the behavior of stable MTs in live cells. We found that, although these MTs were stable, they were far from static and displayed distinct behaviors such as sliding (at ∼0.7 µm/s, Fig. 3, C and F; Video 4), active curvature induction (Fig. 3 E; Video 5), deformation (Fig. S3 A), and breaking (Fig. S3 B), consistent with earlier reports (Xu et al., 2017; Robison et al., 2016; Katrukha et al., 2017). Occasionally, (partial) depolymerization events of stable MTs were observed (Fig. S3 C). Thus, StableMARK is a powerful tool to visualize the behavior of individual long-lived MTs in living cells with high spatiotemporal resolution.

A potential limitation of our approach is that MT-binding proteins can induce artifacts of the MT cytoskeleton when highly overexpressed, as has, for example, been observed with the N-terminal fragment of Ensconsin/MAP7 (Faire et al., 1999). To examine whether overexpression of StableMARK resulted in MT hyper-acetylation as a result of overstabilization of MTs, we measured MT acetylation levels as a function of StableMARK overexpression levels. We expressed StableMARK in U2OS cells and performed immunolabeling for acetylated tubulin and α-tubulin (Fig. 4 A, high-resolution examples; see Fig. S4 A for sample images used for quantification). To quantify the effect of StableMARK expression on MT acetylation levels, we randomly imaged a large number of StableMARK- and non-expressing cells to capture the expression range of StableMARK in any given experiment performed under similar conditions. We determined the intensity of acetylated tubulin and StableMARK, normalized to the intensity of α-tubulin for individual cells. As these types of experiments are very sensitive to variations in sample preparation and microscope performance, results of an independent duplicate experiment are reported separately in the supplemental materials. At high expression levels, we observed a clear correlation between StableMARK expression and MT acetylation levels, with high StableMARK-expressing cells having increased levels of MT acetylation (Fig. 4 B and Fig. S4 B). In addition, high StableMARK expression induced bundling of acetylated MTs (Fig. 4 A). However, at low expression levels, the acetylation levels of StableMARK-positive cells were within the range of acetylation observed for non-expressing cells. This indicates that in the subpopulation of low-expressing cells, MT acetylation levels are not significantly altered. To quantitatively determine the expression levels corresponding to low, medium, and high expression, we performed fluorescence correlation spectroscopy (FCS) experiments (Kim et al., 2007). This revealed that our marker can be found in concentrations ranging around 0.02–72 µM, where concentrations below 2 µM correspond to expression levels where we find no hyper-acetylation (Fig. S4 D).

To also assess the effect of StableMARK expression on the amount of dynamic MTs, we expressed StableMARK in U2OS cells and performed immunolabeling for end-binding protein 1 (EB1). EB proteins bind to the tips of polymerizing MTs and are often used to visualize the subset of dynamic MTs (Akhmanova and Steinmetz 2015). To assess the effect of StableMARK expression on dynamic MTs, we randomly imaged StableMARK- and non-expressing cells. Subsequently, we counted the number of EB1 comets per µm² and measured the StableMARK intensities per cell. We observed a weak inverse correlation between StableMARK intensity and the density of EB1 comets, with a decrease in the density of EB1 comets with increasing StableMARK intensity (Fig. 4, C and D, and see Fig. S4 C for a duplicate experiment). Nonetheless, in low-StableMARK-expressing cells (i.e., at concentrations below 2 µM), the density of EB1 comets was similar to control cells, indicating that the amount of polymerizing MTs in the cell is not affected by low levels of StableMARK (Fig. 4, C and D). In addition, we assessed the MT growth rate in control and low-StableMARK-expressing cells using the growing plus-end marker EB3-tdTomato and found no difference between the two conditions (Fig. 4, E and F).

To examine whether StableMARK-decorated MTs disassemble and reassemble normally, we performed a serum starvation assay. Earlier experiments have shown that the majority of stable MTs, identified by their nocodazole-resistance and immunolabeling for detyrosinated tubulin, disappear upon prolonged serum starvation and reappear again upon the addition of serum in 3T3 cells (Gundersen et al., 1994). To test whether StableMARK-decorated MTs respond to these physiological clues, we performed a serum starvation assay with Swiss 3T3 cells and stained for tyrosinated tubulin (as a marker for dynamic MTs) and acetylated tubulin (as a marker for stable MTs). Control cells cultured in a medium containing serum (full medium) had a clear population of stable MTs in the perinuclear region. Upon prolonged serum starvation, the majority of these MTs was lost and only a few remaining stable MTs could be observed. In cells subjected to prolonged serum starvation followed by 8 h of full medium, a perinuclear network of stable MTs was re-established (Fig. 4 G). The same response to prolonged serum starvation and readdition of serum was observed in Swiss 3T3 cells expressing StableMARK (Fig. 4 H). Additionally, we did not find a difference in the level of tyrosinated tubulin and acetylated tubulin between control and StableMARK-expressing cells in any of the conditions (Fig. 4 I). This indicates that the subset of stable MTs can still undergo changes and respond to physiological cues in presence of StableMARK. In other words, StableMARK-decorated MTs are not overstabilized as they behave similarly to non-StableMARK-decorated (i.e., in non-expressing cells) stable MTs during prolonged serum starvation. Taken together, our data demonstrate

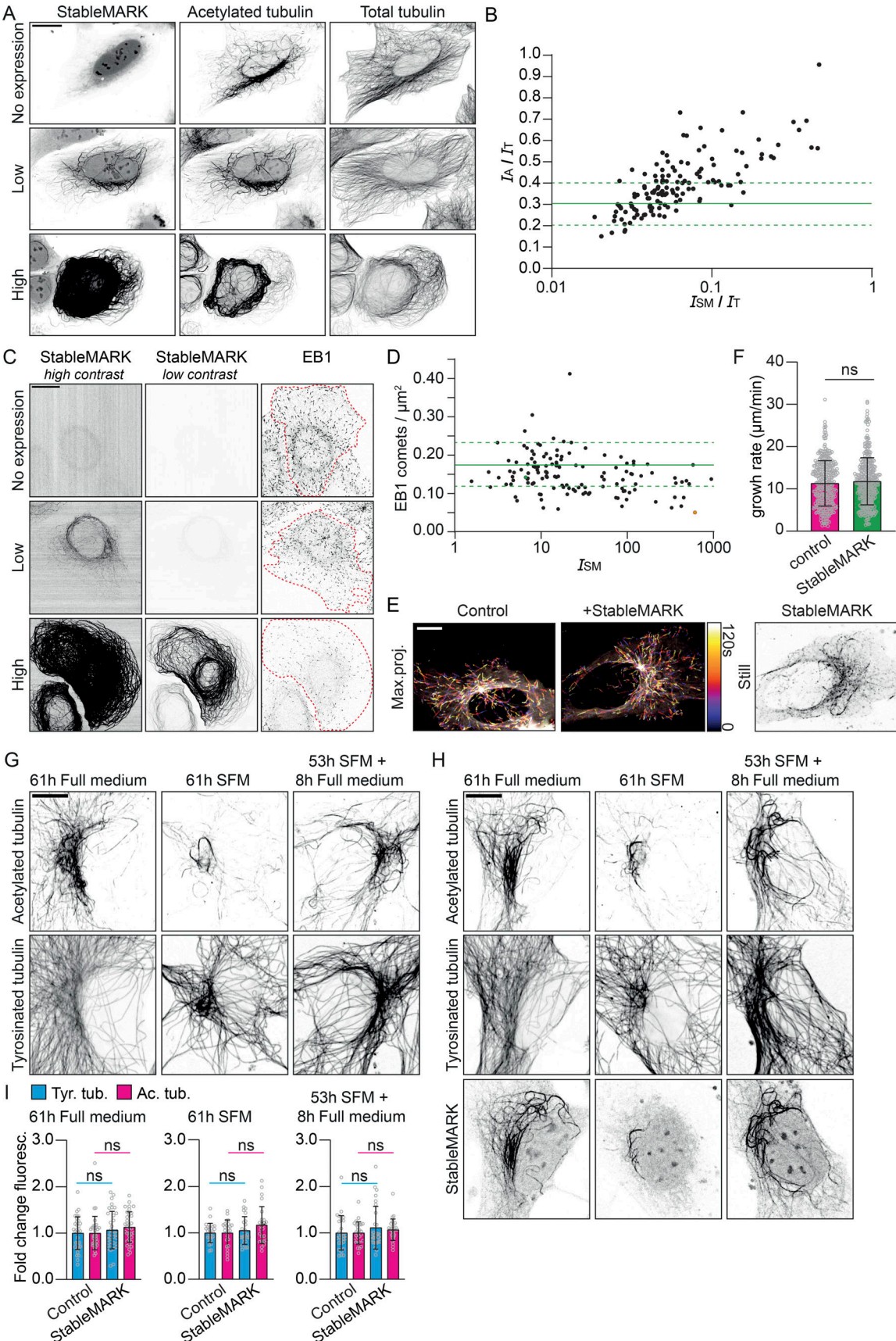

Figure 4. **StableMARK at low expression levels has minimal effects on the MT cytoskeleton. (A)** Fluorescence images of U2OS cells stained for acetylated tubulin and α-tubulin at different levels of StableMARK expression. **(B)** Quantification showing the intensity ratio of acetylated tubulin ($I_A$) over total

tubulin ($I_T$), plotted against the intensity ratio of StableMARK ($I_{SM}$) over total tubulin for individual StableMARK-expressing cells (each dot represents a single cell, n = 137). Solid green line + dashed lines indicate mean ± SD of the intensity ratio of acetylated tubulin over total tubulin for non-expressing cells (n = 125). Data from one experiment; independent duplicate in supplement (Fig. S4, A and B). **(C)** Fluorescence images of U2OS cells immunolabeled for EB1 at different levels of StableMARK expression. **(D)** Quantification showing the number of EB1 comets/µm² for cells with different StableMARK intensities (each dot represents a single cell, n = 131). Green data point is low expression example shown in C; orange datapoint is high expression example shown in C. Solid green line + dashed lines indicated mean ± SD of amount of EB1 comets/µm² for non-StableMARK expressing cells (n = 105). Data from one experiment; independent duplicate in supplement (Fig. S4 C). **(E)** Color-coded maximum intensity projections from stream acquisitions of EB3 comets in control condition or in presence of StableMARK (still of StableMARK channel shown on the right). **(F)** Quantification of MT growth rate determined from EB3 stream acquisitions. Bar graphs represent mean ± SD as well as individual datapoints (gray circles) for 331–344 MTs from 21 cells per condition from three independent experiments. Mann–Whitney U test. **(G)** Immunolabeling of acetylated and tyrosinated MTs in Swiss 3T3 fibroblasts after 61 h in full medium, 53 h in SFM, 61 h in SFM, or after 53 h SFM followed by 8 h in full medium. **(H)** Immunolabeling of acetylated- and tyrosinated MTs in Swiss 3T3 fibroblasts expressing StableMARK under the same conditions as in G. **(I)** Bar graphs showing the fold change in mean fluorescence intensity of tyrosinated tubulin and acetylated tubulin of StableMARK-expressing cells normalized to non-transfected control cells for the conditions described in G and H. Graphs represent mean ± SD as well as individual values (gray circles) for 25–35 cells per condition from three independent experiments. Unpaired t test. Scale bars, 20 µm (A and C), 10 µm (E, G, and H).

that in cells with low expression levels of StableMARK, effects on the properties of the MT cytoskeleton are minimal.

StableMARK occupies the same binding sites on the MT as active Kinesin-1. Therefore, the expression of StableMARK could potentially perturb endogenous Kinesin-1–driven organelle transport. Distribution of mitochondria throughout the cell is Kinesin-1–dependent and the absence of Kinesin-1 or its activating MAP, MAP7, leads to the clustering of mitochondria around the nucleus. Therefore, the spreading of mitochondria from the nucleus toward the periphery of the cell can be used as a read-out for Kinesin-1–mediated transport (Hooikaas et al., 2019; Serra-Marques et al., 2020). To assess the effect of StableMARK expression on Kinesin-1–driven organelle transport, we scored mitochondrial spreading in random StableMARK- and non-expressing cells. While in high-expressing cells, mitochondria were tightly clustered around the nucleus in the majority of cells (67%), in low-expressing cells, mitochondria were spread throughout the cytoplasm in most cells (66%), similar to what is seen for control cells (70%; Fig. 5, A and B). Similar results were found in a duplicate experiment (Fig. S4 E). We next tested whether organelle transport can take place along StableMARK-decorated MTs. Rab6a secretory vesicles are transported from the Golgi to the plasma membrane by the concerted action of Kif5b (Kinesin-1) and Kif13b (Kinesin-3; Serra-Marques et al., 2020). By simultaneous live-cell imaging of StableMARK and Rab6a using TIRF microscopy, we could directly observe the transport of Rab6a vesicles along StableMARK-decorated MTs (Fig. 5, C–E; Video 6). To quantify transport dynamics in the presence of StableMARK, we assessed the dynamics of lysosomes and Rab6-positive vesicles in control and low StableMARK-expressing cells. For both organelles, we found that, in presence of StableMARK, the distribution of vesicle speeds was very similar to the control condition (Fig. 5, F–I). Taken together, these experiments indicate that in the subpopulation of low-expressing cells, organelle transport can still take place along StableMARK-decorated MTs.

Next, we wanted to explore the mechanisms that stabilize the subset of StableMARK-decorated MTs and considered the following two scenarios: (1) long-lived MTs are stabilized by mechanisms that prevent depolymerization of the plus-end; (2) long-lived MTs are stabilized along their whole length, e.g. by stabilizing MAPs or other modifications. Previous laser-ablation experiments have revealed that upon severing of dynamic MTs, the freshly generated plus-end rapidly depolymerizes, whereas the minus-end often remains stable (Jiang et al., 2014; Walker et al., 1989). We reasoned that if the stability of long-lived MTs originates from their plus-end, the MT-ends generated by laser-based severing would still be susceptible to depolymerization. However, if long-lived MTs are stabilized along their whole length, freshly generated plus-ends would remain stable. To test this, we photoablated MTs in the perinuclear area of U2OS cells expressing mCherry-tubulin and StableMARK using a high-power pulsed 355 nm laser (Fig. 6 A). As expected, freshly generated MT-ends that were only labeled by mCherry-tubulin quickly depolymerized after severing (Fig. 6 B; Video 7). For StableMARK-decorated MTs, we observed two responses to photo-ablation: (1) depolymerization of (one of) the freshly generated ends (Fig. S5 A; 52 ± 13% of cut StableMARK-MTs, Fig. 6 C); (2) freshly generated ends remained stable on both sides of the cut (Fig. 6 B; 48 ± 13% of cut StableMARK-MTs, Fig. 6 C). These results indicate that StableMARK-decorated MTs have varying degrees of stability. This could reflect an aging effect in which stabilization starts by stabilization of the plus-end (or other sites on the lattice), followed by stabilization of the MT along the whole lattice, e.g., via lattice expansion.

If stabilization indeed starts at the plus-end, this could mean that the protein composition at the plus-end of stable MTs is different from that of dynamic MTs to prevent depolymerization and perhaps polymerization. To examine whether stable MTs have dynamic ends, we performed simultaneous live-cell imaging of StableMARK and EB3-tdTomato. We observed EB3 comets and hence local MT polymerization events at the tips of a few StableMARK-decorated MTs (Fig. 6, D and E; Video 8), but for the majority of StableMARK-positive MT-ends, no EB3 comets could be observed (92.8% ± 0.3; Fig. 6 F). Note that we could not assess the polarity of StableMARK MT-tips that had no EB3 comet associated, so theoretically, half of these StableMARK-positive MT-tips could be minus-ends. Nonetheless, the absence of (EB3-highlighted) MT growth at the ends of most StableMARK-decorated MTs suggests the presence of proteins that prevent both growth and (based on our severing experiments) shrinkage. Consistently, recent in vitro work has revealed that tyrosinated tubulin enhances the recruitment of CLIP-170 and EB1 to the plus-ends of MTs,

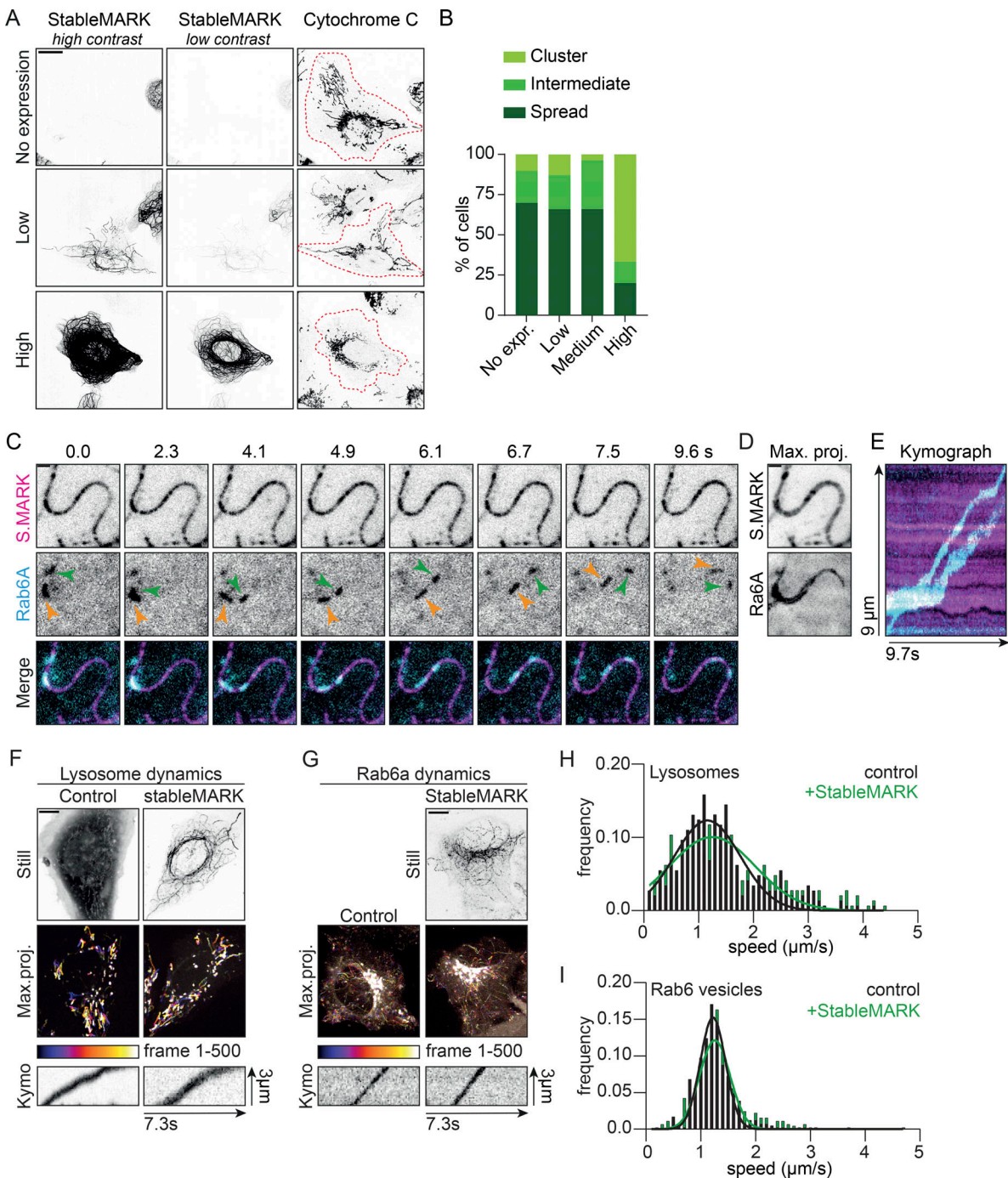

Figure 5. **StableMARK at low expression levels has minimal effects on organelle transport.** **(A)** Fluorescence images of U2OS cells stained for the mitochondrial marker cytochrome C at different levels of StableMARK expression. Cell outlines are indicated with red dashed lines. **(B)** Classification of mitochondria distribution at different levels of StableMARK expression (no expression, n = 166 cells; StableMARK expressing, n = 119 cells [low, n = 47 cells; medium, n = 56 cells; high, n = 15 cells]). Data from one experiment; independent duplicate in supplement (Fig. S4 D). **(C)** Stills from live-cell imaging of U2OS cells expressing StableMARK and mCherry-Rab6a (see also Video 6). **(D)** Maximum projection over time from the stream acquisition shown in C. **(E)** Kymograph showing Rab6a motility along the StableMARK-decorated MT shown in C. **(F)** Live-cell imaging of lysosomes using SiR-lysosome. Stills from GFP fill (control condition) and StableMARK are shown as well as color-coded maximum projections and illustrative kymographs of streams acquired from the SiR-lysosome channel. **(G)** Live-cell imaging of Rab6a-mCherry. Still of StableMARK is shown as well as color-coded maximum projections and illustrative kymographs of streams acquired from Rab6a channel. **(H)** Histogram showing the speed distribution of lysosomes in control cells and low-StableMARK-expressing cells (289–290 events from 14 to 16 cells per condition from two independent experiments). Black line (control condition) and green line (StableMARK condition) show Gaussian fit with a mean of 1.16 µm/s and 1.25 µm/s, respectively. **(I)** Histogram showing the speed distribution of Rab6 vesicles in control cells and low-StableMARK-expressing cells (410–447 events from 20 cells per condition from three independent experiments). Black line (control condition) and green line (StableMARK condition) show Gaussian fit with a mean of 1.22 and 1.26 µm/s, respectively. Scale bars, 20 µm (A), 10 µm (F and G), and 1 µm (C and D). Time, s: ms.

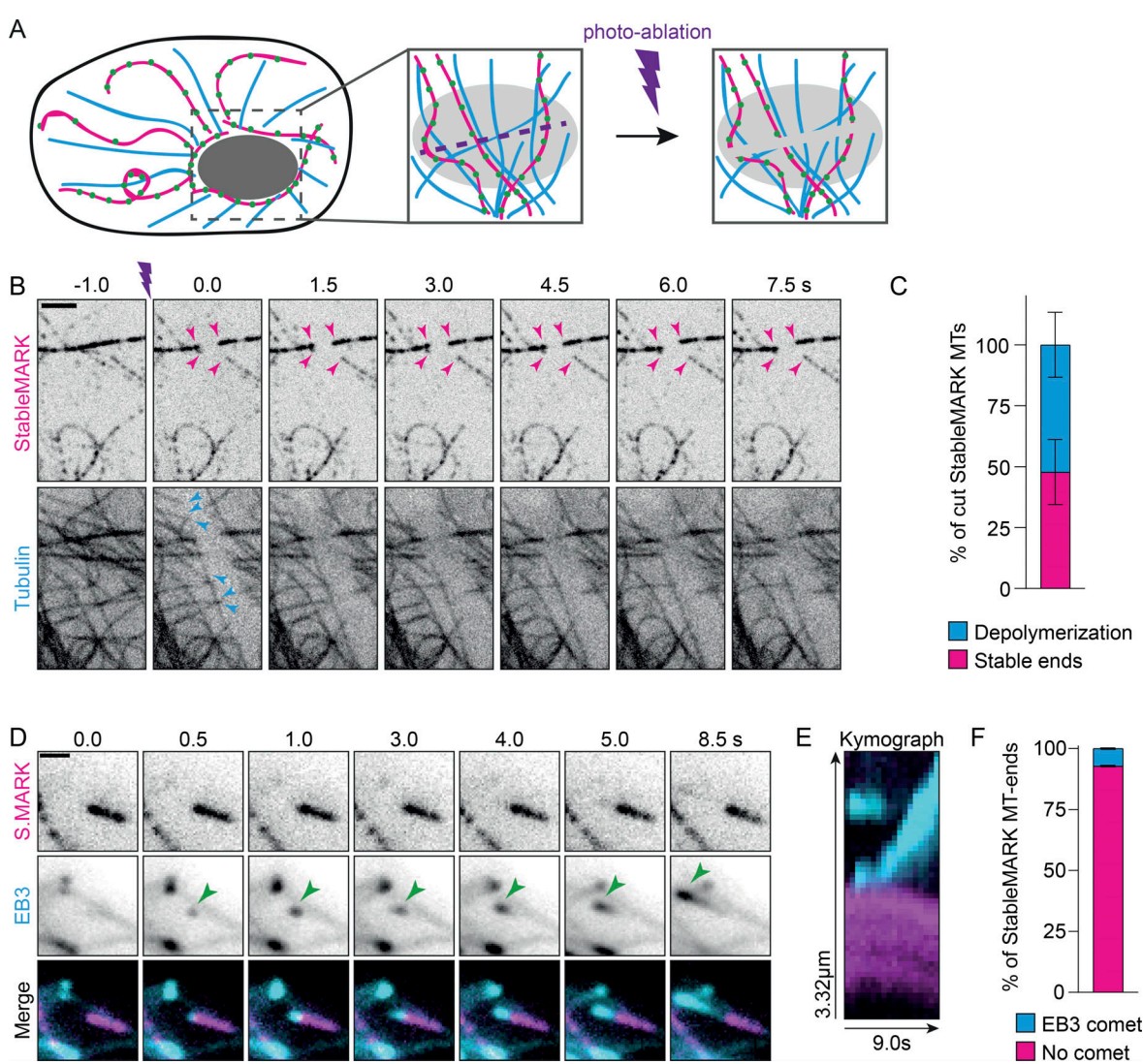

Figure 6.   **Properties of stable MTs. (A)** Cartoon: Stable and dynamic MTs in the nuclear area are severed by a focused laser beam that is moved along a line. **(B)** Stills from live-cell imaging of U2OS cells expressing StableMARK and mCherry-tubulin. Purple marker indicates photo-ablation. Magenta arrowheads in top panels indicate examples of freshly generated, StableMARK-decorated MT-ends that remain stable upon cutting. Cyan arrowheads in the bottom panel point toward freshly generated MTs in the total tubulin channel that depolymerize upon cutting (see also Video 7). **(C)** Stacked bar graph representing the percentage of cut StableMARK-MTs that have stable ends vs. depolymerizing ends. Mean ± SD of in total 119 cut StableMARK-MTs from 36 cells from four independent experiments are shown. **(D)** Stills from live-cell imaging of U2OS cell expressing StableMARK and EB3-tdtomato (see also Video 8). **(E)** Kymograph showing EB3 comet growing at a StableMARK-decorated MT-tip as in D. **(F)** Stacked bar graph showing the percentage of StableMARK-decorated MT-ends with/without EB3 comet. Mean ± SD are shown. 97 StableMARK-decorated MT-tips from 35 cells from three independent experiments were included in this analysis. Scale bars, 2.5 µm (B and G), 1 µm (D).

giving these a higher polymerization speed and catastrophe frequency than detyrosinated MTs in vitro (Chen et al., 2021). To rule out that this was an artifact caused by expression of our marker (or the presence of long dynamic MT stretches), we performed super-resolution microscopy on non-transfected U2OS cells stained for acetylated tubulin (as a marker for stable MTs) and total tubulin. We scored the occurrence of total tubulin stretches (without acetylation) at the tips of acetylated MTs and found that only 11% of these MT-tips had such stretches. This indicates that no polymerization had taken place at the tips of the majority of acetylated (and likely stable) MTs, in good agreement with our live-cell data (Fig. S5, B–D). These findings are in line

with earlier work that showed that, in TC7 cells, MTs that are detyrosinated and turn over slowly also do not serve as templates for MT growth (Infante et al., 2000; Webster et al., 1987).

While imaging our live-cell marker together with mCherry-tubulin using TIRF microscopy, we observed transient interactions of StableMARK and MTs without stably bound StableMARK molecules. Quantification of these interactions revealed an average dwell time of 0.65 s (Fig. 7, A–E; Video 9). A maximum intensity projection of a longer time lapse in a region devoid of StableMARK-decorated, stable MTs, revealed the appearance of MT shadows in the StableMARK channel, which exactly mirrored the MTs present in the total tubulin channel

Figure 7. **Differential binding of StableMARK molecules to stable and dynamic MTs. (A)** Live U2OS cell expressing StableMARK and mCherry-tubulin. **(B)** Stills from live-cell imaging of region indicated with a white dashed box in A (see also Video 9). **(C)** Maximum projection of 25 frames (5 s) of the region indicated with a white dashed box in A. **(D)** Kymographs of StableMARK channel of red dashed boxes in A. Pink ovals in the right kymograph indicate examples of short binding events of StableMARK to a dynamic MT. **(E)** Cumulative histogram ± SD of dwell time of StableMARK on dynamic MTs (5,006 events from 23 cells from three independent experiments). Green line shows fit with $y = (y_1-y_2) * e^{(-k*x)} + y_2$, yielding $\tau = 1/k = 0.65$ s. **(F)** Live U2OS cell expressing StableMARK and mCherry-tubulin. Maximum projection of red dashed box represents 123 frames (122 s). Orange arrowheads point toward a dynamic MT with which StableMARK molecules have transiently interacted. **(G)** Cartoon: StableMARK molecules stably bind stable MTs ($\tau_{bound}$: min—h), whereas StableMARK molecules interact transiently with dynamic MTs ($\tau_{bound}$: ms—s). Scale bars, 2.5 μm (A), 1 μm (F-zoom), 2 μm (B, C, and F). Time, s:ms.

(Fig. 7 F). Based on this data, we propose that StableMARK molecules randomly sample all MTs in the cell, but only become locked in their high-affinity state when encountering a stable MT (Fig. 7 G). This agrees with our in vitro data suggesting that StableMARK prefers MTs with an expanded lattice; perhaps, StableMARK only adopts a two-head-bound or strongly bound state on stable MTs due to their specific lattice structure.

Finally, we exploited our live-cell marker to study the dynamics of stable MTs during the transition from interphase to mitosis and back. To increase experimental efficiency and achieve consistent, low expression levels of StableMARK, we generated a U2OS Flp-In cell line that stably expresses StableMARK upon the addition of doxycycline. To visualize MT network dynamics during the transition from interphase to prophase, we transfected U2OS;StableMARK cells with mCherry-tubulin and treated the cells with doxycycline. Using spinning-disk microscopy, we selected cells in early prophase based on the presence of the developing spindle poles in the total tubulin channel and followed these over time in 3D. Initially, the developing spindles were StableMARK-negative, and the StableMARK-positive MTs appeared fragmented (Fig. 8 A). As the cell progressed through prophase, more StableMARK-decorated MTs appeared in the spindle, indicating local MT stabilization (Fig. 8 A). Strikingly, we also observed the transport of StableMARK-positive MT fragments into the developing spindle (Fig. 8, A and B). The final

distribution of StableMARK in the metaphase spindle suggested that it mostly localized to kinetochore fibers (Fig. 8 C; Video 10). A cold treatment assay (Rieder 1981; Ohta et al., 2015) confirmed the localization of StableMARK to kinetochore fibers (Fig. S6 A).

For anaphase and telophase, earlier work has demonstrated the increased stability of midzone MTs using nocodazole or cold treatment, but the exact onset and time course of stabilization have remained unknown (Landino and Ohi 2016; Hu et al., 2011; Murthy and Wadsworth 2008). We therefore set out to explore changes in MT stability within the spindle during these later stages of mitosis. We selected U2OS;StableMARK cells treated with doxycycline in metaphase and followed them over time using 3D spinning-disk microscopy (because cells in metaphase were particularly sensitive to cellular manipulations and phototoxicity, we only imaged StableMARK). While StableMARK was initially localized to kinetochore fibers (Fig. 8 C and Fig. S6 A; Video 10), it started to appear in the emerging spindle midzone shortly after the onset of anaphase. This was followed by a much stronger enrichment later on when the midzone MTs started to compact to form the cytokinetic bridge (Fig. 8, C, D, and F; and Fig. S6 D; Video 10). This is consistent with earlier reports indicating that midzone MT stabilization coincides with cleavage furrow ingression (Hu et al., 2011; Landino and Ohi 2016; Salmon et al., 1976; Murthy and Wadsworth 2008). We confirmed stabilization of midzone MTs by analyzing the ratio of

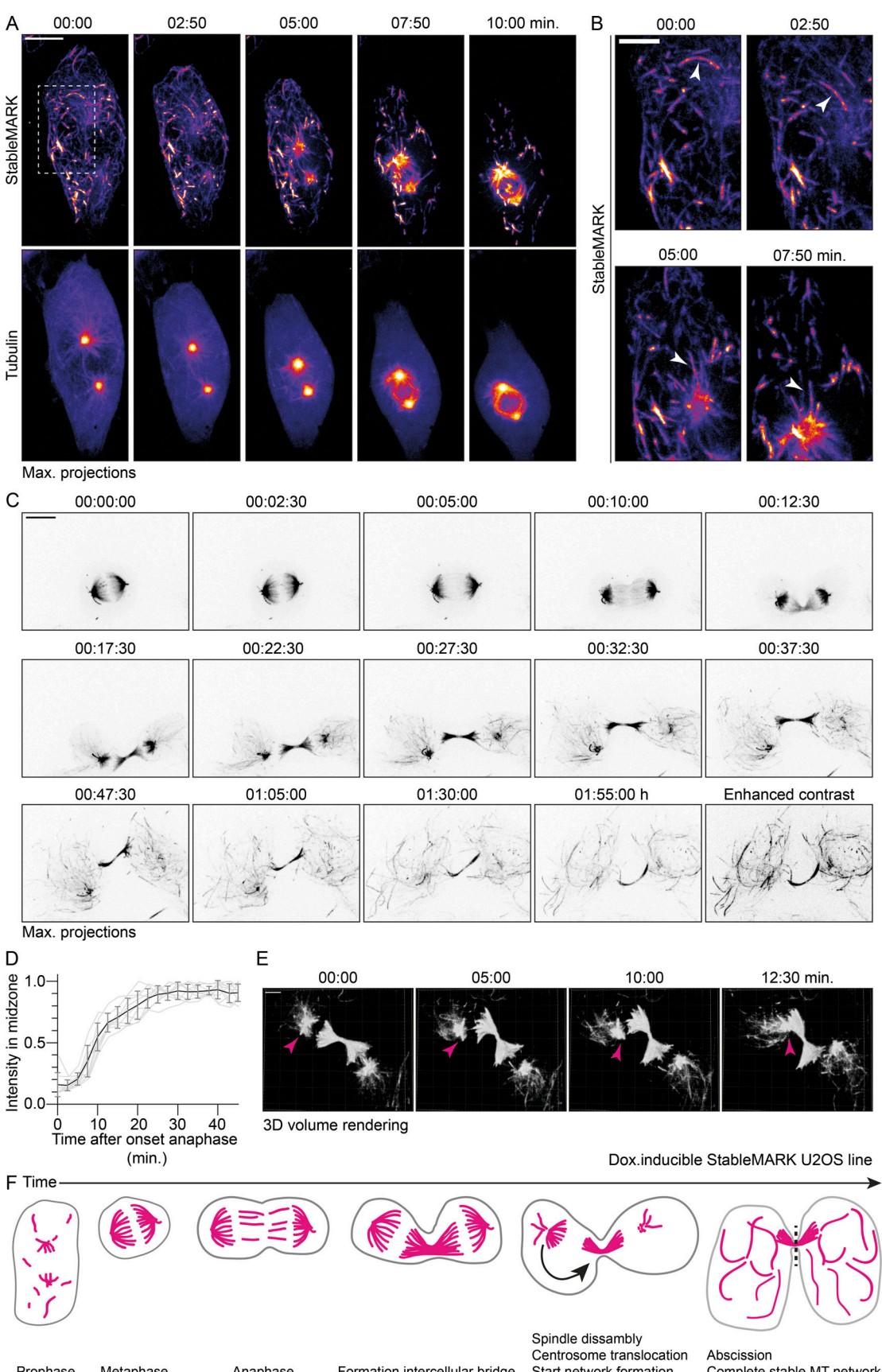

Figure 8. **Dynamics of stable MT before, during, and after cell division. (A)** Stills from live-cell imaging of U2OS:StableMARK cells treated with doxycycline to induce expression and transiently transfected with mCherry-tubulin. For every time point, a maximum intensity projection of a Z-stack of 25 slices with 0.75

µm step size is shown. **(B)** Zooms of region indicated with white, dashed box in A. White arrowheads point toward a StableMARK-decorated MT that is transported toward the developing spindle. Time, min:s. **(C)** Stills from live-cell imaging of U2OS:StableMARK cell(s) treated with doxycycline to induce expression. For every time point, a maximum intensity projection of a Z-stack of 25 slices with 0.75 µm step size is shown. Last time point is also displayed with enhanced contrast. Time, h:min:s (see also Video 10). **(D)** Graph showing the change in normalized fluorescent intensity of StableMARK in the midzone over time. Events were aligned on the onset of anaphase, which was set to $T = 0$. Mean (black line) ± SD and individual trace (light gray) from nine cells from three independent experiments are shown. **(E)** Stills from live-cell imaging of U2OS:StableMARK cell(s) treated with doxycycline to induce expression. For every time point, a Z-stack of 25 slices with 0.75 µm step size is shown as a 3D volumetric render. Magenta arrowheads indicate translocation of spindle remnant toward the intercellular bridge. Time: min:s. **(F)** Cartoon illustrating the organization of the stable MT array from prophase until abscission of the intercellular bridge. Scale bars, 10 µm (A and C), 5 µm (B), 4 µm (E).

acetylated tubulin (as a marker for stable MTs) over total tubulin in non-transfected U2OS cells during cytokinetic bridge formation. Indeed, an increase in the ratio of acetylated tubulin over total tubulin could be observed during midzone formation, suggesting local MT stabilization (Fig. S6, B and C).

After formation of the cytokinetic bridge, three processes could be observed. First, new MTs decorated by StableMARK started to appear from the spindle remnants. Then, the translocation of one of the two spindle remnants toward the cytokinetic bridge could be observed (Fig. 8, C, E, and F; Video 10), which was followed by further disassembly of the spindle remnants (Fig. 8, C and F; Video 10). Over time, the spindle remnants completely disassembled and more StableMARK-decorated MTs gradually appeared throughout the cell, until an interphase network of stable MTs was formed (Fig. 8, C and F; Video 10). Eventually, abscission of the cytokinetic bridge took place, and sometimes the release of the midbody could be observed (Fig. S6 E). These results demonstrate how our novel live-cell marker can be used to study the spatiotemporal regulation of specific MT-subsets throughout the cell cycle.

## Discussion

Recent studies have started to unravel the importance of MT subsets in the spatial organization of complex cells such as neurons (Burute and Kapitein 2019; Akhmanova and Kapitein 2022); however, these studies relied on detecting subsets in fixed cells and could not directly observe the differential dynamics of different types of MTs. Existing approaches to visualize stable MTs in living cells relied on imaging of a marker for total tubulin after nocodazole treatment (Xu et al., 2017), which leaves cells in very non-physiological conditions. Alternatively, researchers have used correlative live-cell and immunofluorescence microscopy (Cai et al., 2009), which is labor-intensive and error-prone given the continuous reorganization of the MT cytoskeleton. Here, we introduced StableMARK as a live-cell marker for stable MTs that allows for the direct visualization of the dynamics of stable MTs. Using a series of pharmacological treatments, we showed that, although StableMARK localizes with high specificity to the subset of MTs that are acetylated in control conditions, it does not detect acetylation (or detyrosination) per se. Instead, it appears to recognize a specific expanded lattice conformation, allowing it to label stable, long-lived MTs in cells, many of which are nocodazole resistant.

Our approach to visualize stable MTs in living cells relies on the overexpression of a truncated and fluorescently tagged MT-binding protein. Similar strategies have been used before to label all MTs in the cell via overexpression of an N-terminal fragment of Ensconsin/MAP7 (Faire et al., 1999; Guo et al., 2018), to label dynamic MTs using a nanobody against tyrosinated tubulin (Kesarwani et al., 2020), and to label MT plus-ends through the use of a truncated EB3 interaction partner (Komarova et al., 2002). In all these cases, high levels of overexpression cause artifacts (Faire et al., 1999; Kesarwani et al., 2020; Komarova et al., 2002). We, therefore, characterized the effects of StableMARK expression on the MT cytoskeleton and found that the MT cytoskeleton was not overstabilized or otherwise perturbed by low expression levels of our marker. For example, StableMARK-decorated MTs were still responsive to external cellular cues: they disappeared upon prolonged serum starvation and reappeared upon addition of serum. In addition, organelle transport could still take place along StableMARK-decorated MTs. Finally, we successfully generated a stable cell line with controlled expression levels that still displays proper mitosis. Although we cannot completely rule out minor disruptions of cellular physiology, these findings suggest that our marker faithfully captures the dynamics of stable MTs.

Our experiments allowed us to confirm many of the known properties of stable MTs and additionally provided new insights. For example, while it has been previously reported that most stable MTs do not have a dynamic plus-end (Schulze and Kirschner 1986; Webster et al., 1987; Infante et al., 2000; Palazzo et al., 2001), our work reveals that long-lived MTs have varying degrees of lattice stabilization, with about half of them not depolymerizing upon laser-induced severing. This variation among StableMARK-decorated MTs suggests that stabilization starts by modifying the MT plus-end to create a long-lived MT, followed by gradual stabilization of the rest of the lattice, for example, by the accumulation of MAPs and/or PTMs, which might eventually also alter the lattice of the MT, inducing lattice expansion. This initial stabilization might start through altering the protein composition at the plus-end; for example, we did not observe EB comets at the ends of most StableMARK-positive MTs.

Fast single-molecule TIRF microscopy revealed that StableMARK motors randomly bind to all MTs. However, StableMARK molecules quickly detach from dynamic MTs while remaining stably bound to long-lived MTs. These findings suggest that the surface of stable MTs is different from the surface of dynamic MTs and that this differentiation of the MT surface is recognized by StableMARK and prompts them to adopt a high MT-affinity state. Consistently, Taxol, an MT stabilizer that induces a small longitudinal expansion of the MT lattice (Estévez-Gallego et al., 2020; Rai et al., 2020; Alushin et al., 2014; Vale et al., 1994),

promotes the binding of StableMARK to all MTs in cells and increases its binding to MTs in vitro. While our in vitro experiments revealed a clear preference for StableMARK for expanded lattices, we did not observe rapid detachment from compacted lattices, suggesting that MT binding and unbinding of StableMARK in cells depends on additional regulatory factors. For example, recent in vitro work has shown that different MAPs can also sense or modulate MT lattice spacing, and some of these could in turn regulate motor protein behavior (Siahaan et al., 2022; Liu and Shima 2022 Preprint).

Using our live-cell marker, we furthermore visualized the spatiotemporal dynamics of stable MTs during different stages of the cell cycle, such as the transition from interphase to metaphase and the transition from anaphase back to interphase, when new networks of stable MTs formed in the daughter cells. In the developing spindle, we found initial evidence for the inclusion of pre-existing stable MTs. We furthermore directly visualized the rapid and remarkably stereotyped enrichment of StableMARK in the spindle midzone during cleavage furrow ingression, highlighting the additional stabilization of the specialized MT array of the midzone (Landino and Ohi 2016; Murthy and Wadsworth 2008). In addition, we demonstrate how the stable interphase MT network emerges from the spindle remnants. Importantly, we could also directly visualize how one of the half-spindle remnants often repositions to the cytoplasmic bridge formed by the midzone MTs after anaphase (Piel et al., 2001) and the release of the spindle midbody during abscission (Peterman and Prekeris 2019). The role of different MT subsets during cell division is often acknowledged yet understudied (Barisic and Maiato 2016), and our live-cell marker offers new opportunities to study the dynamics and function of MT subsets during different stages of cell division.

The live-cell marker for stable MTs introduced here can readily be targeted to specific cell types (Fig. S7) and should be possible to use within various model organisms using cell-type specific promotors. Inducible gene expression systems or synthetic upstream open reading frames (Ferreira et al., 2013) could be used to further fine-tune StableMARK expression as required for the model system of choice. We therefore anticipate that the live-cell marker introduced in this work will help to understand how different MT subsets contribute to cellular organization and transport in different cell types. In addition, it will help to unravel how complex MT arrays with multiple subsets are formed.

# Materials and methods
## Cell culture and transfections
U2OS, COS-7, Vero-E6, HeLa, Caco-2, and Swiss 3T3 cells were cultured in DMEM supplement with 10% FBS and 50 µg/ml penicillin/streptomycin at 37°C and 5% $CO_2$. U2OS Flp-In T-Rex cells were cultured in DMEM supplement with 10% FBS, 50 µg/ml penicillin/streptomycin, and 15 µg/ml blastidicin. U2OS, Vero-E6, and HeLa cells were purchased from ATCC; Swiss 3T3 and COS-7 cells were a kind gift from Anna Akhmanova; U2OS Flp-In T-Rex cells were a kind gift from Prof. Alessandro Sartori (Institute of Molecular Cancer Research, University of Zurich, Zurich, Switzerland); and Caco-2 cells were a kind gift from

Sven van Ijzendoorn (University Medical Center Groningen, Groningen, Netherlands). Cells were confirmed to be free of mycoplasma. Cells were plated on 18-mm coverslips (immunolabeling experiments) or 25-mm coverslips (live-cell experiments) on the day of or 1 or 2 d before transient transfection. Cells were transiently transfected using Fugene6 transfection reagent (Promega) according to the manufacturer's protocol. After 1 d of transfection, cells were used for live-cell imaging, subjected to treatment (drugs/serum starvation), or fixated. For all experiments that involved the expression of mCherry-tubulin, cells were used for live-cell imaging after 2 d of transfection. The isogenic U2OS Flp-In cell line that upon doxycycline-induction stably expresses hKif5b(1–560)G234A-mNeongreen-mNeongreen was derived from the U2OS Flp-In cell line by transfection with the pCDN5/FRT/TO vector (Invitrogen) and pOG44 vector (Invitrogen). The U2OS Flp-In hKif5b(1–560)G234A-L-mNeongreen-L-mNeongreen cell line was cultured in DMEM supplemented with 10% FBS, 50 µg/ml penicillin/streptomycin, 15 µg/ml blastidicin, and 0.5 mg/ml hygromycin B (ant-hg-1, InvivoGen). To induce expression of hKif5b(1–560)G234A-L-mNeongreen-L-mNeongreen, 10 ng doxycycline (ab141091; Abcam) was added to the cells 16–24 h before imaging. U2OS Flp-In hKif5b(1–560)G234A-L-mNeongreen-L-mNeongreen cells were plated on 25-mm coverslips ≥2 d before imaging.

## Drug treatments
For the fixed drug treatment experiments, U2OS cells were seeded on 18-mm coverslips. The next day, cells were incubated with 0.1% DMSO (3 h), 10 µM Taxol (#T7402; Sigma-Aldrich; 2 h), 10 µM tubacin (BML-GR362; Enzo Life Sciences; 2 h), 10 µM nocodazole (Cat#M1404; Sigma-Aldrich; 1 h), 10 µM Taxol (2 h) followed by 10 µM Taxol + 10 µM nocodazole (1 h), or 10 µM tubacin (2 h) followed by 10 µM tubacin + 10 µM nocodazole (1 h) in full medium and subsequently fixed. For the detyrosination assay, U2OS cells were seeded on 18-mm coverslips and transfected with VSH1-GFP and SBVP-FLAG the next day. 24 h after transfection, cells were treated with 0.1% DMSO (1 h) or 10 µM nocodazole (1 h) and subsequently fixed. For the drug treatments of StableMARK-expressing cells, U2OS cells were seeded on 18-mm coverslips and transfected with StableMARK the next day. 24 h after transfection, cells were treated with 0.1% DMSO (2 h), 10 µM Taxol (2 h), or 10 µM tubacin (2 h) and subsequently fixed. For the nocodazole treatment of StableMARK-expressing cells during live-cell imaging, U2OS cells were plated on 25-mm coverslips and transfected the next day with StableMARK and mCherry-tubulin. The following day, nocodazole was added with a final concentration of 10 µM to the live cells on stage during a time-lapse acquisition. For live-cell imaging of lysosomes, cells were incubated with 1 µM SiR-lysosome (Spirochrome) and 10 µM Verapamil for 1 h and subsequently imaged in a medium without SiR-lysosome and Verapamil. For cold treatment, cells were incubated on ice for 10 min and subsequently extracted and fixed with precooled reagents.

## Serum starvation assay
For the serum starvation assay, Swiss 3T3 fibroblasts were seeded on 18-mm coverslips. The next day, cells were

transfected with StableMARK. 24 h later, cells were washed 1× with starvation medium (DMEM supplemented with 50 µg/ml penicillin/streptomycin) or full medium (DMEM supplement with 10% FBS and 50 µg/ml penicillin/streptomycin) and then incubated in starvation medium or full medium, respectively. 53 h later, cells were washed 1× with starvation medium or full medium and were consecutively subjected to (a) fixation, (b) removal of starvation medium and addition of full medium, or (c) continuously incubated in fresh starvation medium or full medium. 8 h later, cells were fixed.

### Plasmids and cloning
mCherry-α-tubulin (Kapitein et al., 2010), EB3-tdTomato (#50708; Addgene), and FKBP-mCherry-Rab6A (Schlager et al., 2014) were described before. The presence of the N-terminal FKBP domain in the Rab6a construct has no detectable effects on the behavior of this marker (Schlager et al., 2014; Serra-Marques et al., 2020). βtubulin-GFP was a kind gift from Anna Akhmanova (Utrecht University, Utrecht, Netherlands). VSH1-FLAG, VSH1-GFP, and SVBP-FLAG were described before (Nieuwenhuis et al., 2017) and were a kind gift from Thijn Brummelkamp (Netherlands Cancer Institute, Amsterdam, Netherlands).

hKif5b(1-560)-rigor-L-mNeonGreen-L-mNeonGreen was cloned into the mammalian expression vector pβactin-16-pl (chicken β-actin promoter; Kaech et al., 1996) and generated by a combination of PCR-based cloning and Gibson assembly. The G234A rigor mutation was described previously (Rice et al., 1999). mNeonGreen (Shaner et al., 2013) and flanking linker sequences were provided by Allele Biotechnology. We also generated pβactin-16-pl- hKif5b(1-560)-rigor-L-GFP-L-SspB(nano). Here, the GPF domain is flanked by synthetic, 29 amino acids GGGS linkers (Nijenhuis et al., 2020).

For bacterial expression and purification, hKif5b(1-560)-rigor-L-mNeonGreen was amplified by PCR and cloned into the pGEX-6p-1 backbone by Gibson assembly, generating GST-hKif5b(1-560)-rigor-L-mNeonGreen. The construct was verified by sequencing.

To generate a stable, isogenic U2OS Flp-In cell line, hKif5b(1-560)G234A-L-mNeonGreen-L-mNeonGreen was subcloned into pCDNA5/FRT/TO (Invitrogen) via Gibson assembly. The FLP recombinase expression vector is encoded in pOG44 (Invitrogen).

### Protein purification
For in vitro assays, GST-tagged rigor was purified from *Escherichia coli* BL21 cells. Briefly, after transformation, bacteria were cultured until OD600 ≈ 0.7 at 37°C. Cultures were cooled, after which protein expression was induced with 0.15 mM IPTG at 18°C overnight. Cells were then pelleted by centrifugation at 4,500 × $g$, snap-frozen in liquid nitrogen, and stored at –80°C until use. Cells were rapidly thawed at 37°C before being resuspended in chilled lysis buffer (1 × PBS supplemented with 0.1% [vol/vol] Tween 20, 250 mM NaCl, 5 mM MgCl$_2$, 1 mM DTT, 0.5 mM ATP, and 1 × EDTA-free cOmplete protease inhibitor; pH 7.4). Bacteria were lysed by sonication (five rounds of 30 s), supplemented with 2 mg/ml lysozyme, and then incubated on

ice for 45 min. The lysate was clarified by centrifuging at 27,000 × $g$ for 30 min and filtered through a 0.22-mm pore size filter before being incubated with equilibrated Glutathione Sepharose 4B resin for 2 h. Beads were then pelleted and resuspended in five column volumes (CV) wash buffer (1 × PBS supplemented with 0.1% [vol/vol] Tween 20, 250 mM NaCl, 5 mM MgCl$_2$, 1 mM DTT, and 0.5 mM ATP; pH 7.4) and transferred to a BioRad column. Once settled, the resin was washed three times with 10 CV wash buffer and once with 10 CV cleavage buffer (50 mM Tris-HCl supplemented with 0.1% [vol/vol] Tween 20, 100 mM NaCl, 5 mM MgCl$_2$, 1 mM EDTA, 1 mM DTT, and 0.5 mM ATP; pH 8.0). Then, 80 units of PreScission Protease in three CV cleavage buffer were added and the column was sealed and incubated overnight with rotation for removal of the GST tag. The following morning, once the resin was settled, the eluent was collected, concentrated by spinning through a 3,000 kD MWCO filter, supplemented with 0.5 mM ATP, 1 mM DTT, and 10% [wt/vol] sucrose, aliquoted, flash-frozen in liquid nitrogen, and stored at –80°C. Concentration was determined with a BSA standard gel. All steps from lysis onwards were performed at 4°C.

### Fluorescence microscopy
For live-cell imaging experiments, coverslips were mounted into metal imaging rings and immersed in full medium with (imaging of green fluorescence only) or without phenol red (imaging of green and red fluorescence). The U2OS Flp-In hKif5b(1-560) G234A-L-mNeongreen-L-mNeongreen cells were imaged in DMEM containing phenol red supplemented 10% FBS, 50 µg/ml penicillin/streptomycin, 15 µg/ml blasticidin, 0.5 mg/ml hygromycin B, and 10 ng doxycycline. TIRF microscopy images (azimuthal spinning TIRF) were acquired on an inverted research Nikon Eclipse Ti-E microscope (Nikon) equipped with a perfect focus system (Nikon), ASI motorized stage MS-2000-XY (ASI), Apo TIRF 100× 1.49 NA oil objective (Nikon), iLas2 system (Roper Scientific, now Gataca systems), and Evolve Delta 512 EMCCD camera (Photometrics). The microscope setup was controlled by MetaMorph software 7.8 (Molecular Devices). For imaging, Stradus 488 nm (150 mW; Vortran) and OBIS 561 nm (100 mW; Coherent) lasers were used together with the ET-GFP filter set (49002; Chroma) or ET-GFP/mCherry filter set (59022; Chroma). Optosplit III beamsplitter (Cairn Research Ltd) equipped with a double emission filter cubed with ET525/50 m, ET630/75 m, and T585LPXR (Chroma) was used during simultaneous imaging of green and red fluorescence. Images were projected onto the EMCCD chip with a 2.5× intermediate lens (Nikon C mount adapter 2.5×) at a magnification of 0.065 µm/pixel and 16-bit pixel depth. To keep cells at 37°C, we used a stage top incubator (model INUBG2E-ZILCS; Tokai Hit). Images were acquired at 30-s interval (Fig. 3 B), 1-s interval (Fig. 3, A, C, and E; and Fig. S3, A–C), and at a frame rate of 2 (Fig. 6, B and D), 5 (Fig. 7, A–D), or 10 frames/s (Fig. 5 G).

The same microscope, filters, and lasers were used for the in vitro assays (Fig. S2), but images were acquired with a CoolSNAP MYO CCD camera (Teledyne Photometrics) with a pixel size of 0.045 µm/pixel, and the samples were kept at 30°C for these experiments. A single image of MTs was acquired at

each position, and a 1-min video with a 1-s interval was acquired for the rigor.

For Video 1, the live nocodazole assay, for assessing EB3/lysosome/rab6 dynamics and following StableMARK-expressing cells through the cell cycle, images were acquired with spinning-disk confocal microscopy on an inverted research microscope Nikon Eclipse Ti-E (Nikon), equipped with the perfect focus system (Nikon), Plan Fluor 40× 1.3 NA oil objective (Nikon)/Plan Apo VC 60× 1.4 NA oil objective (Nikon), and a spinning-disk-based confocal scanner unit (CSU-X1-A1; Yokogawa). The system was also equipped with ASI motorized stage with the piezo plate MS-2000-XYZ (ASI), Photometrics Evolve Delta 512 EMCCD camera (Photometrics), and controlled by MetaMorph 7.8 software (Molecular Devices). For imaging, Stradus 488 nm (150 mW; Vortran), 561 Jive (100 mW; Cobolt), and Vortran Stradus 643 lasers were used, together with ET-GFP (49002; Chroma), ET-mCherry (49008; Chroma), and ET-Cy5 (49006; Chroma) filter sets. For simultaneous imaging of green and red fluorescence, ET-GFP/mCherry (59022) filter set together with DV2 beamsplitter (Photometrics) were used. Images were projected onto the EMCCD chip with intermediate lens 2.0× (Edmund Optics) at a magnification of 0.164 µm/pixel (40×) or 0.110 µm/pixel (60×) and 16-bit pixel depth. To keep cells at 37°C, we used a stage top incubator (model INUBG2E-ZILCS; Tokai Hit). Video 1 was acquired using the 60× objective at an imaging interval of 1.5 s. The live nocodazole assay was acquired using the 40× objective at an imaging interval of 2.5 min (Fig. S1 H). EB3 dynamics were captured using the 60× objective with a stream acquisition of 2 frames/s (Fig. 4, E and F). Lysosome and Rab6a dynamics were captured using the 60× objective with a stream acquisition of 10 frames/s (Fig. 5, C–F). The time-lapse acquisitions showing the establishment of a stable MT network after metaphase were acquired using the 60× objective at an imaging interval of 2.5 min (Fig. 8, A–D; and Fig. S6, A and B). A Z-stack of 25 slices and 0.75 µm Z-steps was acquired at every time point.

For the photoablation experiments, the same setup was used. Photoablation was performed with FRAP/PhotoAblation scanning system iLas (Roper Scientific France, now Gataca systems) mounted on a custom-ordered illuminator (MEY10021; Nikon) and 355 nm passively Q-switched pulsed laser (Teem Photonics) combined with S Fluor 100× N.A. 0.5–1.3 oil objective (Nikon). 16-bit images were projected onto the EMCCD chip at a magnification of 0.066 µm/pixel. Images were acquired at 2 frames/s (Fig. 6 B and Fig. S7 A).

Immunolabelled cell samples were acquired on a Zeiss Axio Observer Z1 LSM700 (Zeiss) using the Plan-apochromat 63×1.4 NA oil DIC objective (Zeiss). For the acetylation, EB1, and cytochrome C control experiments, transfected and non-transfected cells from the same coverslip were randomly selected for imaging. Illumination settings were chosen so that low-StableMARK-expressing cells could be detected, while at the same time, no pixel saturation occurred for high-expressing cells. Illumination settings were kept the same for all images acquired from a particular coverslip. For all other fixed experiments, illumination settings were kept similar for all conditions.

The super-resolution images shown in Fig. S5, B and C, were acquired on a Leica TCS SP8 STED 3× microscope (Leica), equipped with a pulsed (80 MHz) white-light laser, HyD detectors, and a HC PL APO 100× 1.40 NA Oil STED WHITE objective (15506378; Leica). For the dyes Alexa 594 (A11007; Moleuclar Probes/Life Technologies) and Abberior STAR 635P (ST635P-1002; Abberior GmbH), we used 580 and 633 nm laser lines for excitation, and a 775 nm synchronized pulsed laser for depletion, with a time gating of 0.3–8.5 ns and 0.3–3.8 ns, respectively. Emission detection windows were 590–622 nm for Alexa 594 and 636–768 nm for Abberior STAR 635P. The STED images were acquired in line-sequential mode.

**Immunofluorescence**

For immunofluorescence experiments, different fixation methods were exploited depending on the antibodies used. For immunocytochemistry of MTs, cells were extracted for 1 min in prewarmed extraction buffer (0.3% Triton X-100 and 0.1% glutaraldehyde in MRB80 buffer (MRB80 buffer: 80 mM Pipes, 1 mM EGTA, and 4 mM $MgCl_2$, pH 6.8) and subsequently fixed in prewarmed 4% PFA in PBS for 10 min. For immunocytochemistry of cytochrome C, cells were fixed in prewarmed 4% PFA in PBS for 10 min. For immunocytochemistry of EB1 and detyrosinated MTs, cells were fixed in ice-cold methanol for 10 min. Samples prepared for STED imaging were extracted for 1 min in pre-warmed extraction buffer (0.3% Triton X-100 and 0.1% glutaraldehyde in MRB80 buffer and subsequently fixed in prewarmed 4% PFA (15,170; Electron Microscopy Sciences) in MRB80 buffer for 10 min. After fixation, cells were washed with PBS, permeabilized with 0.25% Triton X-100 in PBS, washed again with PBS, and subsequently blocked for 1 h with 3% BSA in PBS. Cells were incubated with primary antibody diluted in 3% BSA in PBS for 1 h at RT, washed with PBS, and incubated with secondary antibody diluted in 3% BSA in PBS for 1 h at RT. Samples prepared for STED imaging were incubated for 2 h in primary and 2 h in secondary antibody. After washing with PBS, cells were dipped in MilliQ water, air-dried, and mounted on microscopy slides using Prolong Diamond (Molecular Probes). The following primary antibodies were used in this study: Cytochrome C (6H2.B4; BD Biosciences), EB1 (5/EB1 BD Biosciences), acetylated tubulin (6-11B-1; Sigma-Aldrich), α-tubulin (EP1332Y; Abcam), α-tubulin (B-5-1-2; Sigma-Aldrich), tyrosinated tubulin (YL1/2; Abcam), detyrosinated tubulin (AB3210; Merck), and GFP (GFP-1010; Aves Lab). DAPI (Molecular Probes) was used to visualize DNA.

**FCS**

FCS measurements were performed on the Leica TCS SP8 STED 3× microscope (Leica), equipped with a pulsed (80 MHz) white-light laser, HyD detectors, and using a HC PL APO 86× 1.2 NA W motCORR STED (15506333; Leica) water-immersion objective with correction collar. Cells were kept at 37°C during imaging using a Ludin Cube. The microscope was operated with Leice Application Suite, Advanced Fluorescence software in FCS mode. For FCS measurements, the microscope was connected to a PicoHarp 300 stand-alone TCSPC Module (PicoQuant) operated from SymPhoTime 64 software (PicoQuant). We validated

the dynamic range of our FCS setup by measuring a dilution series of fluorescein sodium salt (518-47-8; Sigma-Aldrich) in PBS and concluded that we could reliably measure concentrations ranging between 10 nM and 100 μM. For FCS measurements of StableMARK in living cells, U2OS cells were plated on #1.5 18-mm coverslips. The correction collar of the objective was adjusted for every imaged coverslip. 488 nm laser line for excitation and an emission detection window of 500–600 nm were used. Cells were incubated in 10 μM nocodazole for 1 h before the start of FCS measurements. By eye, cells were selected that were classified as low, medium, or high expressing. Per cell, three FCS measurements were performed at different subcellular locations outside the nucleus. The triplet stage model was used to fit the FCS traces and calculate the intracellular concentration of StableMARK.

### Dual-chamber in vitro assay

Double-cycled MT seeds were prepared by combining TRITC-labeled (49%), biotinylated (18%), and unlabeled tubulin (33%; Cytoskeleton) reconstituted in MRB80 (80 mM K-Pipes, 1 mM EGTA, 4 mM $MgCl_2$; pH 6.80 with KOH) to a final concentration of 20 μM with 1 mM GMPCPP (Jena Bioscience) on ice. The mixture was incubated at 35°C for 30 min to allow MTs to polymerize. Seeds were pelleted by centrifugation in an airfuge (Beckman coulter) at 20 psi for 5 min, resuspended in MRB80, and depolymerized on ice for 25 min. The tubulin was then repolymerized upon the addition of fresh GMPCPP by incubating at 35°C for 30 min. These seeds were pelleted by centrifugation in an airfuge at 20 psi for 5 min, resuspended and diluted sixfold in MRB80 supplemented with 10% [vol/vol] glycerol, aliquoted, flash-frozen in liquid nitrogen, and stored at –80°C until use.

To prepare the chambers, a clean glass coverslip was plasma-treated and fixed to a clean glass slide using strips of double-sided tape to create two parallel chambers of ~10 μl. The surface was blocked and functionalized by incubating with a mix of 95% PLL-g-PEG and 5% PLL-g-PEG-biotin (0.1 mg/ml in 10 mM Hepes, pH 7.40; SuSoS) for 10 min. After washing with MRB80 supplemented with 40% [vol/vol] glycerol (MRB80-gly40), NeutrAvidin was introduced and incubated for 10 min. After washing, 50-fold diluted GMPCPP seeds were introduced and incubated for 5 min before washing once more and then incubating with K-casein for >3 min.

All reaction mixtures (MT mix, expansion mix, rigor mix, washout mix) were prepared at double the volume for the paired compacted/expanded lattice samples and split into two equal parts prior to the addition of DMSO (compacted control) or 20 μM Taxol (expanded). Reagents were added to MRB80-gly40 such that the effective glycerol concentration in the MT mix was 20% and in the other mixes was ~27%. All mixes contained 0.1% [wt/vol] methylcellulose, 0.5 mg/ml K-casein, 50 mM glucose, 0.2 mg/ml catalase, 0.5 mg/ml glucose oxidase, and 10 mM DTT. The MT mix additionally contained 1 mM GTP, 10.8 μM porcine tubulin (Cytoskeleton), and 0.6 μM TRITC-labeled porcine tubulin (Cytoskeleton). The expansion mix additionally contained 50 mM KCl and 20 μM Taxol (or the equivalent dilution of DMSO). The rigor mix additionally contained 50 mM KCl, 20 μM

Taxol (or the equivalent dilution of DMSO), 2 mM ATP, and 15.2 pM StableMARK. The washout mixture additionally contained 50 mM KCl, 20 μM Taxol (or the equivalent dilution of DMSO), and 2 mM ATP. After preparation, these mixtures were spun in an airfuge at 20 psi for 5 min, transferred to clean tubes, and kept on ice until use.

Samples were then moved to the TIRF microscope equipped with a stage-top incubator to maintain them at a constant temperature of 30°C. MTs were grown by flowing in two chamber volumes (ChV) of the MT mix and letting it incubate for 15 min. Subsequently, the chambers were flushed with five ChV MRB80-gly40. Next, the lattices were (mock) expanded by adding two ChV expansion mix (or DMSO equivalent) and incubating for 10 min. Next, two ChV rigor mix was added and incubated for 90 s. Finally, four ChV washout mix was added before imaging. For imaging, the following sequence was used: 2 × Taxol, 4 × DMSO, 4 × Taxol, 4 × DMSO, and either 2 × Taxol or 4 × Taxol and 2 × DMSO (8 or 10 images/condition/assay), and images were taken at similar heights within the channels.

### Image processing and analysis

To prepare images and movies for publication, FIJI was used to adjust contrast levels and perform background corrections, for bleach correction using histogram matching, to generate maximum intensity projections, and to generate kymographs using the FIJI-plugin KymoResliceWide (https://github.com/ekatrukha/KymoResliceWide). Chromatic correction of dual-color live-cell imaging data was performed using the FIJI-plugin DoM_Utrecht (https://github.com/ekatrukha/DoM_Utrecht). 3D volume renders were generated in Arivis Vision 4D (v3.4.0). GraphPad Prism 9 was used for the fitting of data, statistical testing of data, and the generation of graphs.

To quantify the fold change in fluorescence for acetylated tubulin and α-tubulin upon treatment with different drugs, a region of interest (ROI) was drawn around every individual cell and around a region in the background and the mean gray value (MGV) for every ROI in both channels was measured. Data was transferred to Excel and the measured fluorescence intensities were background corrected. Subsequently, all values from the three independent experiments were pooled per channel and per condition. The average fluorescence intensity of acetylated tubulin and total tubulin in the DMSO condition were calculated and all background corrected; fluorescent intensity values were divided by the average value for acetylated tubulin and total tubulin of the DMSO condition, thus calculating fold changes in fluorescence intensity compared to the DMSO control. Data were checked for normality by Shapiro–Wilk test. Because data did not pass the normality test, Kruskal–Wallis test with Dunn's multiple comparisons test was used to compare the different conditions to the DMSO control. To quantify the fold change in fluorescence for α-tubulin and detyrosinated tubulin in non-transfected vs. VSH1/SVBP-expressing cells and to quantify the fold change in fluorescence for tyrosinated tubulin and acetylated tubulin in the serum starvation assay, an ROI was drawn around every individual cells and around a region in the background and fluorescence intensities were measured. Data were transferred to Excel and corrected for background and ROI

area. In the detyrosination assay, fold changes in fluorescence intensity for the VSH1/SVBP condition compared with the non-transfected control condition were calculated. For the serum starvation assay, fold change in fluorescence intensity for StableMARK-expressing cells compared to the non-transfected control condition were calculated per treatment regime (61 h full medium, 61 h serum-free medium [SFM], 53 h SFM + 8 h full medium).

Colocalization coefficients for StableMARK-expressing cells subjected to drug treatments were obtained using the FIJI-plugin JACoP (Bolte and Cordelières 2006). For colocalization of acetylated MTs with StableMARK-decorated MTs and vice versa, manually thresholded Manders' coefficients were calculated from one 12.5 × 12.5 µm ROI per cell. The same procedure was followed to calculate colocalization coefficients of detyrosinated MTs with StableMARK-decorated MTs and vice versa. Data were checked for normality by Shapiro–Wilk test. Because data did not pass the normality test, Kruskal–Wallis test with Dunn's multiple comparisons test was used to compare the colocalization coefficients of acetylated tubulin to StableMARK for the different conditions.

The speeds of StableMARK-MT movements, lysosomes, and Rab6a vesicles were quantified from stream acquisition that were acquired at a speed of 10 frames/s using kymograph analysis. Data were transferred to GraphPad Prism 9 and the frequency distribution of the data was calculated with a bin size of 0.1. Subsequently, data were fitted with a Gaussian least square fit or Gaussian robust fit. The MT growth rates were quantified from stream acquisitions of EB3 comets at a speed of 2 frames/s using kymograph analysis.

To assess the effect of StableMARK expression on MT acetylation levels, an ROI was drawn around every individual cell and the MGV of every channel was measured. Data were transferred to Excel and the MGV of every channel was background corrected. For individual cells, the ratio between (MGV acetylated tubulin) and (MGV α-tubulin) was calculated. For StableMARK-expressing cells, the ratio between (MGV Stable-MARK) and (MGV α-tubulin) was also calculated.

To assess the effect of StableMARK expression on the amount of EB1 comets, an ROI was drawn around individual cells and their surface area was measured. The amount of EB1 comets was counted using the ComDet plugin for FIJI (https://github.com/ekatrukha/ComDet). For StableMARK-expressing cells, the MGV of the StableMARK channel was measured. Data were transferred to Excel, and for every cell, the amount of EB1 comets per µm² was calculated. MGVs of the StableMARK channel were background corrected.

To assess the effect of StableMARK expression on mitochondrial spreading, mitochondria distribution was classified as (a) spread throughout the cell, (b) clustered around the nucleus, or (c) intermediate phenotype where clustering around the nucleus was observed as well some spreading through the cytoplasm. During the classification of mitochondria distribution, the observer was unaware of the transfection status of the classified cell. For StableMARK-expressing cells, an ROI was drawn around the cell and MGV of the StableMARK channel was measured. Data were transferred to Excel and MGVs of the

StableMARK channel were background corrected. The resulting values were converted to Log10 values. Cells were classified based on their Log10(MGV) value into low, medium, or high expression, with every class containing a range of values representing one third of the difference between the Min and Max Log10 value of the dataset.

The percentage of StableMARK-decorated MT-ends from which an EB3 comet was growing was quantified by visual inspection of the dual-color stream acquisition, maximum projections, and kymographs. Stream acquisitions were 200 frames long. Some StableMARK-decorated MT-ends disappeared out of the field of view or focus during that time window. StableMARK-decorated MT-ends were included in the analysis if they could be tracked for ≥50 frames (≥25 s). The percentage of acetylated MT-ends that had a dynamic end (as manifested by a stretch in the total tubulin channel extending from the acetylated MT) was quantified by visual inspection of the super-resolution data.

The behavior of freshly generated, StableMARK-decorated MT-ends after photo-ablation was classified as "stable" or "depolymerization" based on visual inspection of dual-color stream acquisitions and kymographs.

To assess the dwell time of StableMARK molecules bound to dynamic MTs, ROIs that were devoid of stable MTs and where binding-unbinding events were clearly visible were selected. StableMARK molecules were automatically detected and fitted using the FIJI-plugin DoM_Utrecht (https://github.com/ekatrukha/DoM_Utrecht). Subsequently, to estimate the time StableMARK molecules stayed bound to dynamic MTs, detected molecules were linked in consecutive frames with a maximum distance of four pixels between detected particles in consecutive frames and a maximum linking gap in frames of 1 using the FIJI-plugin DoM_Utrecht. Data were transferred to Excel and the track lengths were multiplied by 0.2 s to calculate binding times. Data were transferred to GraphPad Prism 9 to calculate cumulative frequency distributions. Subsequently, data was fitted with $y = (y_1 - y_2) * e^{(-k*x)} + y_2$ to get the dwell time $t = 1/k$.

To quantify the accumulation of StableMARK in the midzone over time, events were aligned for the onset of anaphase. For 19 subsequent frames (corresponding to 45 min), an ROI was drawn around the midzone to measure StableMARK fluorescent intensity. Data were transferred to Excel and corrected for background and ROI area. Next, fluorescent intensity values were normalized per cell by dividing all values by the maximum value. To quantify the stabilization of the forming intracellular bridge over time, U2OS WT cells stained for acetylated tubulin and α-tubulin were imaged at three different stages of bridge formation. An ROI was drawn around the midzone/developing bridge and fluorescence intensities were measured. Data were transferred to Excel and was corrected for background and ROI area. Next, the value of acetylated tubulin was divided by the value of total tubulin per individual cell.

To quantify the binding of StableMARK to expanded and compacted MT lattices in vitro, isolated GDP sections of MTs were traced in ImageJ as segmented lines to quantify the length of MT analyzed. This was compared for MT images before and after imaging StableMARK to ensure the MT did not depolymerize. Subsequently, the number of molecules along the traced

MTs was counted by eye based on the timelapse acquisition of the molecules (61 frames) and sum projections thereof to ensure no molecules were overlooked. For each field of view (8–10 per independent experiment, three independent experiments), the number of counted molecules was divided by the length of GDP MT analyzed and the concentration of motor (0.0152 nM). Statistical significance was determined using a ratio paired t-test of the means in GraphPad Prism 9. For this test, normality was assumed, but not formally tested given the sample size of $N = 3$, and Taxol- and DMSO-treated samples from the replicates were assumed to be paired.

### Online supplemental material

Fig. S1 shows further evidence that StableMARK labels the subset of stable MTs. Related to Figs. 1 and 2. Fig. S2 demonstrates that StableMARK prefers binding to expanded lattices in vitro. Related to Fig. 2. Fig. S3 data depicts live-cell imaging of the behavior of individual stable MTs. Related to Fig. 3. Fig. S4 shows further evidence that StableMARK at low levels has minimal effects on MTs and organelle transport. Related to Figs. 4 and 5. Fig. S5 gives further data on the dynamics of StableMARK-decorated MTs. Related to Fig. 6. Fig. S6 provides more data on stable MTs during cell division. Related to Fig. 8. Fig. S7 shows data that demonstrate the localization of Stable-MARK to stable MTs in different cell lines. Video 1 (related to Fig. 3) shows StableMARK and mCherry-tubulin in a U2OS cell. Video 2 (related to Fig. 3 A) shows StableMARK and mCherry-tubulin in a U2OS cell. Video 3 (related to Fig. 3 B) shows StableMARK and mCherry-tubulin in a U2OS cell. Video 4 (related to Fig. 3 C) shows StableMARK in U2OS cell. Video 5 (related to Fig. 3 E) shows StableMARK in U2OS cell. Video 6 (related to Fig. 5 C) shows cesicles labeled with mCherry-Rab6a moving over StableMARK-positive MT in a U2OS cell. Video 7 (related to Fig. 6 B) shows laser-induced severing of MTs in U2OS cell expressing StableMARK and mCherry-tubulin. Video 8 (related to Fig. 6 D) shows EB3-tdTomato comet growing from StableMARK-labeled MT in U2OS cell. Video 9 (related to Fig. 7 B) shows transient binding of StableMARK to MTs labeled with mCherry-tubulin in U2OS cell. Video 10 (related to Fig. 8 C) shows mitosis in stable U2OS Flp-In cell(s) expressing StableMARK.

### Acknowledgments

We thank Casper Hoogenraad (Genentech, Inc., South San Francisco, California, USA) for the mCherry-α-tubulin and FKBP-mCherry-Rab6A constructs, Anna Akhmanova (Utrecht University, Utrecht, Netherlands) for fruitful discussion and for providing the Swiss 3T3 cells, and for the ß-tubulin-GFP construct, Alessandro Sartori (University of Zurich, Zurich, Switzerland) for providing the U2OS Flp-In T-Rex cells and Sven van Ijzendoorn (University Medical Center Groningen, Groningen, Netherlands) for providing the Caco-2 cells. We thank Leanne de Jager (Utrecht University, Utrecht, Netherlands) for subcloning hKif5b(1-560)G234A-L-mNeongreen-L-mNeongreen into pCDNA5/FRT/TO, and we thank Eugene Katrukha (Utrecht University, Utrecht, Netherlands) for technical support and image analysis advice.

This work is supported by the Netherlands Organization for Scientific Research (NWO-Graduate program project 022.006.001 to K.I. Jansen), European Molecular Biology Organization (EMBO long-term fellowship [EMBO ALTF 407-2017] to M. Burute), and the European Research Council (ERC Consolidator Grant 819219 to L.C. Kapitein).

Author contributions: K.I. Jansen and L.C. Kapitein designed the study. K.I. Jansen created reagents, performed the fixed- and live-cell experiments, and analyzed data. M.K. Iwanski created reagents, performed all in vitro reconstitution experiments, and analyzed data. M. Burute performed additional experiments. K.I. Jansen, M.K. Iwanski, and L.C. Kapitein interpreted data and wrote the manuscript. L.C. Kapitein supervised the project.

Disclosures: The authors declare no competing interests exist.

Submitted: 17 June 2021

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

# Supplemental material

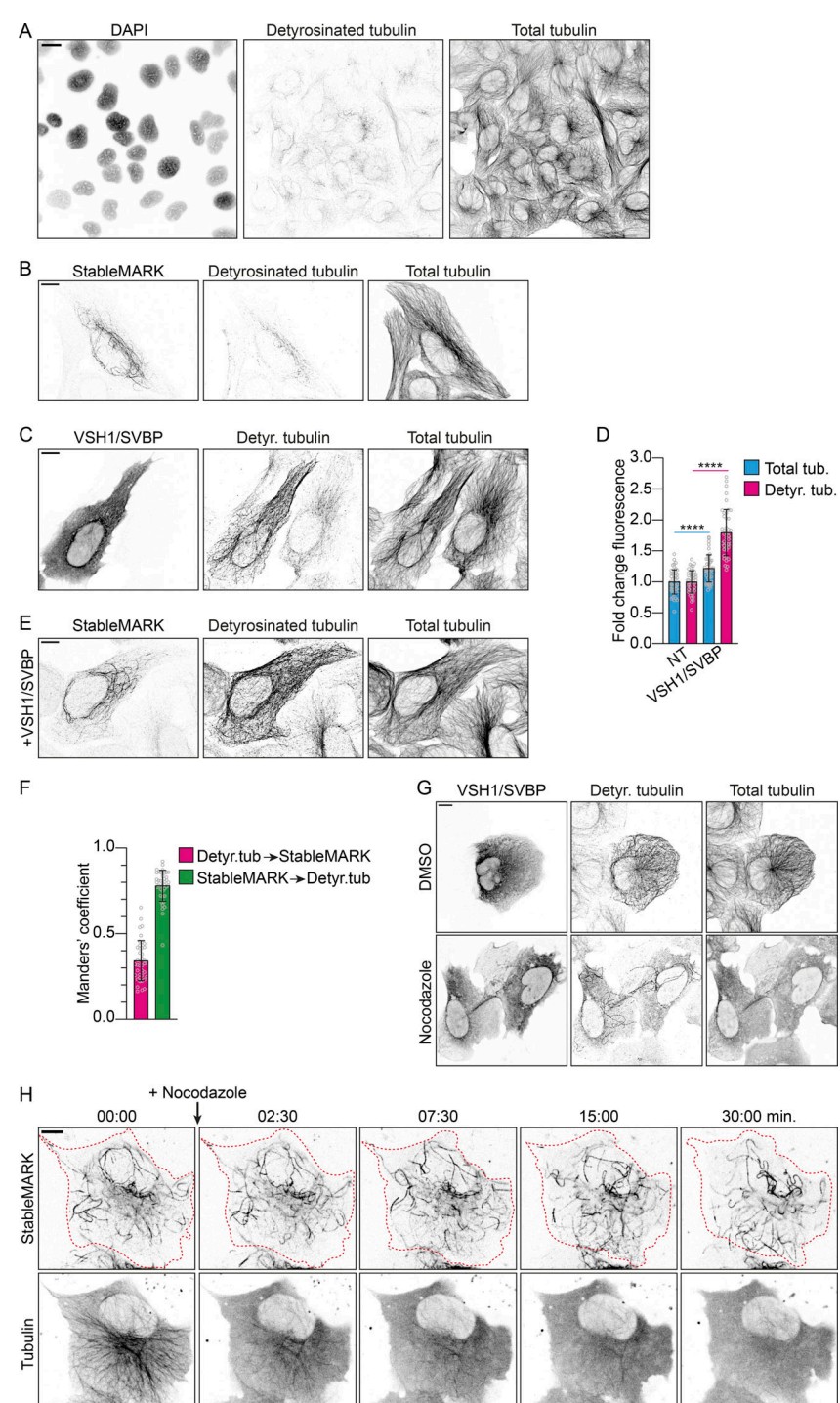

Figure S1. **StableMARK labels the subset of stable MTs.** Related to Figs. 1 and 2. **(A)** Fluorescence images of U2OS cells stained for DAPI and immunolabeled for detyrosinated tubulin and α-tubulin are shown in inverted contrast. **(B)** Fluorescence images of U2OS cell expressing StableMARK and immunolabeled for detyrosinated tubulin and α-tubulin. **(C)** U2OS cell expressing VSH1-GFP and SVBP-FLAG (left cell) and a non-transfected cells (right cell) immunolabeled for GFP, detyrosinated tubulin, and α-tubulin. **(D)** Bar graph showing the fold change in mean fluorescence intensity of total tubulin and detyrosinated tubulin of VSH1/SVBP expressing cells normalized to non-transfected control cells. Graph represents mean ± SD as well as individual values (gray circles) for 43–46 cells per condition from three independent experiments. Unpaired *T* test, ****P ≤ 0.0001. **(E)** Fluorescence images of U2OS cell expressing VSH1-FLAG, SBVP-FLAG, and StableMARK immunolabeled for detyrosinated tubulin and α-tubulin. **(F)** Bar graph showing thresholded Manders' coefficients measured from 12.5 × 12.5 μm ROIs per cell for the condition described in E. Pink bar represents the colocalization coefficient for detyrosinated MTs to StableMARK-decorated MTs. Green bar represents the colocalization coefficient for StableMARK-decorated MTs to detyrosinated MTs. Graph represents mean ± SD as well as individual values (gray circles) for 41 cells per condition from three independent experiments. **(G)** Fluorescence images of U2OS cells expressing VSH1-GFP and SBVP-FLAG treated for 1 h with 0.1% DMSO or 10 μM nocodazole and immunolabelled for detyrosinated tubulin and α-tubulin. **(H)** Stills from live-cell imaging of StableMARK and mCherry-tubulin in U2OS cell upon addition of 10 μM nocodazole. Time: min:s. Scale bars, 20 μm (A), 10 μm (B, C, E, G, and H).

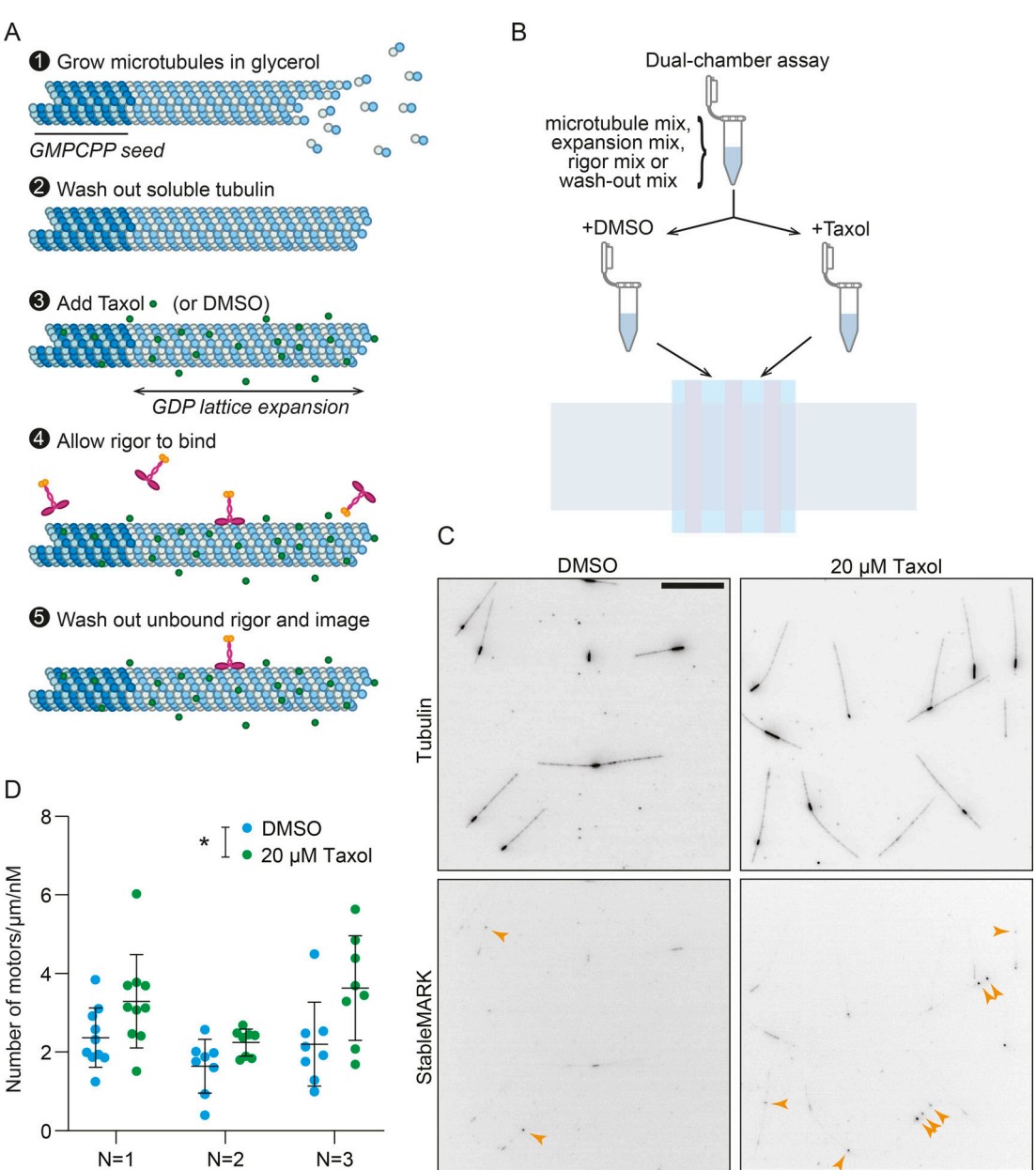

Figure S2. **StableMARK shows a preference for expanded lattices in vitro.** Related to Fig. 2. **(A)** Schematic showing the assay setup. All steps were done in high glycerol buffer. MTs were polymerized from immobilized GMPCPP-stabilized seeds for 15 min (1). After washing out soluble tubulin (2), 20 µM Taxol was added and allowed to incubate to expand the MT lattice (3), or DMSO was added as a control. Subsequently, 15.2 pM StableMARK was added and allowed to bind for 90 seconds (4), before washing out unbound rigor and imaging (5). **(B)** Assays were performed paired, with buffers for all steps prepared together and then split into two equal parts for the addition of Taxol or DMSO. These buffers were introduced into two flow cells on the same coverslip, treated, and imaged concomitantly. **(C)** Representative images showing the MTs (top; dark: GMPCPP seed, light: GDP lattice) and StableMARK (bottom; sum projection). Arrowheads indicate rigor bound to GDP segments. **(D)** Quantification thereof with the number of bound molecules counted per µm of GDP lattice traced per nM of StableMARK added. For each independent experiment, there is an increase in the average density of motors on the GDP lattice in the Taxol-treated condition, suggesting that StableMARK shows a preference for expanded lattices. $n$ = 8–10 fields of view for $N$ = 3 independent experiments. The mean ± SD is shown for each experiment. *P < 0.05 (ratio paired $t$ test of the means; normality assumed, not formally tested). Note that the off-rate of StableMARK is ∼0 in all conditions tested in vitro and this could limit the difference observed. Scale bar, 10 µm (C).

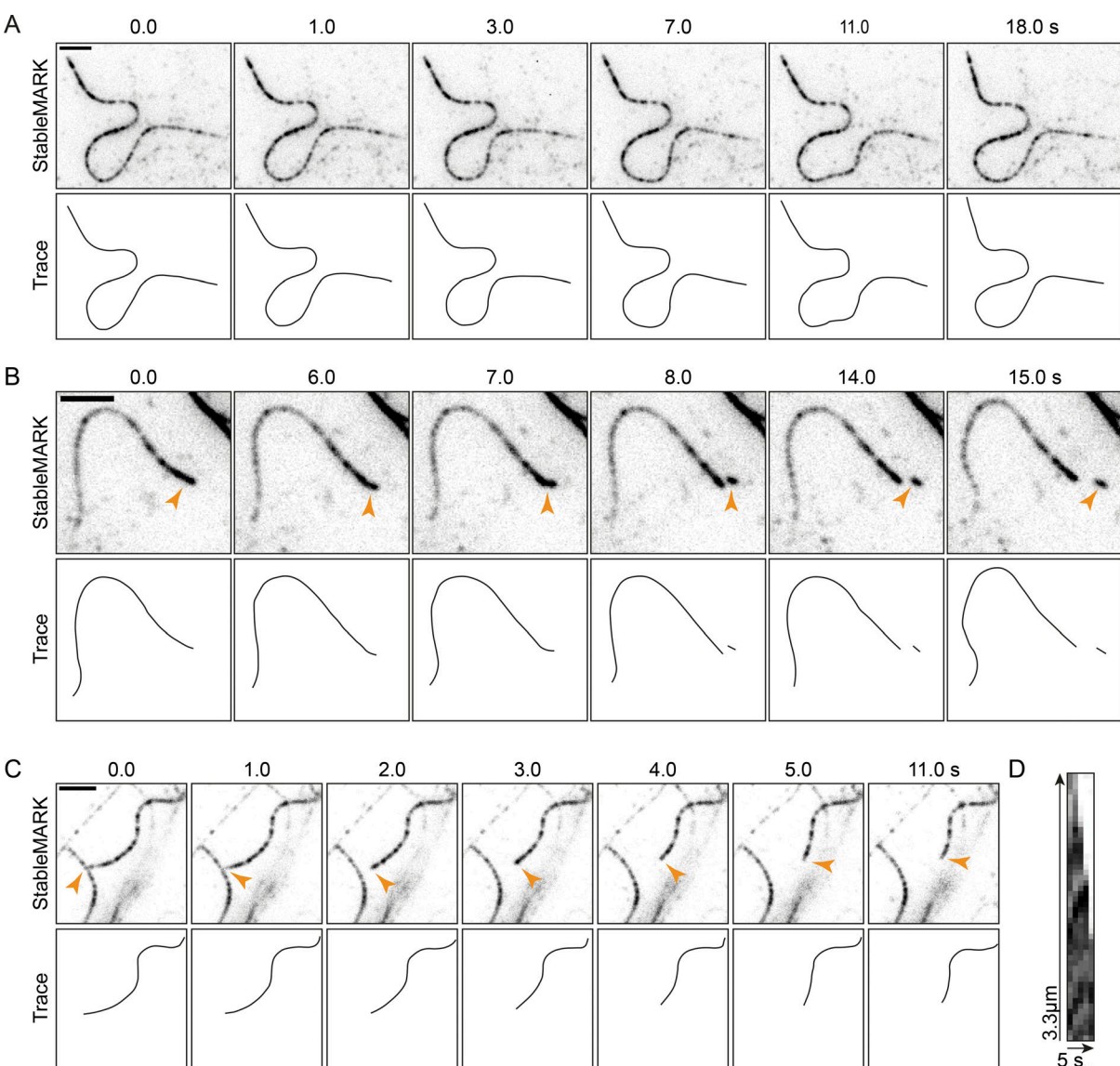

Figure S3. **Live-cell imaging of the behavior of individual stable MTs.** Related to Fig. 3. **(A–C)** Stills and schematic representations from live-cell imaging of StableMARK in U2OS cells depicting MT deformation, MT breakage (indicated by orange arrowhead), and partial MT depolymerization (indicated by orange arrowhead), respectively. **(D)** Kymograph of the partial depolymerization event is shown in C. Scale bars, 2.5 µm. Time, min:s.

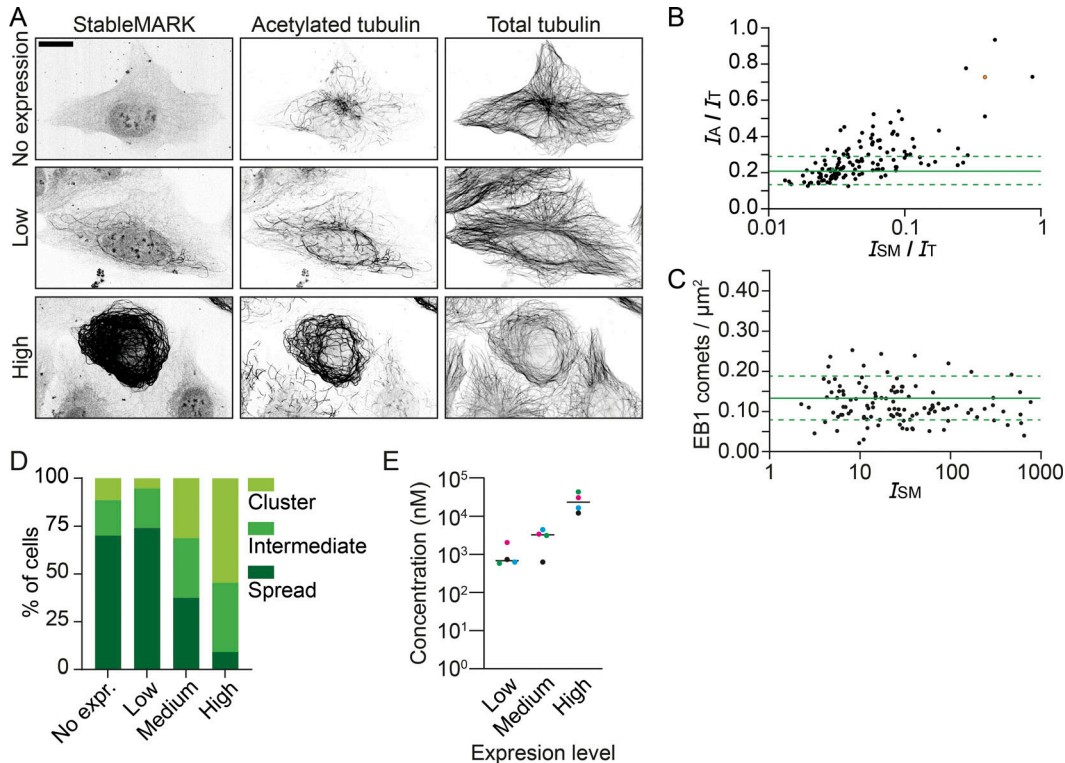

Figure S4. **StableMARK at low levels has minimal effects on MTs and organelle transport.** Related to Figs. 4 and 5. **(A–C)** Independent replicates of the experiments shown in Fig. 4, A–D. **(A)** Fluorescence images of U2OS cells stained for acetylated tubulin and α-tubulin at different levels of StableMARK expression as analyzed in B. **(B)** Quantification showing the intensity ratio of acetylated tubulin ($I_A$) over total tubulin ($I_T$), plotted against the intensity ratio of StableMARK ($I_{SM}$) over total tubulin for individual StableMARK-expressing cells (each dot represents a single cell, $n = 131$, $N = 1$; see Fig. 4 for replicate). Solid green line + dashed lines indicate mean ± SD of the intensity ratio of acetylated tubulin over total tubulin for non-expressing cells ($n = 120$). Green data points represent low expression example from A. Orange datapoint represents high expression example from A. **(C)** Quantification showing the number of EB1 comets/μm² for cells with different StableMARK intensities (each dot represents a single cell, $n = 115$, $N = 1$; see Fig. 4 for replicate). Solid green line + dashed lines indicated mean ± SD of amount of EB1 comets/μm² for non-StableMARK-expressing cells ($n = 99$). **(D)** Graph depicting the intracellular concentration of StableMARK measured using FCS in cells classified by eye as low, medium, or high expressing. Median (line) and averages per independent experiment (colored dots) are shown, representing data from 16 to 20 cells per condition from four independent experiments. **(E)** Classification of mitochondria distribution at different levels of StableMARK expression (no expression, $n = 124$ cells; StableMARK expressing, $n = 101$ cells [low, $n = 58$ cells; medium, $n = 32$ cells; high, $n = 11$ cells]). Scale bar, 20 μm (A).

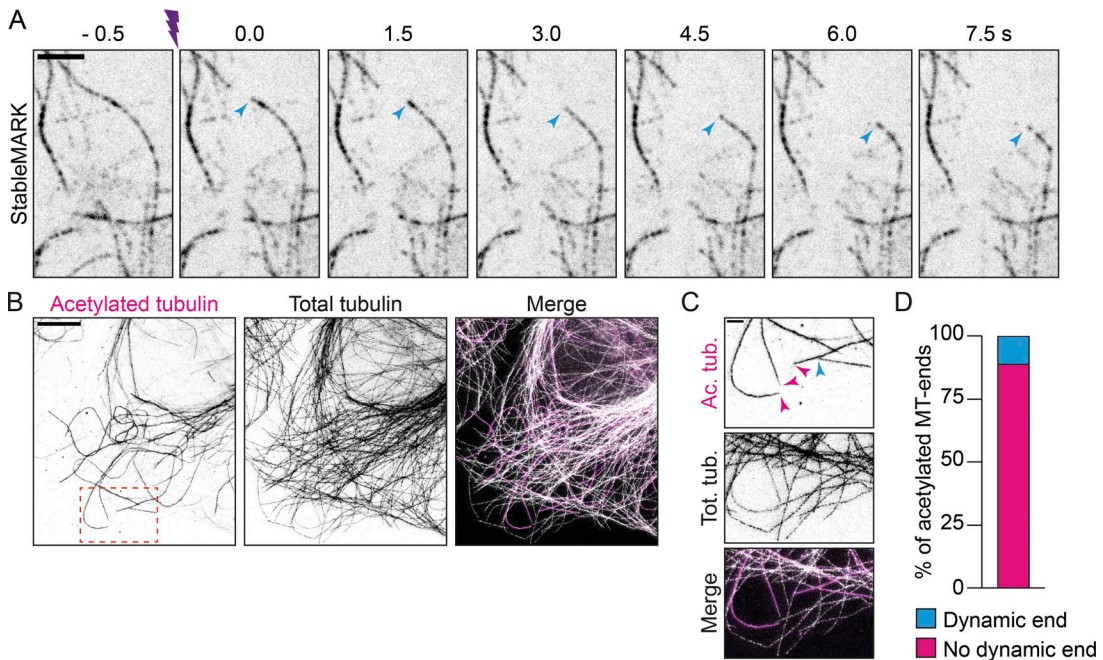

Figure S5. **Dynamics of StableMARK-decorated MTs.** Related to Fig. 6. **(A)** Stills from live-cell imaging of U2OS cell expressing StableMARK (shown) and mCherry-tubulin. Purple marker indicates photo-ablation. Cyan arrowheads point toward freshly generated, StableMARK-decorated MT-end that depolymerizes upon cutting. **(B)** Fluorescence STED images of U2OS cell immunolabeled for acetylated tubulin and α-tubulin. **(C)** Zoom of region indicated with red dashed boxed in B. Magenta arrowheads indicate acetylated MT-tips that do not have dynamic ends. Cyan arrowhead points toward an acetylated MT-tip that does have a dynamic end, as manifested by a total tubulin stretch extending from the acetylated MT-tip. **(D)** Stacked bar graph showing the percentage of acetylated MT-ends with/without a dynamic end. 243 acetylated MT-tips from 42 cells from one independent experiment were included in this analysis. Scale bar, 5 µm (A and B), 1 µm (C). Time, s:ms.

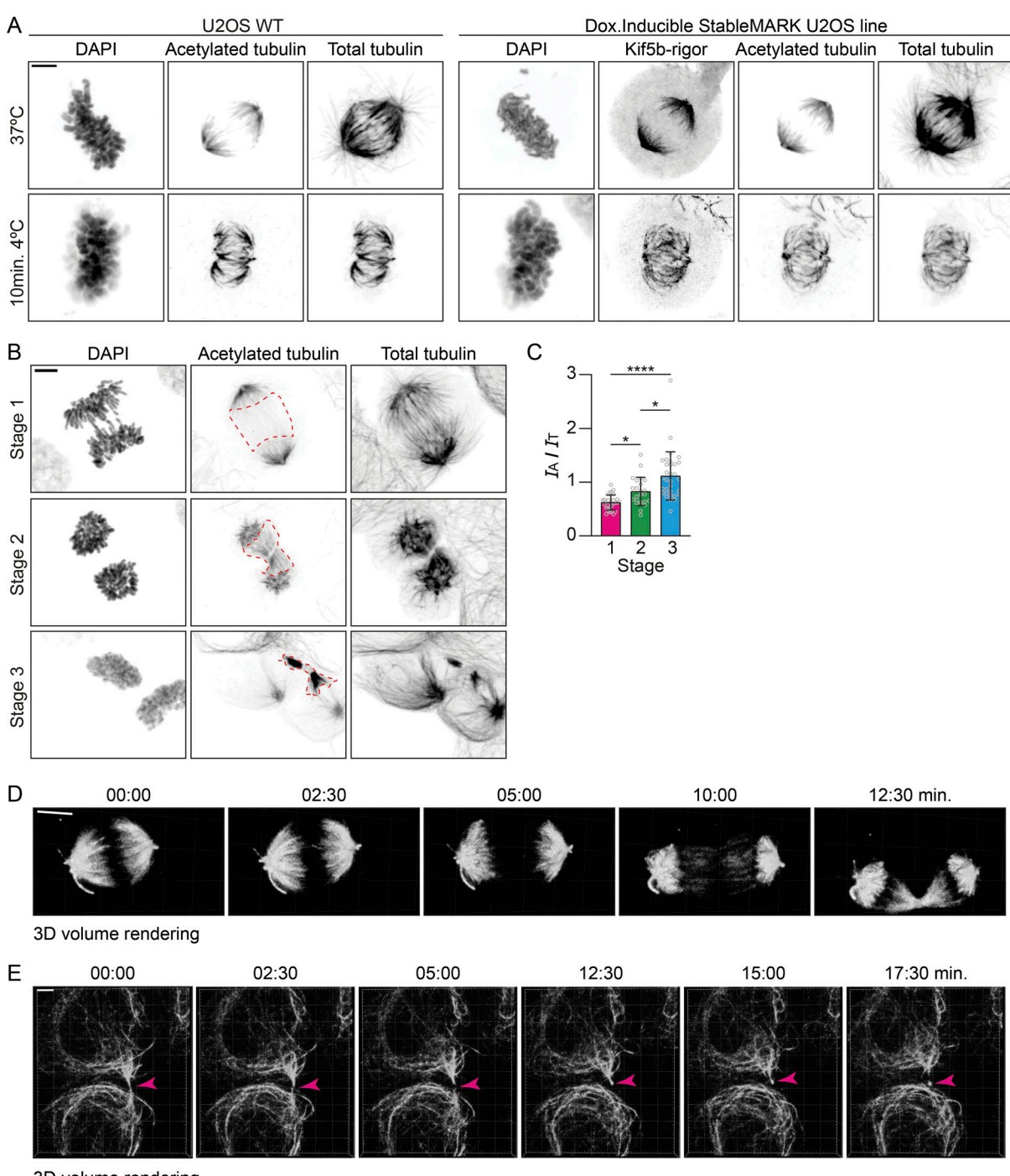

Figure S6. **Stable MTs during cell division.** Related to Fig. 8. **(A)** Maximum intensity projection images of U2OS WT and U2OS;StableMARK cells treated with doxycycline to induced expression of StableMARK and stained for DAPI, acetylated tubulin, and total tubulin during metaphase at 37°C or after 10 min at 4°C. **(B)** Maximum intensity projection images of U2OS WT cells stained for DAPI, acetylated tubulin, and total tubulin at different stages of intracellular bridge formation. Red dashed boxes indicate the regions that are analyzed for the data plotted in C. **(C)** Quantification showing the intensity ratio of acetylated tubulin ($I_A$) over total tubulin ($I_T$) in the regions indicated in B. Graph represents mean ± SD as well as individual values (gray circles). 23–32 regions per stage from three independent experiments were included in the analysis. Kruskal–Wallis test with Dunn's multiple comparisons test. *P ≤ 0.05, ****P ≤ 0.0001. **(D)** 3D volumetric rendering of first five frames shown in Fig. 8 C. **(E)** Stills from live-cell imaging of U2OS;StableMARK cells treated with doxycycline to induce expression. For every time point, a Z-stack of 25 slices with 0.75 µm step size is shown as a 3D volumetric render. Magenta arrowheads indicate abscission and release of the midbody remnant. Scale bars, 5 µm (A, B, and D), 10 µm (C), 8 µm (E). Time, min:s.

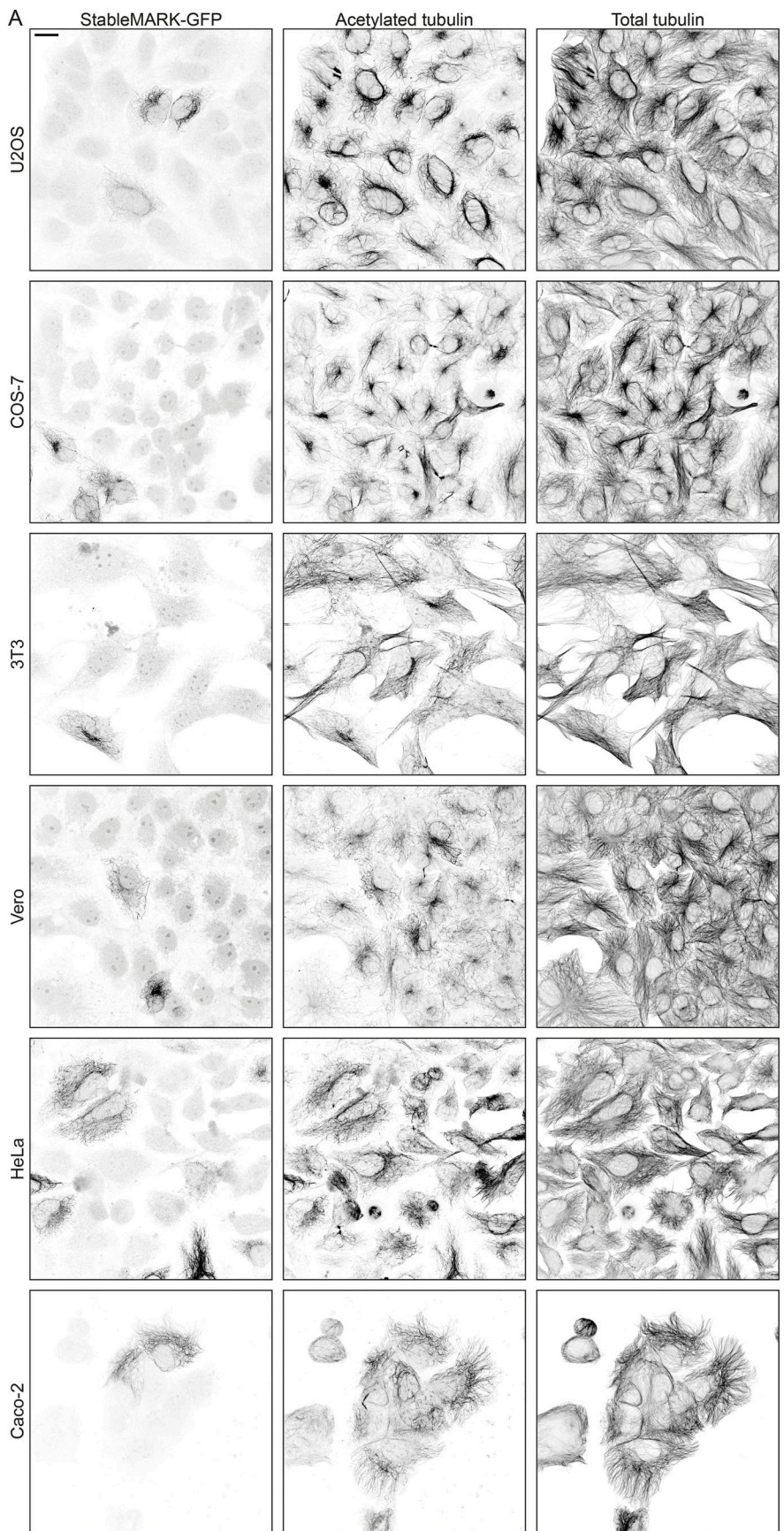

Figure S7.   **Localization of StableMARK to stable MTs in different cell lines. (A)** StableMARK (GPF version) in U2OS, COS-7, 3T3, Vero, Hela, and Caco-2 cells immunolabeled for GFP, acetylated tubulin, and α-tubulin. Scale bar, 20 μm.

Video 1.   **StableMARK and mCherry-tubulin in U2OS cell.** This video corresponds to Fig. 3. Total time: 3 min. Acquired using spinning-disk confocal microscopy with 1.5 s interval between frames. 30× sped up.

Video 2.   **StableMARK and mCherry-tubulin in a U2OS cell.** This video corresponds to Fig. 3 A. Total time: 1 min and 10 s. Acquired using TIRF microscopy with 1 s interval between frames. 20× sped up.

Video 3.   **StableMARK and mCherry-tubulin in a U2OS cell.** This video corresponds to Fig. 3 B. Total time: 30 min. Acquired using TIRF microscopy with 30 s interval between frames. 225× sped up.

Video 4.   **Movement of StableMARK-positive MT in U2OS cell.** This video corresponds to Fig. 3 C. Total time: 45 s. Acquired using TIRF microscopy with 1 s interval between frames. 10× sped up.

Video 5.   **Movement of StableMARK-positive MT in U2OS cell.** This video corresponds to Fig. 3 E. Total time: 33 s. Acquired using TIRF microscopy with 1 s interval between frames. 10× sped up.

Video 6.   **Vesicles labeled with mCherry-Rab6a moving over StableMARK-positive MT in a U2OS cell.** This video corresponds to Fig. 5 C. Total time: 9.7 s. Stream acquisition acquired using TIRF microscopy at a speed of 10 frames/s. 2× sped up.

Video 7.   **Laser-induced severing of MTs in U2OS cell expressing StableMARK and mCherry-tubulin.** This video corresponds to Fig. 6 B. Total time: 8.5 s. Stream acquired using spinning-disk confocal microscopy at 2 frame/s. 5× sped up.

Video 8.   **EB3-tdTomato comet growing from StableMARK-labeled MT in U2OS cell.** Related to Fig. 6 D. Total time: 8.5 s. Stream acquisition acquired using TIRF microscopy at a speed of 2 frames/s. 5× sped up.

Video 9.   **Transient binding of StableMARK to MTs labeled with mCherry-tubulin in U2OS cell.** Related to Fig. 7 B. Total time: 4.8 s. Stream acquisition acquired using TIRF microscopy at a speed of 5 frames/s. 6× sped up.

Video 10.   **Mitosis in stable U2OS Flp-In cell(s) expressing StableMARK.** Related to Fig. 8 C. Total time: 1 h and 55 min. Stream acquired using spinning-disk confocal microscopy with imaging interval of 2.5 min. 750× sped up.

