## [Peer Review File · The Journal of Cell Biology]

A live-cell marker to visualize the dynamics of stable microtubules throughout the cell cycle

Klara Jansen, Malina Iwanski, Mithila Burute, and Lukas Kapitein

Corresponding Author(s): Lukas Kapitein, Utrecht University

Review Timeline:

Submission Date:	2021-06-17
Editorial Decision:	2021-07-20
Revision Received:	2022-01-08
Editorial Decision:	2022-02-09
Revision Received:	2022-06-27
Editorial Decision:	2022-10-05
Revision Received:	2023-01-26
Editorial Decision:	2023-01-27
Revision Received:	2023-02-07

Monitoring Editor: Rebecca Heald

Scientific Editor: Dan Simon

Transaction Report:

DOI: <https://doi.org/10.1083/jcb.202106105>

July 20, 2021

Re: JCB manuscript #202106105

Prof. Lukas Kapitein
Utrecht University
Padualaan 8
Utrecht 3533 CH
Netherlands

Dear Prof. Kapitein,

Thank you for submitting your Tools manuscript entitled "A live-cell marker to visualize the dynamics of stable microtubules" to Journal of Cell Biology. Please accept our apologies for the delay in the processing of your manuscript, the journal has been understaffed for a long time leading to delays in processing. As part of our normal reviewing procedure, your paper has been evaluated by at least two editors and an editorial statement is provided below. You will see that, in the consensus opinion of our editors, although we are interested in the concepts presented in this study, the manuscript is too preliminary for review. We have thus decided not to subject your manuscript to an external review process. We would be willing to consider a revised manuscript containing data addressing the detailed editorial comments below, assuming the novelty of the findings has not been compromised in the interim.

Because Journal of Cell Biology addresses a wide and diverse audience of cell biologists, we must give priority to manuscripts that provide a substantial advance of broad appeal to the cell biology community, even though many others also present interesting and important advances for researchers in a particular field.

I am sorry that our answer on this occasion is not more positive, and I hope that this outcome will not dissuade you from submitting other manuscripts to us in the future.

Thank you for your interest in Journal of Cell Biology.

With kind regards,

Jodi Nunnari
Editor-in-Chief
Journal of Cell Biology

Editorial Statement:

In this study the authors describe the development and initial characterization of a probe for live imaging of stable microtubules based on a rigor mutant of kinesin-1. Using this sensor they image stable microtubules in U2OS cells and report that while the microtubules themselves are stable they display dynamic behaviors such as undulation, looping, and sliding. The manuscript is well written and the work is of high quality. Similar probes, such as those based on the N-terminal domain of ensconsin, that image stable microtubules in live cells have been reported before, and these are not discussed here nor is there a comparison of the kinesin-1 sensor to these probes. Prior studies have also revealed that stable microtubules exhibit looping and sliding behaviors and the present study also does not provide a thorough and quantitative exploration of the dynamic behavior of stable microtubules using the new kinesin-1 based probe. Thus, regrettably, in its present form this work does not establish that this new sensor allows for interrogation of cell biological problems in ways previously impossible nor provide novel cell biological insights as a proof of principle, which are required for consideration as a JCB Tools paper. We would, however, be interested in seeing a more complete study in which these issues were addressed, if the authors wish to do so.

February 9, 2022

Re: JCB manuscript #202106105R-A

Prof. Lukas Kapitein
Utrecht University
Padualaan 8
Utrecht 3533 CH
Netherlands

Dear Lukas,

Thank you for submitting your manuscript entitled "A live-cell marker to visualize the dynamics of stable microtubules throughout the cell cycle" to JCB. We have now heard back from three reviewers, and as you will see, they agree that a marker specific for stable microtubules would be a very useful tool. However, a number of concerns are raised, in particular important issues to clarify include what exactly your marker is detecting and whether its binding affects microtubule dynamics. I refer you to the comments for details, but the scope of revision that would be necessary is substantial. Therefore, we are rejecting the current version of your manuscript, but would be willing to consider a new version that fully addresses the points raised by the reviewers.

Although your manuscript is intriguing, I feel that the points raised by the reviewers are more substantial than can be addressed in a typical revision period. If you wish to expedite publication of the current data, it may be best to pursue publication at another journal. Our journal office can transfer your reviewer comments to another journal upon request.

Given interest in the topic, I would be open to resubmission to JCB of a significantly revised and extended manuscript that fully addresses all of the reviewers' concerns and is subject to further peer-review. If you would like to resubmit this work to JCB, please contact the journal office with a revision plan to discuss an appeal of this decision. Please note that priority and novelty would be reassessed at resubmission.

Regardless of how you choose to proceed, we hope that the comments below will prove constructive as your work progresses. We would be happy to discuss the reviewer comments further once you've had a chance to consider the points raised in this letter. You can contact the journal office with any questions, cellbio@rockefeller.edu or call (212) 327-8588.

I thank you again for your interest in publishing your work in JCB.

Best regards,
Rebecca

Rebecca Heald, PhD
Monitoring Editor
Journal of Cell Biology

Dan Simon, PhD
Scientific Editor
Journal of Cell Biology

Reviewer #1 (Comments to the Authors (Required)):

Cells contain a mixture of stable microtubules with slow turnover and dynamic microtubules that turn over rapidly. These different microtubule populations frequently are post-translationally modified, mainly acetylation, deetyrosination and glutamylation. However, while antibodies for these modifications have been in use for a long time and have served as proxy markers for stable microtubules, there are no live markers for stable microtubules. The lack of such markers has been an impediment into deeper cell biological investigations. The authors explore in this study the use of a rigor form of Kinesin-1 to preferentially label and image stable microtubules in live cells. They show they can image these stable microtubules over long times and observe their fate during cell division. While the idea is interesting and the imaging is well-done (not surprising from the Kapitein group), I have strong reservations about the rigor and significance of this work as it currently stands. This field has been muddled by a lot of phenomenology and a lack of mechanism. The work described here, while potentially interesting and useful, will only further confuse the field in its current state because it is not clear (i) what this rigor kinesin construct recognizes and (ii) what its effect is on microtubule dynamics. These points would have to be addressed in order for this marker to be

reliable. In addition, given that this is a resource paper, the authors should show that their observations hold in at least one other cell line. There is also a lack of scholarship, with both old and new work not being cited. The authors fail to cite and discuss relevant studies that have direct impact on the interpretation of their experiments. My specific comments are below. I hope the authors will consider them.

1. In Figure 1 the authors focus a lot of their attention on acetylation. They show that this marker is marking stable microtubules but not necessarily the acetylated ones. They compare the colocalization of Kin-1 and acetylated MTs on taxol-stabilized and hyper-acetylated microtubules. In the former their marker binds all MTs whereas without the taxol stabilization it binds a subset of MTs. Could this be due to their Kinesin rigor construct binding differently to taxol and non taxol stabilized MTs? Their interpretation is supported by live cell nocodazole treatment, but it would be reassuring to see a quantification of the amount of stable MTs before and after washout with and without their marker. A small point but the dashed lines in fig 1b and S1b showing where the line profiles in s1c were taken from are black and very difficult to see, perhaps make them more obvious to the eye?
2. In Figure 2, the high curvature of stabilized microtubules has been observed by other groups before. Citation of this earlier work is encouraged (for example, see Friedman et al JCB 2010).
3. One major issue with the argument that the authors are trying to make is that it is not clear whether their rigor kinesin is actually stabilizing microtubules. Microtubule dynamics parameters need to be determined in control cells and cells expressing their rigor kinesin. Just reporting EB1 comet numbers is not enough -what are the microtubule dynamic parameters? These are highly feasible experiments, especially for this lab. I am sure that at higher concentration levels the rigor kinesin will affect microtubule dynamics. What is the expression range for which this is a passive marker?
4. In vitro work showed that de-tyrosination affects the binding of kinesin-1 more so than acetylation (Kaul et al 2014 - also not cited) yet the authors seem to have focused on acetylation. Is it possible that this is a marker for de-tyrosinated microtubules? They show that at low levels the amount of acetylated tubulin is about the same with and without the rigor mutant. (Figure 3A). It would be reassuring to see the same for de-tyr tubulin, and to see a quantification for this. If the marker responds more to tyrosination than acetylation they also may be missing a fraction of the stabilized microtubules.
5. The Rab6A transport data needs to be quantified to see whether transport speeds are changed by the presence of more possible roadblocks (ie rigor kinesin) on the MTs.(Figure 3G). The transport kinetics of other cargos should also be analyzed that use other motors than the two used for Rab6A transport (for example, a connection between lysosome transport and acetylated MTs has been previously established). The serum starvation experiments are nice, but all they show is that the marked MTs can be depolymerized by this assay (already known). In general, the comment that "our data demonstrate that in the subpopulation of low rigor-expressing cells, artifacts of the MT cytoskeleton are minimal" is not backed up by sufficient data. It might very well be the case, but they need to do more to prove this so that this can be a marker that can be used with confidence in the field.
6. Their investigation of the mechanism of stabilization is technically well-performed but I have reservations with their interpretation. Upon ablation they see ~50% of the MTs depolymerize from one end which they interpret as an aging effect, where stabilization starts at the +tip and is followed by stabilization along the lattice. There have been several papers on motor induced damage and long-range lattice effects, including the stabilization of the lattice by even substoichiometric amounts of kinesin (Andreu-Carbo et al, 2022, Triclin et al 2021; Peet et al 2018). These need to be discussed here as they point to the possible perturbing role that their marker can actually have. If these microtubules also are healed more, they will also be more stable when ablated because of the incorporation of GTP tubulin (Aumeier et al. 2016; Vemu et al 2018).
7. When discussing laser ablation experiments and the asymmetry in depolymerization between the plus and minus ends, the authors strangely cite a paper from 2014. This is textbook work from Salmon and colleagues for example and should be cited.
8. The authors make the interesting observation that the rigor kinesin coated MTs do not have dynamic (EB3) tips. Based on this they speculate that therefore there is a "special tip structure" on stabilized microtubules that prevents both growth and shrinkage. Could it be that their rigor mutant interferes with normal EB3 binding?
9. What is the effect of this rigor mutant on microtubule dynamics in vitro? These are highly feasible experiments that should be performed. And does it affect EB binding? Mechanistic data on what the effect of this mutant is on microtubule dynamics and binding of some ubiquitous effectors is needed.
10. The rigor kinesin seems to be decorating only a portion of the stable MT populations in the spindle - why? This question brings me back to my biggest reservation for this work: the authors do not know what features this kinesin mutant recognizes. So, yes it might mark some stable microtubules, but maybe that will change with circumstance (if it is not stability that it marks ,but something else).

Reviewer #2 (Comments to the Authors (Required)):

This paper describes the use of a rigor mutant of kinesin-1 to identify stable microtubules (MTs) in living cells. The authors show convincing data that a 2X-mNeon-tagged version of the rigor kinesin-1 marks stable, acetylated MTs in cells. They then use the rigor kinesin to follow stable MT dynamics in interphase and mitotic cells and show a number of behaviors of these MTs, many of which have been reported previously. Overall, the development of a live cell probe for stable MTs is an unmet need for cell biologists and has the potential to reveal novel behaviors of these MTs. However, the paper has some major issues (see below) that need to be addressed before the rigor kinesin-1 is characterized well enough to be a reliable live cell marker for stable MTs.

Key issues:

1. The authors need to establish the basis for recognition of the stable MTs by the rigor kinesin-2 sensor. The rigor mutant labels stable, acetylated MTs, yet they also show by manipulating acetylation that it is not the basis for the detection of the stable MTs. Without understanding the basis for the recognition of the stable MTs by the rigor mutant, MTs detected by the rigor kinesin are only operationally defined as stable. This feature of the paper has great potential to create confusion/controversy when drawing conclusions from results with the rigor kinesin obtained in different systems and by different investigators. There is abundant evidence that kinesin-1 exhibits elevated binding to MTs with detyrosinated tubulin (e.g., Liao G and Gundersen GG, JCB, 1996; Dunn S et al, JCS 2008; Cai D et al, PLOS Biol, 2009; Sirajuddin M. et al, NCB, 2014), and the rigor kinesin should be tested against this posttranslational modification.

2. The relationship between the level of rigor kinesin-1 expression and artifacts caused by its expression are incompletely documented. The authors conclude (and I am not sure I agree, for example EB1 comets and hence dynamic MTs are clearly affected in Fig. 3C), that at "low" expression there is little effect on MTs, whereas at "high" expression, stable MTs increase, dynamic MTs are reduced, and overall MT distribution is disrupted. "Low" and "high" expression can mean different things to different people. Consequently, for the method to be reproducible and widely useful, it is critical that the authors define the level of rigor kinesin-1 expression that begins to show the artifacts they observe. For example, a comparison of the level of the rigor kinesin-1 to the endogenous level of kinesin-1 (or tubulin) would be helpful. Given the cell-cell variability with transient expression, it might be easier to make this comparison with stable cell lines with low and high expression.

3. There is very little new about stable MTs that is reported with the rigor kinesin leading me to wonder whether it will be all that useful. Part of this reflects the fact that the authors repeat previous findings that they are apparently unaware of. For example, previous studies have shown that stable MTs are curly (Piperno G et al, JCB, 1987 and others), do not grow for extended periods (Schulze E and Kirschner M, JCB, 1986; Webster D et al., PNAS, 1987; Infante AS et al, JCS, 2000; Palazzo AF et al, NCB, 2001), and depolymerize slowly when severed (Walker RA et al, JCB, 1989; Infante AS et al, JCS, 2000). The authors also seem unaware of all the molecular studies implicating numerous factors (Rho, CLASPs, ELKs, mDia, INFD2) in stable MT formation. These studies should be acknowledged and additional studies should be considered to show the utility of the rigor kinesin.

Additional comments:

4. The experiments showing that acetylated tubulin accumulates in MTs resistant to nocodazole and increases in taxol treated MTs shown in Fig 1F-G are not new or news. The data should be moved to the supplement.

5. The description of the behaviors of rigor kinesin decorated MTs are imprecise and in some cases over interpreted. For example, it is unclear how the authors draw the conclusion that the MT shown in Fig. 2C is sliding as it could also be treadmilling or trapped in actin flow. The MT in Suppl Fig2C might be moving rather than depolymerizing. And, I do not see active looping in fig. 2E as described.

6. I observe a substantial change in the overall distribution of MTs cause by "high" rigor kinesin expression in Fig. 3A. This should be mentioned along with the other alterations cause by expression of the rigor kinesin.

7. The comparison in Fig3B between rigor kinesin expression and acetylated tubulin levels needs to include the acetylation levels in non-expressing cells. Otherwise, their claim that expression of the rigor kinesin does not affect acetylated tubulin levels is not valid. Additionally, normalizing the rigor kinesin levels to the total tubulin levels makes no sense to me - the graph should simply show rigor kinesin expression.

8. The authors should plot EB1 comets per cell rather than EB1 comets/um to capture the clear reduction in the numbers of comets in both "low" and "high" expressing cells shown in Fig. 3C.

9. The data that Rab6A transport is unaffected needs quantification beyond the anecdotal example shown in Fig. 3G-I.

10. Many have noted that the turnover of spindle MTs (excluding mid-zone MTs, which are well known to be more stable), is an order of magnitude faster than interphase MTs. The rigor kinesin seems to decorate a subset of spindle MTs raising the question whether it is really detecting stable MTs in all cases. This result also illustrates the uncertainty with what the rigor kinesin is actually detecting.

Reviewer #3 (Comments to the Authors (Required)):

Jansen et al. describe a new live-cell sensor that specifically detects stable microtubules. This is an important advance for cell biology, as so far, labelling of distinct microtubule species in cells was restricted to the use of antibodies, i.e. fixed cells. Being able to detect specific microtubule subtypes in living cells is a huge advance that will help the community to substantially advance the understanding of roles of microtubule specialization in living cells, and to image how those microtubules behave. The authors already provide an exciting glimpse on those opportunities by showing how they new live-cell marker allows them to follow the fate of stable microtubules throughout cell division.

The presence of stable microtubules within a dynamic microtubule network as it is found in undifferentiated interphase cells has

for many decades intrigued the scientific community. Because these microtubules are the only ones in cells to accumulate posttranslational modifications such as acetylation and detyrosination, it has for long been disputed whether these modifications are the cause or the consequence of stabilization of those microtubules. Recent work showing that acetylation decreases the flexural rigidity of microtubules, thereby making them more resilient to repeated mechanical stress suggested that acetylation could cause at least longevity of microtubules in cells. However, whether the observed stable microtubules in cells are solely stable because of their posttranslational modifications has remained an open question.

The current work now provides an innovative approach to study stable microtubules in cells. The authors found that a rigor mutant of the motor protein Kif5b, when expressed at low levels in cells, specifically binds to stable microtubules, which are typically acetylated (tubulin acetylation at K40) in cells. To test whether the species of microtubules that their sensor detects could simply be the acetylated microtubules, the authors performed a series of experiments with drugs that either stabilize microtubules, or prevent their polymerization, or boost tubulin acetylation. Strikingly, they find that boosting acetylation does not increase microtubule stability in cells, and accordingly their sensor does not label more microtubules in cells with boosted acetylation. However, it does so in cells treated with the microtubule stabilizer taxol, indicating that the nature of stable microtubules - detected with the new sensor - is not restricted to their posttranslational acetylation.

After showing that their new sensor can be used as a live-cell marker for stable microtubules, the authors demonstrate by different complementary experiments that the binding of (low doses) of their new sensor does not perturb overall microtubule behavior in cells, including posttranslational modifications, microtubule dynamics, motility of kinesin motors, or intracellular architecture. They go on demonstrating that the microtubules that are specifically labelled with their sensor are indeed stable: they demonstrate that they do not depolymerize after laser cutting, which typically would happen for dynamic microtubules. They further demonstrate that the new sensor is not exclusively binding stable microtubules, but binds other microtubules in cells much more transiently, suggesting a mechanism that increases its binding affinity specifically to the stable microtubules.

Finally, the authors demonstrate how the new sensor can be used to monitor the fate and dynamics of stable microtubules in dividing cells. These impressive movies illustrate the potential of their new tool to obtain unprecedented insight into the behavior of specific microtubule subtypes in cells.

The manuscript is carefully written, figures are of high quality and most experiments have been performed thoroughly with appropriate controls. Nonetheless, several points need to be address before the paper can be considered for publication.

Major points:

1) In Fig. 3A the authors show that low expression levels of their sensor do not increase tubulin acetylation in cells. However, as a control example, they show a cell with highly bundled microtubules around the nucleus (row 1), which is not the case of the cell expressing the sensor (row 2). One could speculate that the acetylation levels in the control are higher due to the microtubule bundling, which is absent from the cell in row 2 - and thus conclude that low expression levels of the sensor lead to levels of acetylation that can only be attained in control cells upon microtubule bundling. To show that this is not the case, the authors should choose cells that are more similar in terms of microtubule bundling for this figure, i.e. a control without microtubule bundles.

2) Line 175, Fig 3G-I: this is the only experiment in this figure which does not show statistical analyses of the observation - the authors show one selected movie. How reproducible are these observations?

3) In Fig. 5D, the authors show how stable microtubules accumulate in the spindle midzone over time using their new sensor. However, they do not measure the total microtubule load of the midzone, thus it is not clear whether what they measure is simply an accumulation of microtubules.

4) One of the major drawbacks of their tool is that they need to select cells with "low expression levels". What this actually means remained unclear throughout the manuscript, nor did they mention how many cells in a population of transfected cells would fulfill this criterium and could thus be used for analysis. The authors should thus show a large field of view of a typical cell culture transfected with their sensor, and label the cells in this field that they consider "low expression" and "high expression". They might also want to give numbers for this, which should be easy by re-analyzing their experiments.

Did the authors consider the use of lentivirus to transduce cells with their sensor? This would most likely allow a better tuning of the expression levels.

Minor points:

1) The authors have developed a great tool that will certainly be used a lot in the future. However, the name they have chosen for their binder, rigor-2xmNeongreen, is hard to memorize and even hard to pronounce. They should think of a more simple, intuitive name. Also, they should use the same name throughout the manuscript - currently they use different nomenclatures in the figures and the text. Along the same lines, the name for the stable cell line - U2OS-FlpIN;Kif5b-rigor-2xmNeongreen - is unpronounceable.

2) Introduction line 34: the authors chose a difficult-to-understand sentence to say that microtubules carrying low levels of tubulin PTMs, which is often visualized by the presence of the C-terminal tyrosine on alpha-tubulin, are also dynamic. They should clarify this for readers not familiar with the field of tubulin PTMs. It is also important to mention that microtubules are not strictly

partitioned in tyrosinated and detyrosinated microtubules in cells, both forms do co-exist in the same microtubules and some microtubules have more detyr-tubulin, while others are more labelled with tyr-tubulin.

3) Along the lines of the point mentioned above, in line 76 the authors mention that their new sensor labels a subset of detyr-microtubules. Obviously, this reflects that the stable microtubules the sensor detects are not necessarily identical with the detyr-pool of microtubules, but it might also reflect that being detyr-positive can represent a range of detyr-levels on these microtubules. If the authors want to make a strong point about this, they should co-stain their cells with the tyr-tubulin antibody to check whether the detyr-microtubules that are not bound by the sensor are perhaps more tyrosinated? Alternatively, they should at least point out this possibility, and given that the figure is in the supplement, this could be done in the figure legend.

4) In the laser cutting experiments (Fig. 4) the authors show that microtubules labelled with their sensor do not depolymerize when cut. While they have shown in previous experiments that the new sensor does not appear to change microtubule dynamics, they have never formally shown that it does not prevent depolymerization, thus this possibility does still remain. The authors should mention this. They might consider showing that another recently published live-cell sensor of microtubules that is specific to tyrosinated microtubules (thus labelling the more dynamic microtubules) does not prevent depolymerization under the conditions the authors use in their experiments.

5) The observation in Fig. 4L that their sensor does also label dynamic microtubules, but much more transiently, is highly intriguing. Why does the sensor fall off the microtubules so quickly? In the light of the current literature, did the authors consider the possibility that the rigor mutant of kinesin actually could extrude single tubulin molecules from the microtubule lattice?

Point-to-point response for:

A live-cell marker to visualize the dynamics of stable microtubules throughout the cell cycle

Klara I. Jansen et al.

Point-to-point response to the reviewer comments

Reviewer #1

Cells contain a mixture of stable microtubules with slow turnover and dynamic microtubules that turn over rapidly. These different microtubule populations frequently are post-translationally modified, mainly acetylation, deetyrosination and glutamylation. However, while antibodies for these modifications have been in use for a long time and have served as proxy markers for stable microtubules, there are no live markers for stable microtubules. The lack of such markers has been an impediment into deeper cell biological investigations. The authors explore in this study the use of a rigor form of Kinesin-1 to preferentially label and image stable microtubules in live cells. They show they can image these stable microtubules over long times and observe their fate during cell division. While the idea is interesting and the imaging is well-done (not surprising from the Kapitein group), I have strong reservations about the rigor and significance of this work as it currently stands. This field has been muddled by a lot of phenomenology and a lack of mechanism. The work described here, while potentially interesting and useful, will only further confuse the field in its current state because it is not clear (i) what this rigor kinesin construct recognizes and (ii) what its effect is on microtubule dynamics. These points would have to be addressed in order for this marker to be reliable. In addition, given that this is a resource paper, the authors should show that their observations hold in at least one other cell line. There is also a lack of scholarship, with both old and new work not being cited. The authors fail to cite and discuss relevant studies that have direct impact on the interpretation of their experiments. My specific comments are below. I hope the authors will consider them.

- We are grateful to the reviewer for the thoughtful feedback on our manuscript. The reviewer raises two key questions: what does the marker recognize and what is the effect on microtubule dynamics?

In our manuscript, we describe a multitude of experiments that, in our opinion, convincingly demonstrate that our marker labels a subset of stable, long-lived microtubules. The existence of these microtubules is well-known and earlier correlative work has revealed that these stable microtubules are nocodazole-resistant, enriched in post-translational modifications, sensitive to serum starvation, and vary in abundance during the cell cycle. We directly confirm these findings in live cells using our new marker, which provides new opportunities to monitor the dynamics of stable microtubules during the cell cycle, as demonstrated in our mitosis experiments. In our opinion, this wide array of experiments validates our marker as a live-cell marker for the subset of long-lived, stable microtubules, even when it remains unknown what exactly this motor recognizes. In a similar fashion, EB proteins were used as markers for dynamic microtubules many years before it was understood what exactly they recognize at the microtubule plus-end.

Nonetheless, we were also very keen on understanding the selectivity of our marker and therefore started a collaboration with structural biologists when we first established our marker. Based on the emerging evidence in the literature that microtubules can exist in different lattice states and that these can be both recognized and reinforced by different microtubule-associated proteins, we hypothesized that stable microtubules inside cells feature an expanded microtubule lattice and that our marker recognizes this expanded lattice state. Our collaborators therefore analyzed microtubule lattice spacing inside cells using cryo-focused ion beam milling and cryo-electron microscopy, which revealed that a fraction of microtubules in wildtype cells indeed have an expanded lattice. Moreover, correlative cryo-light and electron microscopy revealed that our marker specifically decorates microtubules that have an expanded lattice. Together, these results demonstrate that stable microtubules inside cells display an expanded lattice and that this is what our marker recognizes. Given the enormous amount of highly-specialized work performed by our

collaborators and following the editorial guidance that we received, these results will be presented in a separate (follow-up) manuscript with different first and last authors.

With respect to the second point, the effect on microtubule properties, we realize that there is always an interplay between recognizing a conformation and inducing a conformation. Therefore, if we overexpress our marker at high levels it could alter the lattice of non-stable microtubules and thereby promote stabilization by promoting the binding of additional stabilizing proteins. Our manuscript has carefully addressed this by examining the effect of different levels of overexpression on acetylation, microtubule organization, and intracellular transport. Upon request of the reviewer, we performed additional controls and we have measured the concentration range in which our marker is a passive marker that does not alter microtubule physiology. These results are discussed in point 3 below.

In addition to these two key points, we now show the use of our marker in a panel of commonly used cell lines. We also went carefully over our references and added more citations to earlier work.

1. In Figure 1 the authors focus a lot of their attention on acetylation. They show that this marker is marking stable microtubules but not necessarily the acetylated ones. They compare the colocalization of Kin-1 and acetylated MTs on taxol-stabilized and hyper-acetylated microtubules. In the former their marker binds all MTs whereas without the Taxol stabilization it binds a subset of MTs. Could this be due to their Kinesin rigor construct binding differently to Taxol and non Taxol stabilized MTs? Their interpretation is supported by live cell nocodazole treatment, but it would be reassuring to see a quantification of the amount of stable MTs before and after washout with and without their marker. A small point but the dashed lines in fig 1b and S1b showing where the line profiles in s1c were taken from are black and very difficult to see, perhaps make them more obvious to the eye?

- Our data is consistent with the notion that acetylation is a modification that accumulates on stable microtubules, but does not directly confer significant stability to microtubules (i.e. protection to nocodazole-induced depolymerization). Thus, in control situations acetylation is found on stable, long-lived microtubules, while chemically promoting acetylation leads to the acetylation of dynamic microtubules without stabilizing them, as confirmed by nocodazole treatment. Our finding that our marker (now termed StableMARK) binds to Taxol-stabilized microtubules is consistent with the established result that Taxol induces an expanded lattice (shown in vitro, as well as in the new manuscript of our collaborators mentioned above).

For the revised manuscript, we quantified the data from the serum starvation assay and demonstrate that StableMARK-decorated MTs are not over-stabilized, as they behave similarly to the control condition upon prolonged serum starvation. In addition, reappearance of stable MTs upon release of prolonged serum starvation is similar between StableMARK-expressing and non-transfected cells. This data is presented in Fig.4G-I. Finally, we have also changed the dashed line in Fig.1B from black to red to increase visibility.

2. In Figure 2, the high curvature of stabilized microtubules has been observed by other groups before. Citation of this earlier work is encouraged (for example, see Friedman et al JCB 2010).

We agree that the high curvature of acetylated microtubules is a well-known feature that has been reported by many groups. We did not intend to suggest that these were new findings, but rather wanted to demonstrate that our marker specifically recognizes these stable microtubules and that we can now observe the spatiotemporal dynamics of this specific subset in live cells. We clarified this in the text and included more citations to earlier work.

3. One major issue with the argument that the authors are trying to make is that it is not clear whether their rigor kinesin is actually stabilizing microtubules. Microtubule dynamics parameters need to be determined in control cells and cells expressing their rigor kinesin. Just reporting EB1 comet numbers is not enough -what are the microtubule dynamic parameters? These are highly feasible experiments, especially for this lab. I am sure that at higher concentration levels the rigor kinesin will affect microtubule dynamics. What is the expression range for which this is a passive marker?

- Indeed, many microtubule-associated proteins will stabilize microtubules when expressed at high enough levels. Furthermore, live-cell markers are generally known to alter the system of interest when expressed at too high levels. This is well-known for LifeAct, EB1/3, and MT markers such as GFP-tubulin or the MT-binding domain of MAP7 (see Supporting Figure 1, below). Despite the artifacts that all of these widely used markers can induce, they remain well-accepted tools for cell biology because experimenters can be trained to recognize normal versus abnormal cells.

For our new marker, we have extensively analyzed how the level of stable and dynamic microtubules changes as a function of expression level. This revealed that the level of stable microtubules in cells (as identified by staining for acetylation) does not change at the expression levels that we use in our imaging. As expected, in higher expressing cells, the level of acetylation does increase, which is most likely the consequence of microtubule over-stabilization leading to more long-lived microtubules that accumulate more modifications. Importantly, our serum starvation experiments reveal that stable microtubules can still disappear when decorated with our marker, demonstrating that our marker does not over-stabilize this specific subset.

Now the question emerges whether experimenters that can be trained to identify normal cells while using LifeAct, EB1/3 or other MT markers can also be trained to identify cells with the proper level of our marker for stable microtubules. We think this is possible for a number of reasons. As shown in our manuscript, our marker overlaps with acetylated microtubules, which have a very stereotyped, perinuclear distribution and are often highly curved (see point 2). In our experience, it is quite straightforward to identify the population of cells whose expression levels result in a distribution of the marker that is consistent with stainings for acetylation in control cells. In addition, for the revised manuscript we have performed a series of fluorescence correlation spectroscopy experiments in order to measure the concentration of our marker in cells that we classify as low, medium or high expressing. This revealed that our marker can be found in concentrations ranging around 0.02-72 μM , where concentrations below 2 μM correspond to expression levels where we find no over-acetylation. This data is present in Sup.Fig.3D

Therefore, we believe that the general behavior of our marker (mimicking wildtype cells at levels below 2 μM and gradually inducing over-stabilization in a concentration-dependent manner above 3 μM) is very similar to many other, widely used live cell markers and that the various control experiments presented in our manuscript should be sufficient to guide new users towards the proper use of this marker in their model systems.

In our revised manuscript we also present further evidence that many stable microtubules do not have a dynamic end. Figure 6D-F shows that EB3 comets only rarely emerge from StableMARK-decorated microtubules, whereas Sup. Fig.4 B-D shows that also in wildtype cells most acetylated microtubules do not have a non-modified extension. Thus, in cells with expression levels between 0.02-2 μM , dynamic microtubules are not decorated with our marker. Consistently, we do not see a difference in the growth of microtubule plus ends between cells with and without our marker (Fig. 4F).

4. In vitro work showed that de-tyrosination affects the binding of kinesin-1 more so than acetylation (Kaul et al 2014 - also not cited) yet the authors seem to have focused on acetylation. Is it possible that this is a marker for de-tyrosinated microtubules? They show that at low levels the amount of acetylated tubulin is about the same with and without the rigor mutant. (Figure 3A). It would be reassuring to see the same for de-tyr tubulin, and to see a quantification for this. If the marker responds more to tyrosination than acetylation they also may be missing a fraction of the stabilized microtubules.

- For the revised manuscript, we have now also tested the overlap between our marker and de-tyrosinated microtubules. We find that in control U2OS cells, de-tyrosinated microtubules are very sparse and that our marker clearly marks more microtubules. When we increase de-tyrosination by overexpression of VSH1 and SVBP most microtubules become de-tyrosinated (but not stabilized), while our marker still labels a subset of microtubules. This data is presented in Sup.Fig.1A-G. Based on this new data and the earlier data on acetylation, we conclude that our marker does not recognize either of these modifications directly, but does localize to microtubules that are acetylated. As discussed above, we have now found that the stable microtubules recognized

by kinesin-1 have an expanded lattice and we think that this is what determines the selective binding of our marker, either directly or indirectly.

5. *The Rab6A transport data needs to be quantified to see whether transport speeds are changed by the presence of more possible roadblocks (ie rigor kinesin) on the MTs. (Figure 3G). The transport kinetics of other cargos should also be analyzed that use other motors than the two used for Rab6A transport (for example, a connection between lysosome transport and acetylated MTs has been previously established).*

- We have now quantified the speeds of lysosomes and Rab6a vesicles in control cells and low StableMARK expressing cells. We found that the distributions of vesicle speeds in control cells and rigor-expressing cells are very similar. This data is presented in Fig.5F-I.

The serum starvation experiments are nice, but all they show is that the marked MTs can be depolymerized by this assay (already known). In general, the comment that "our data demonstrate that in the subpopulation of low rigor-expressing cells, artifacts of the MT cytoskeleton are minimal" is not backed up by sufficient data. It might very well be the case, but they need to do more to prove this so that this can be a marker that can be used with confidence in the field.

- As argued in our response to point 3, we believe that our data demonstrates that, when used at the proper expression level, our marker labels the existing subset of stable microtubules and does not lead to stabilization of additional microtubules. Nonetheless, our marker could still lead to additional stabilization of the stable microtubules. Earlier work has shown that upon serum starvation in 3T3 cells, stable microtubules disappear. We reasoned that if our marker would confer a strong additional stabilization, these microtubules might no longer disappear upon serum starvation. We therefore repeated this classic experiment and found that decoration with our marker did not prevent the depolymerization of stable microtubules, indicating that our marker does not overstabilize stable microtubules. In the revised manuscript, we have further quantified these experiments by comparing the level of acetylation with and without serum starvation in cells with and without expression of our marker and found no difference between non-transfected control cells and low StableMARK-expressing cells (Fig.4G-I).

6. *Their investigation of the mechanism of stabilization is technically well-performed but I have reservations with their interpretation. Upon ablation they see ~50% of the MTs depolymerize from one end which they interpret as an aging effect, where stabilization starts at the +tip and is followed by stabilization along the lattice. There have been several papers on motor induced damage and long-range lattice effects, including the stabilization of the lattice by even substoichiometric amounts of kinesin (Andreu-Carbo et al, 2022, Triclin et al 2021; Peet et al 2018). These need to be discussed here as they point to the possible perturbing role that their marker can actually have. If these microtubules also are healed more, they will also be more stable when ablated because of the incorporation of GTP tubulin (Aumeier et al. 2016; Vemu et al 2018).*

- From our observation that not all rigor-decorated (and nocodazole-resistant) microtubules remain stable after laser ablation, we suggest that stable microtubules might display different levels of stability, which increase over time. This idea is not inconsistent with the papers cited by the reviewer, which all evoke a positive feedback between kinesin-based stabilization and the recruitment of more kinesins and additional stabilization, thereby implying that the level of stabilization can vary from microtubule to microtubule.

Recent studies have indeed reported that motor proteins can promote tubulin turnover in reconstitution assays with purified proteins (Triclin et al., 2021, Andreu-Carbo et al., 2022). Likewise, recent in vitro work with spastin has also shown that this microtubule destabilizer can promote the incorporation of GTP-tubulin (Vemu., et al 2018). However, the evidence for tubulin turnover along the microtubule lattice in living cells and its role in microtubule stabilization is less conclusive. Importantly, even if active motors could induce lattice defects that somehow promote stabilization, this process is generally believed to depend on the energy derived from ATP hydrolysis. Since our marker does not hydrolyze ATP it is unlikely that it will cause the type of microtubule damage that has been reported in the above mentioned in vitro assays.

Other in vitro experiments (Peet et al 2018, Shima et al 2018) have revealed that kinesins can induce other changes in the microtubule lattice that are more subtle, but perhaps also more consequential. Purified kinesin motors can cause small extensions of in vitro polymerized microtubules, most likely by inducing a different microtubule lattice spacing (Peet et al 2018, Shima et al 2018). Thus, even without inducing the exchange of tubulin subunits, kinesins can induce a GTP-like (expanded) tubulin conformation within a GDP-tubulin lattice. As explained above, our collaborators have now found that a fraction of microtubules in wildtype cells indeed features an expanded lattice. Moreover, correlative light and electron microscopy revealed that our marker specifically decorates microtubules with an expanded lattice. These results support a model in which stable microtubules inside cells feature a specific lattice, which might be recognized and reinforced by a large number of MAPs, including kinesin-1. Since lattice expansion appears to be a cooperative effect, this also explains why our marker (and many other MAPs) can induce microtubule stabilization when expressed too highly, but is mostly reading out the lattice state at low concentrations. Following the editorial guidance that we received, these results will be presented in a separate manuscript by our collaborators.

7. When discussing laser ablation experiments and the asymmetry in depolymerization between the plus and minus ends, the authors strangely cite a paper from 2014. This is textbook work from Salmon and colleagues for example and should be cited.

➤ We cited the 2014 paper because it had used the exact same experimental configuration. In our revised manuscript, we now also cite the original work from the Salmon lab.

8. The authors make the interesting observation that the rigor kinesin coated MTs do not have dynamic (EB3) tips. Based on this they speculate that therefore there is a "special tip structure" on stabilized microtubules that prevents both growth and shrinkage. Could it be that their rigor mutant interferes with normal EB3 binding?

➤ To address this question we reasoned that if the lack of dynamic tips on stable microtubules would be induced by our marker, we should be able to find dynamic ends on many acetylated microtubules in non-transfected cells. We performed super-resolution microscopy (STED) on U2OS cells stained for acetylated tubulin and total tubulin and quantified the percentage of acetylated MT tips that had a dynamic end (as manifested by a stretch of total tubulin extending from the acetylated MT tip). We found that 89% of the acetylated MT tips did not have a dynamic end, which is very similar to what we found in our live-cell imaging experiments using StableMARK and EB3. We therefore conclude that the limited polymerization at the tip of stable MTs is not caused by the presence of low levels of StableMARK. This new data is presented in Sup.Fig.4B-D.

9. What is the effect of this rigor mutant on microtubule dynamics in vitro? These are highly feasible experiments that should be performed. And does it affect EB binding? Mechanistic data on what the effect of this mutant is on microtubule dynamics and binding of some ubiquitous effectors is needed.

➤ As discussed in the previous comment, we now provide additional support for our finding that most of the stable microtubules recognized by our marker in U2OS cells do not have dynamic end. Therefore, in vitro assays for microtubule dynamics that use seeds and free tubulin do not really recapitulate the properties of this specific subset of microtubules. Nonetheless, we initiated the purification of the rigor mutant to test its effect on in vitro microtubules. Unfortunately, we experienced some unexpected difficulties and delays in setting up these experiments. Because, as explained, there is no straightforward connection between such reconstitution assays and our cellular assays, we decided to focus on addressing the other comments.

10. The rigor kinesin seems to be decorating only a portion of the stable MT populations in the spindle - why? This question brings me back to my biggest reservation for this work: the authors do not know what features this kinesin mutant recognizes. So, yes it might mark some stable microtubules, but maybe that will change with circumstance (if it is not stability that it marks, but something else).

➤ In our revised manuscript we show the distribution of stable (acetylated) and total MTs in the spindle of U2OS WT cells. In cells expressing the StableMARK, this distribution is very similar,

and the rigor localizes clearly to the subset of acetylated MTs. In addition, we performed cold-treatments to only preserve the kinetochore fibers. This showed that our marker indeed localizes to the more stable kinetochore fibers during metaphase. See Sup.Fig.5A.

Reviewer #2

This paper describes the use of a rigor mutant of kinesin-1 to identify stable microtubules (MTs) in living cells. authors show convincing data that a 2X-mNeon-tagged version of the rigor kinesin-1 marks stable, acetylated MTs in cells. They then use the rigor kinesin to follow stable MT dynamics in interphase and mitotic cells and show a number of behaviors of these MTs, many of which have been reported previously. Overall, the development of a live cell probe for stable MTs is an unmet need for cell biologists and has the potential to reveal novel behaviors of these MTs. However, the paper has some major issues (see below) that need to be addressed before the rigor kinesin-1 is characterized well enough to be a reliable live cell marker for stable MTs.

- We thank the reviewer for the thoughtful comments on our manuscript. We believe that the additional data presented in our revised manuscript addresses the key concerns of the reviewer.

Key issues:

1. The authors need to establish the basis for recognition of the stable MTs by the rigor kinesin-2 sensor. The rigor mutant labels stable, acetylated MTs, yet they also show by manipulating acetylation that it is not the basis for the detection of the stable MTs. Without understanding the basis for the recognition of the stable MTs by the rigor mutant, MTs detected by the rigor kinesin are only operationally defined as stable. This feature of the paper has great potential to create confusion/controversy when drawing conclusions from results with the rigor kinesin obtained in different systems and by different investigators. There is abundant evidence that kinesin-1 exhibits elevated binding to MTs with detyrosinated tubulin (e.g., Liao G and Gundersen GG, JBC, 1996; Dunn S et al, JCS 2008; Cai D et al, PLOS Biol, 2009; Sirajuddin M. et al, NCB, 2014), and the rigor kinesin should be tested against this posttranslational modification.

- In our opinion, an operational definition of microtubule stability is the most correct one. Microtubules are stable if they resist cold-treatment or nocodazole treatment, or if they remain intact after being mechanically challenged (e.g. by laser ablation). This is also the reason why we validated our marker in these different conditions. The exact mechanism of stabilization might differ from cell to cell, because many different microtubule-associated proteins can contribute to stability and their expression differs from cell to cell.

Nonetheless, we were also very keen on understanding the selectivity of our marker and therefore started a collaboration with structural biologists when we first established our marker. Based on the emerging evidence in the literature that microtubules can exist in different lattice states and that these can be both recognized and reinforced by different microtubule-associated proteins, we hypothesized that stable microtubules inside cells feature an expanded microtubule lattice and that our marker recognizes this expanded lattice state. Our collaborators therefore analyzed microtubule lattice spacing inside cells using cryo-focused ion beam milling and cryo-electron microscopy, which revealed that a fraction of microtubules in wildtype cells indeed have an expanded lattice. Moreover, correlative cryo-light and electron microscopy revealed that our marker specifically decorates microtubules that have an expanded lattice. Together, these results demonstrate that stable microtubules inside cells display an expanded lattice and that this is what our marker recognizes. Given the enormous amount of highly-specialized work performed by our collaborators and following the editorial guidance that we received, we decided to present these results in a separate (follow-up) manuscript with different first and last authors.

For the revised manuscript, we have now also tested the overlap between our marker and detyrosinated microtubules. We find that in control U2OS cells, detyrosinated microtubules are very sparse and that our marker clearly marks more microtubules. When we increase detyrosination by overexpressing VSH1 and SVBP, most microtubules are detyrosinated (but not stabilized), while our marker still labels a subset of microtubules. This data is presented in Sup.Fig.1A-G. Based on this new data and the earlier data on acetylation, we conclude that our marker does not recognize either of these modifications, but does localize to microtubules that are acetylated. As discussed above, we have now found that the stable microtubules recognized by kinesin-1 have an expanded

lattice and we think that this is what determines the selective binding of our marker, either directly or indirectly.

2. The relationship between the level of rigor kinesin-1 expression and artifacts caused by its expression are incompletely documented. The authors conclude (and I am not sure I agree, for example EB1 comets and hence dynamic MTs are clearly affected in Fig. 3C), that at "low" expression there is little effect on MTs, whereas at "high" expression, stable MTs increase, dynamic MTs are reduced, and overall MT distribution is disrupted. "Low" and "high" expression can mean different things to different people. Consequently, for the method to be reproducible and widely useful, it is critical that the authors define the level of rigor kinesin-1 expression that begins to show the artifacts they observe. For example, a comparison of the level of the rigor kinesin-1 to the endogenous level of kinesin-1 (or tubulin) would be helpful. Given the cell-cell variability with transient expression, it might be easier to make this comparison with stable cell lines with low and high expression.

➤ Indeed, many microtubule-associated proteins will stabilize microtubules when expressed at high enough levels. Furthermore, live-cell markers are generally known to alter the system of interest when expressed at too high levels. This is well-known for LifeAct, EB1/3, and MT markers such as mCherry-tubulin or the MT-binding domain of MAP7 (see Supporting Figure 1, above). Despite the artifacts that all of these widely used markers can induce, they remain well-accepted tools for cell biology because experimenters can be trained to recognize normal versus abnormal cells.

For our new marker, we have extensively analyzed how the level of stable and dynamic microtubules changes as a function of expression level. This revealed that the level of stable microtubules in cells (as identified using staining for acetylation) does not change at the levels that we use in our imaging. As expected, in higher expressing cells, the level of acetylation does increase, which is most likely the consequence of microtubule overstabilization leading to more long-lived microtubules that accumulate more modifications. Furthermore, our serum starvation experiments reveal that stable microtubules can still disappear when decorated with our marker, demonstrating that our marker also does not overstabilize this specific subset.

Now the question emerges whether experimenters that can be trained to identify normal cells while using LifeAct, EB1/3 or other MT markers can also be trained to identify cells with the proper level of our marker for stable microtubules. We think this is possible for a number of reasons. As shown in our manuscript, our marker overlaps with acetylated microtubules, which have a very stereotyped, perinuclear distribution and are often highly curved. In our experience, it is quite straightforward to identify the population of cells whose expression levels result in a distribution of the marker that is consistent with stainings for acetylation in control cells. In addition, for the revised manuscript we have performed a series of fluorescence correlation spectroscopy experiments in order to measure the concentration of our marker in cells that we classify as low, medium or high expressing. This revealed that our marker can be found in concentrations ranging around 0.02-72 μM , where concentrations below 2 μM corresponds to expression levels where we find no over-acetylation. This data is present in Sup.Fig.3D.

Therefore, we believe that the general behavior of our marker (mimicking wildtype cells at levels between 0.02-2 μM and gradually inducing overstabilization in a concentration-dependent manner above 3 μM) is very similar to many other, widely used live cell markers and that the various control experiments presented in our manuscript should be sufficient to guide new users towards the proper use of this marker in their model systems.

3. There is very little new about stable MTs that is reported with the rigor kinesin leading me to wonder whether it will be all that useful. Part of this reflects the fact that the authors repeat previous findings that they are apparently unaware of. For example, previous studies have shown that stable MTs are curly (Piperno G et al, JCB, 1987 and others), do not grow for extended periods (Schulze E and Kirschner M, JCB, 1986; Webster D et al., PNAS, 1987; Infante AS et al, JCS, 2000; Palazzo AF et al, NCB, 2001), and depolymerize slowly when severed (Walker RA et al, JCB, 1989; Infante AS et al, JCS, 2000). The authors also seem unaware of all the molecular studies implicating numerous factors (Rho, CLASPs, ELKs, mDia, INF2) in stable MT formation. These studies should be acknowledged and additional studies should be considered to show the utility of the rigor kinesin.

- As recognized by all three reviewers, *'the development of a live cell probe for stable MTs is an unmet need for cell biologists and has the potential to reveal novel behaviors of these MTs.'* The goal of this manuscript was to carefully validate this new marker and to demonstrate that we can reproduce earlier findings on stable microtubules obtained more indirectly. As such our demonstration of curly microtubules, lack of growth, slow depolymerization upon severing etc. etc. was not intended to make new claims, but to validate our marker. We clarified this in the text and included more references to earlier work. In addition, we demonstrate new insights into the (re)organization of stable microtubules throughout the cell cycle. However, we never intended to extensively study the multiple mechanisms of stabilization by the well-known factors listed by the reviewer. After depositing our preprint, we have already shipped out our plasmid to many labs that have specific research questions and are excited to finally have a validated marker for live-cell imaging of stable microtubules, for example in specific cell types of various model organisms.

As outlined in point 1, we have now also used our marker in a collaborative study with cryo-EM experts to study the lattice spacing of stable microtubules inside cells. Cryo-EM is an extremely powerful technique for structural studies inside cells, because it preserves cellular ultrastructure so well. Nonetheless, it is impossible to identify different microtubule subset, because the fixation procedures are incompatible with antibody-based labeling procedures. So this is an example where our live-cell marker is extremely useful. Because it already decorates a subset of microtubules in live cells, it can also be used to identify these microtubules in cryo-fixed cells and to compare the microtubule lattice of stable and dynamic microtubules.

Additional comments:

4. *The experiments showing that acetylated tubulin accumulates in MTs resistant to nocodazole and increases in taxol treated MTs shown in Fig 1F-G are not new or news. The data should be moved to the supplement.*

- As pointed out by the reviewers, there are widely varying ideas about the causes and consequences of microtubule stabilization. These experiments are connected to the subsequent experiments in control cells and StableMARK-expressing cells, which together convey an important point and we prefer to present them together. Based on the feedback we received when presenting this work, it also seems that, despite earlier key work from the Gundersen lab that we cite, the result that hyperacetylation does not lead to stabilization of the newly acetylated microtubules is not considered general knowledge.

5. *The description of the behaviors of rigor kinesin decorated MTs are imprecise and in some cases over interpreted. For example, it is unclear how the authors draw the conclusion that the MT shown in Fig. 2C is sliding as it could also be treadmilling or trapped in actin flow. The MT in Suppl Fig2C might be moving rather than depolymerizing. And, I do not see active looping in fig. 2E as described.*

- The speckled labeling by our marker enabled us to discriminate between sliding and treadmilling. The kymograph shown in Fig. 2D (now Fig.3D) shows that the speckles are moving together, which indicates that the microtubule moves as a whole and rules out treadmilling. Because the velocity of such motile events was in the order of the velocity of microtubule-based motors (see original Fig. 2F (now Fig.3F) and much faster than the typical speed of actin flow, we reasoned that sliding was the most logical interpretation. In Suppl Fig. 2C, our interpretation that this is a partial depolymerization event was based on the observation that the speckled pattern on the remaining microtubule does not change over time, as also shown in the kymograph in Suppl Fig. 2D. For Fig. 2E (now Fig.3E), we changed the wording to active curvature induction.

6. *I observe a substantial change in the overall distribution of MTs cause by "high" rigor kinesin expression in Fig. 3A. This should be mentioned along with the other alterations cause by expression of the rigor kinesin.*

- This was already mentioned in the text, but we will better highlight this.

7. *The comparison in Fig3B between rigor kinesin expression and acetylated tubulin levels needs to include the acetylation levels in non-expressing cells. Otherwise, their claim that expression of the rigor*

kinesin does not affect acetylated tubulin levels is not valid. Additionally, normalizing the rigor kinesin levels to the total tubulin levels makes no sense to me - the graph should simply show rigor kinesin expression.

- Fig.3B the mean + SD for non-expressing cells is shown by the green lines. We normalized the rigor kinesin to the total tubulin levels in order to robustly compare images taken on different days, with perhaps slightly different illumination conditions.

8. The authors should plot EBI comets per cell rather than EBI comets/um to capture the clear reduction in the numbers of comets in both "low" and "high" expressing cells shown in Fig. 3C.

- We chose to quantify per area, because we noted that larger (untransfected) cells typically have more comets.

9. The data that Rab6A transport is unaffected needs quantification beyond the anecdotal example shown in Fig. 3G-I.

- For the revised version, we quantified the speeds of lysosomes and Rab6a vesicles in control cells and low StableMARK expressing cells. We found that the distributions of vesicle speeds in control cells and rigor-expressing cells are very similar. This data is presented in Fig.5F-I.

10. Many have noted that the turnover of spindle MTs (excluding mid-zone MTs, which are well known to be more stable), is an order of magnitude faster than interphase MTs. The rigor kinesin seems to decorate a subset of spindle MTs raising the question whether it is really detecting stable MTs in all cases. This result also illustrates the uncertainty with what the rigor kinesin is actually detecting.

- In our revised manuscript we show the distribution of stable (acetylated) and total MTs in the spindle of U2OS WT cells. In cells expressing the rigor, this distribution is very similar, and the rigor localizes clearly to the subset of acetylated MTs. In addition, we performed cold-treatments to only preserve the kinetochore fibers. This showed that our marker indeed localizes to the more stable kinetochore fibers during metaphase. See Sup.Fig.5A.

Reviewer #3

Jansen et al. describe a new live-cell sensor that specifically detects stable microtubules. This is an important advance for cell biology, as so far, labelling of distinct microtubule species in cells was restricted to the use of antibodies, i.e. fixed cells. Being able to detect specific microtubule subtypes in living cells is a huge advance that will help the community to substantially advance the understanding of roles of microtubule specialization in living cells, and to image how those microtubules behave. The authors already provide an exciting glimpse on those opportunities by showing how their new live-cell marker allows them to follow the fate of stable microtubules throughout cell division.

The presence of stable microtubules within a dynamic microtubule network as it is found in undifferentiated interphase cells has for many decades intrigued the scientific community. Because these microtubules are the only ones in cells to accumulate posttranslational modifications such as acetylation and deetyrosination, it has for long been disputed whether these modifications are the cause or the consequence of stabilization of those microtubules. Recent work showing that acetylation decreases the flexural rigidity of microtubules, thereby making them more resilient to repeated mechanical stress suggested that acetylation could cause at least longevity of microtubules in cells. However, whether the observed stable microtubules in cells are solely stable because of their posttranslational modifications has remained an open question.

The current work now provides an innovative approach to study stable microtubules in cells. The authors found that a rigor mutant of the motor protein Kif5b, when expressed at low levels in cells, specifically binds to stable microtubules, which are typically acetylated (tubulin acetylation at K40) in cells. To test whether the species of microtubules that their sensor detects could simply be the acetylated microtubules, the authors performed a series of experiments with drugs that either stabilize microtubules, or prevent their polymerization, or boost tubulin acetylation. Strikingly, they find that boosting acetylation does not increase microtubule stability in cells, and accordingly their sensor does not label more microtubules in cells with boosted acetylation. However, it does so in cells treated with the microtubule stabilizer taxol, indicating that the nature of stable microtubules - detected with the new sensor - is not restricted to their posttranslational acetylation.

After showing that their new sensor can be used as a live-cell marker for stable microtubules, the authors demonstrate by different complementary experiments that the binding of (low doses) of their new sensor does not perturb overall microtubule behavior in cells, including posttranslational modifications, microtubule dynamics, motility of kinesin motors, or intracellular architecture. They go on demonstrating that the microtubules that are specifically labelled with their sensor are indeed stable: they demonstrate that they do not depolymerize after laser cutting, which typically would happen for dynamic microtubules. They further demonstrate that the new sensor is not exclusively binding stable microtubules, but binds other microtubules in cells much more transiently, suggesting a mechanism that increases its binding affinity specifically to the stable microtubules.

Finally, the authors demonstrate how the new sensor can be used to monitor the fate and dynamics of stable microtubules in dividing cells. These impressive movies illustrate the potential of their new tool to obtain unprecedented insight into the behavior of specific microtubule subtypes in cells. The manuscript is carefully written, figures are of high quality and most experiments have been performed thoroughly with appropriate controls. Nonetheless, several points need to be addressed before the paper can be considered for publication.

- We thank the reviewer for the positive assessment of our manuscript and the helpful suggestions that have helped us to improve our manuscript.

Major points:

1) In Fig. 3A the authors show that low expression levels of their sensor do not increase tubulin acetylation in cells. However, as a control example, they show a cell with highly bundled microtubules around the nucleus (row 1), which is not the case of the cell expressing the sensor (row 2). One could speculate that the acetylation levels in the control are higher due to the microtubule bundling, which is absent from the cell in row 2 - and thus conclude that low expression levels of the sensor lead to levels of acetylation that can only be attained in control cells upon microtubule bundling. To show that this is

not the case, the authors should choose cells that are more similar in terms of microtubule bundling for this figure, i.e. a control without microtubule bundles.

- We changed the example shown for the control condition. In addition, we now added Sup. Fig.6A, which shows for six different cell types a large field of view with multiple rigor-transfected and non-transfected cells. Here, the rigor-transfected cells are indiscernible from the non-transfected cells in the acetylated and total tubulin channel.

2) Line 175, Fig 3G-I: this is the only experiment in this figure which does not show statistical analyses of the observation - the authors show one selected movie. How reproducible are these observations?

- In our revised manuscript, we quantified the speeds of lysosomes and Rab6a vesicles in control cells and in low expressing StableMARK cells. We found that the distribution of vesicle speeds in control cells and rigor-expressing cells are very similar. This data is presented in Fig.5F-I.

3) In Fig. 5D, the authors show how stable microtubules accumulate in the spindle midzone over time using their new sensor. However, they do not measure the total microtubule load of the midzone, thus it is not clear whether what they measure is simply an accumulation of microtubules.

- Because mitotic cells are very sensitive to photo-toxicity, we could not follow the accumulation of StableMARK and a total tubulin marker over time in live cells. Therefore, we quantified the accumulation of acetylated tubulin and total tubulin during different stages of cytokinetic bridge formation in U2OS WT cells. This clearly showed the relative increase in acetylated tubulin (and thus stabilization of MTs) during the formation of the cytokinetic bridge. This data is presented in Sup.Fig.5B,C.

4) One of the major drawbacks of their tool is that they need to select cells with "low expression levels". What this actually means remained unclear throughout the manuscript, nor did they mention how many cells in a population of transfected cells would fulfill this criterium and could thus be used for analysis. The authors should thus show a large field of view of a typical cell culture transfected with their sensor, and label the cells in this field that they consider "low expression" and "high expression". They might also want to give numbers for this, which should be easy by re-analyzing their experiments. Did the authors consider the use of lentivirus to transduce cells with their sensor? This would most likely allow a better tuning of the expression levels.

- We agree that the need to select cells with the proper expression level could be perceived as a drawback. However, in our experience the situation is not different for many other widely used live-cell markers. For example LifeAct, EB1/3, and MT markers such as mCherry-tubulin or the MT-binding domain of MAP7 are generally known to alter the system of interest when expressed at too high levels. Despite the artifacts that all of these widely used markers can induce, they remain well-accepted tools for cell biology because experimenters can be trained to recognize normal versus abnormal cells.

Now the question emerges whether experimenters that can be trained to identify normal cells while using LifeAct, EB1/3 or other MT markers can also be trained to identify cells with the proper level of our marker for stable microtubules. We think this is possible for a number of reasons. As shown in our manuscript, our marker overlaps with acetylated microtubules, which have a very stereotyped, perinuclear distribution and are often highly curved. In our experience, it is quite straightforward to identify the population of cells whose expression levels result in a distribution of the marker that is consistent with stainings for acetylation in control cells. In addition, for the revised manuscript we have performed a series of fluorescence correlation spectroscopy experiments in order to measure the concentration of our marker in cells that we classify as low, medium or high expressing. This revealed that our marker can be found in concentrations ranging from 0.02 μM – 72 μM , where concentrations below 2 μM corresponds to expression levels where we find no over-acetylation. This data is presented in Sup.Fig.3D.

Therefore, we believe that the general behavior of our marker (mimicking wildtype cells at levels below 2 μM and inducing overstabilization above 3 μM) is very similar to many other, widely

used live cell markers and that the various control experiments presented in our manuscript should be sufficient to guide new users towards the proper use of this marker in their model systems.

Minor points:

1) *The authors have developed a great tool that will certainly be used a lot in the future. However, the name they have chosen for their binder, rigor-2xmNeongreen, is hard to memorize and even hard to pronounce. They should think of a more simple, intuitive name. Also, they should use the same name throughout the manuscript - currently they use different nomenclatures in the figures and the text. Along the same lines, the name for the stable cell line - U2OS-FlpIN;Kif5b-rigor-2xmNeongreen - is unpronounceable.*

➤ Following the reviewer's suggestion, we have changed the name of our marker to StableMARK (Stable Microtubule-Associated Rigor-Kinesin) and applied it uniformly in text and figures.

2) *Introduction line 34: the authors chose a difficult-to-understand sentence to say that microtubules carrying low levels of tubulin PTMs, which is often visualized by the presence of the C-terminal tyrosine on alpha-tubulin, are also dynamic. They should clarify this for readers not familiar with the field of tubulin PTMs. It is also important to mention that microtubules are not strictly partitioned in tyrosinated and detyrosinated microtubules in cells, both forms do co-exist in the same microtubules and some microtubules have more detyr-tubulin, while others are more labelled with tyr-tubulin.*

➤ We clarified this in our revised manuscript.

3) *Along the lines of the point mentioned above, in line 76 the authors mention that their new sensor labels a subset of detyr-microtubules. Obviously, this reflects that the stable microtubules the sensor detects are not necessarily identical with the detyr-pool of microtubules, but it might also reflect that being detyr-positive can represent a range of detyr-levels on these microtubules. If the authors want to make a strong point about this, they should co-stain their cells with the tyr-tubulin antibody to check whether the detyr-microtubules that are not bound by the sensor are perhaps more tyrosinated? Alternatively, they should at least point out this possibility, and given that the figure is in the supplement, this could be done in the figure legend.*

➤ For the revised manuscript, we have now also tested the overlap between our marker and detyrosinated microtubules. We find that in control U2OS cells detyrosinated microtubules are very sparse and that our marker clearly marks more microtubules than strictly the set of detyrosinated microtubules. When we increase detyrosination by overexpressing VSH1 and SVBP most microtubules are detyrosinated (but not stabilized), while our marker still labels a subset of microtubules. This data is presented in Sup.Fig.1A-G. Based on this new data and the earlier data on acetylation, we conclude that our marker does not recognize either of these modifications, but does localize to microtubules that are acetylated. As discussed in point 5 below, we now also have structural data revealing that the stable microtubules recognized by kinesin-1 have an expanded lattice and we think that this is what determines the selective binding of our marker, either directly or indirectly.

4) *In the laser cutting experiments (Fig. 4) the authors show that microtubules labelled with their sensor do not depolymerize when cut. While they have shown in previous experiments that the new sensor does not appear to change microtubule dynamics, they have never formally shown that it does not prevent depolymerization, thus this possibility does still remain. The authors should mention this. They might consider showing that another recently published live-cell sensor of microtubules that is specific to tyrosinated microtubules (thus labelling the more dynamic microtubules) does not prevent depolymerization under the conditions the authors use in their experiments.*

➤ Earlier laser ablation experiments in cells without our marker revealed that, even in the absence of minus-end stabilizing proteins, a subset of microtubules would not depolymerize upon laser-induced severing (Jiang et al., Dev Cell 2014). This indicates that slow depolymerization of a subset of microtubules is not necessarily induced by the expression of our marker. To examine whether our marker could potentially overstabilize stable microtubules and prevent their depolymerization, we have performed serum starvation experiments. Previous work has shown that stable

microtubules disappear upon starvation. We found that in cells expressing our marker, starvation-induced depolymerization was not impaired, indicating that our marker does not prevent depolymerization. We have further quantified these findings in the revised manuscript (see Fig.3I).

5) The observation in Fig. 4L that their sensor does also label dynamic microtubules, but much more transiently, is highly intriguing. Why does the sensor fall off the microtubules so quickly? In the light of the current literature, did the authors consider the possibility that the rigor mutant of kinesin actually could extrude single tubulin molecules from the microtubule lattice?

- Recent studies have indeed reported that motor proteins can promote tubulin turnover in reconstitution assays with purified proteins (Triclin et al., 2021, Andreu-Carbo et al., 2022). This process is generally believed to depend on the energy derived from ATP hydrolysis. Since our marker does not hydrolyze ATP, it is unlikely that it will cause the type of microtubule damage that has been reported in the above mentioned in vitro assays.

Importantly, other in vitro experiments (Peet et al 2018, Shima et al 2018) have revealed that kinesins can also induce more subtle changes in the microtubules lattice. Purified kinesin motors can cause small extensions of in vitro polymerized microtubules, most likely by inducing a different microtubule lattice spacing (Peet et al 2018, Shima et al 2018). Thus, even without inducing the exchange of tubulin subunits, kinesins can induce a GTP-like (expanded) tubulin conformation within a GDP-tubulin lattice. Because all these results were from in vitro experiments with purified components, we were curious to understand how this related to cells and thus started a collaboration with structural biologists to study microtubule lattice spacing inside cells. This revealed that a fraction of microtubules in wildtype cells indeed has an expanded lattice. Moreover, correlative light and electron microscopy revealed that our marker specifically decorated microtubules with an expanded lattice. These results support a model in which stable microtubules inside cells feature a specific lattice, which might be recognized and reinforced by a large number of microtubule-associated proteins, including kinesin-1. We therefore think that the transient binding to dynamic microtubules reflects a much lower affinity for compacted lattices and the inability of our marker to adopt the two-head bound rigor-conformation on these microtubules. More specifically, we hypothesize that the inter-dimer spacing of the expanded lattice promotes the rigor to adopt a two-head-bound state, while for the compacted lattice, this is not be the case. These structural studies of microtubule lattice spacings in cells were spearheaded by our collaborators and are documented in a complementary manuscript, following the editorial guidance that we received from Journal of Cell Biology.

Supporting figure 1: microtubule network artefacts induced by high over-expression of commonly used markers

October 5, 2022

Re: JCB manuscript #202106105RR-A

Prof. Lukas Kapitein
Utrecht University
Padualaan 8
Utrecht 3533 CH
Netherlands

Dear Lukas,

Thank you for submitting your revised manuscript entitled "A live-cell marker to visualize the dynamics of stable microtubules throughout the cell cycle" to JCB. Please accept our apologies for the delay with the review process and thank you for your patience. It has now been examined by the original three reviewers. While the reviewers continue to be overall positive about the work in terms of its suitability for JCB, you will see that Reviewer 1 feels strongly that a clear demonstration showing that the marker recognizes an expanded microtubule lattice is required, and suggests what appears to be a straightforward experiment. We hope that you can address this as well as the other minor points raised by the reviewers, and submit a final version of the manuscript.

Please note that JCB policy for co-submitted papers is that each one should stand on its own, therefore the final acceptance of your paper will not depend on the status of the companion manuscript, but will require conclusive in vitro evidence of binding to expanded lattice microtubules as well as discussion of any caveats about specificity.

Our general policy is that papers are considered through only one revision cycle; however, given that the suggested changes are relatively minor we are open to one additional short round of revision. Please note that I will expect to make a final decision without additional reviewer input upon resubmission.

Please submit the final revision within one to two months, along with a cover letter that includes a point by point response to the remaining reviewer comments.

Thank you for this interesting contribution to Journal of Cell Biology. You can contact me or the scientific editor listed below at the journal office with any questions, cellbio@rockefeller.edu or call (212) 327-8588.

Sincerely,
Rebecca

Rebecca Heald, PhD
Monitoring Editor
Journal of Cell Biology

Dan Simon, PhD
Scientific Editor
Journal of Cell Biology

Reviewer #1 (Comments to the Authors (Required)):

I thank the authors for their detailed reviewers' response. However, I remain skeptical about what this marker actually recognizes, and I am afraid that without a clear understanding of this, the current manuscript, the way it stands, will only add more confusion to this field.

The authors state in their response to the reviewers:

"In our opinion, this wide array of experiments validates our marker as a live-cell marker for the subset of long-lived, stable microtubules, even when it remains unknown what exactly this motor recognizes."

and later

"In a similar fashion, EB proteins were used as markers for dynamic microtubules many years before it was understood what exactly they recognize at the microtubule plus-end."

I will have to disagree with the comparison that the authors are making: we knew that EB1 proteins recognize plus ends when we started using them! Yes, we did not know the exact mechanism they use to recognize the microtubule ends, but we knew that they recognize dynamic ends. I would argue that the authors do not know what their marker recognizes. This revision now

states that it might recognize different lattices (expanded) based on cryo-EM work presented in an accompanying manuscript. The cellular cryo-EM work looking at this still requires additional clarification and validation, and the authors of the current manuscript were not able to show for the revision the results of a straightforward experiment: does their rigor kinesin indeed recognize expanded lattices in vitro? This is an easy, straightforward experiment that can easily be performed by the authors (they have expertise in this area). So, I remain puzzled by this. Thus, I cannot recommend publication of this manuscript unless (1) In vitro data is added to show that indeed the rigor kinesin does indeed recognize the features that the authors propose ie an expanded lattice. This can be easily done by looking at the in vitro binding of this rigor kinesin to GDP (compressed) and GMPCPP (expanded) lattices (a routine assay - see for example PMID: 28322917; PMID: 35996000) Sometimes it is not possible to perform the experiments needed to sort out mechanism (because technically not feasible or because expertise is not existent in the lab), but this is not the case here. This is a straightforward experiment using reagents and methodologies that are standard in the field and available in the authors' lab. These are not experiments that will take months.

(2) The accompanying cryo-EM manuscript is revised and accepted

A few more comments:

1. The authors should change their wording throughout the manuscript so that stability is not conflated with resistance to nocodazole or cold treatment-induced depolymerization. Those are two different things. Microtubules can have increased stability (ie increased lifetimes), but still depolymerize when exposed to cold or drugs. So, care should be taken when using the terms "stability" since its loose use in the literature has been confusing and detrimental to understanding cellular mechanism. The authors should refrain from using "stable" and state instead "do not depolymerize when subjected to nocodazole" so that the nature of the "stability" is very clear.

For example "In addition, we found that increasing MT detyrosination levels by overexpression of VSH1/SVBP does not confer stability to MTs, as the majority of detyrosinated MTs was lost upon treatment with nocodazole (Sup.Fig.1G), in agreement with earlier work (Khawaja et al. 1988). " should be changed for example to: "In addition, we found that increasing MT detyrosination levels by overexpression of VSH1/SVBP does not protect detyrosinated MTs from depolymerization upon treatment with nocodazole (Sup.Fig.1G), in agreement with earlier work (Khawaja et al. 1988).", since the authors have not shown that detyrosination does not confer stability. What they show is that some unknown degree of detyrosination does not protect microtubules against depolymerization by nocodazole.

A few pages later after stating that detyrosination did not confer stability against drug induced depolymerization, they then state starting on line 239: "These findings are also in line with earlier work that showed that detyrosinated MTs in extracted cells are not growth competent and are resistant to depolymerization (Infante et al. 2000). " Which one is it? This is completely confusing.

2. The authors interpret the fact that only 48% of the MTs remained stable on both sides of their laser induced cut as an indication that the MTs could be stabilized starting from the microtubule ends and then eventually over the entire lattice. I am not sure I follow this argument: could it also not indicate that the process of stabilization is distributed randomly along the lattice (and not propagating from the end)?

3. There is almost complete overlap between tubulin detyrosination and their stable marker - ie all microtubules marked by their marker are also detyrosinated. Recent work has shown that detyrosination leads to loss of plus end proteins that leads to slower growing microtubules that catastrophe less (PMID: 34022132), something the authors could add when discussing their proposal that the initial stabilization starts at the end creating a long-lived microtubule

Reviewer #2 (Comments to the Authors (Required)):

The revised paper has been substantially improved. The authors have renamed their probe for stable microtubules (StableMARK) and have included new data that StableMark labeling does not increase in cells with elevated detyrosinated tubulin (Suppl Fig 1) showing that it does not detect detyrosinated tubulin. They have quantified previous data including microtubule growth rate (Fig. 4F) and Rab6 and lysosome vesicle speeds (Fig. 5H&I) showing that StableMARK does not perturb microtubule growth or kinesin-1 vesicle motility. And they have increased citations to previous work.

I still have some qualms about what StableMARK is detecting, but apparently, an editorial decision has been made that a forthcoming paper will resolve this in part. It is clear that StableMARK is detecting stable microtubules, but to me, the lack of information about how it detects stable microtubules is problematic for interpreting results with the probe. Just one example, whereas the authors data shows convincingly that it detects some stable microtubules, they are not able to comment whether it detects all or only some of them.

There are two remaining issues that should be addressed before publication.

1. There is no evidence that StableMARK will be a useful probe for stable microtubules in cells other than U2OS cells. At least some of the simpler tests should be made in a distinct cell type.
2. In the Discussion (lines 227-240) the assertion that the lack of incorporation of tubulin (or EB comets) at the ends of stable microtubules is a novel finding is simply incorrect. The authors should discuss their findings in light of previous studies that have revealed similar findings, e.g., (Schulze E and Kirschner M, JCB, 1986; Webster DR et al., PNAS, 1987; Infante AS et al,

JCS,2000; Palazzo AF et al, NCB, 2001).

Reviewer #3 (Comments to the Authors (Required)):

In the revised version of their manuscript, the authors have carefully addressed my concerns. They have added a number of important experiments and textual changes that have strongly improved the manuscript, which I think could be considered for publication in its current form.

Point-to-point response for:

A live-cell marker to visualize the dynamics of stable microtubules throughout the cell cycle

Klara I. Jansen et al.

Reviewer #1

I thank the authors for their detailed reviewers' response. However, I remain skeptical about what this marker actually recognizes, and I am afraid that without a clear understanding of this, the current manuscript, the way it stands, will only add more confusion to this field.

The authors state in their response to the reviewers: "In our opinion, this wide array of experiments validates our marker as a live-cell marker for the subset of long-lived, stable microtubules, even when it remains unknown what exactly this motor recognizes." and later "In a similar fashion, EB proteins were used as markers for dynamic microtubules many years before it was understood what exactly they recognize at the microtubule plus-end." I will have to disagree with the comparison that the authors are making: we knew that EB1 proteins recognize plus ends when we started using them! Yes, we did not know the exact mechanism they use to recognize the microtubule ends, but we knew that they recognize dynamic ends. I would argue that the authors do not know what their marker recognizes. This revision now states that it might recognize different lattices (expanded) based on cryo-EM work presented in an accompanying manuscript. The cellular cryo-EM work looking at this still requires additional clarification and validation, and the authors of the current manuscript were not able to show for the revision the results of a straightforward experiment: does their rigor kinesin indeed recognize expanded lattices in vitro? This is an easy, straightforward experiment that can easily be performed by the authors (they have expertise in this area). So, I remain puzzled by this. Thus, I cannot recommend publication of this manuscript unless

(1) In vitro data is added to show that indeed the rigor kinesin does indeed recognize the features that the authors propose ie an expanded lattice. This can be easily done by looking at the in vitro binding of this rigor kinesin to GDP (compressed) and GMPCPP (expanded) lattices (a routine assay - see for example PMID: 28322917; PMID: 35996000). Sometimes it is not possible to perform the experiments needed to sort out mechanism (because technically not feasible or because expertise is not existent in the lab), but this is not the case here. This is a straightforward experiment using reagents and methodologies that are standard in the field and available in the authors' lab. These are not experiments that will take months.

➤ We have spent the last few months establishing these experiments, which have now been included as Supplemental Figure 2. In our experience, the experiments turned out to be not as trivial as suggested by the reviewer and we needed to optimize various aspects of the assay to get consistent results. In the end, what worked best was to compare the binding of StableMARK to GDP lattices with and without taxol, which revealed that StableMARK binds approximately 1.5 x more to microtubules with an expanded lattice than to microtubules with a compacted lattice. This preference of StableMARK for expanded lattices is consistent with the cryo-EM observations mentioned by the reviewer and could therefore help to explain the preference of the motor for stable microtubules in cells.

(2) The accompanying cryo-EM manuscript is revised and accepted

➤ We were informed by the editors that final acceptance of co-submitted papers will not depend on the status of the companion manuscript.

A few more comments:

1. The authors should change their wording throughout the manuscript so that stability is not conflated with resistance to nocodazole or cold treatment-induced depolymerization. Those are two different

things. Microtubules can have increased stability (ie increased lifetimes), but still depolymerize when exposed to cold or drugs. So, care should be taken when using the terms "stability" since its loose use in the literature has been confusing and detrimental to understanding cellular mechanism. The authors should refrain from using "stable" and state instead "do not depolymerize when subjected to nocodazole" so that the nature of the "stability" is very clear.

For example "In addition, we found that increasing MT detyrosination levels by overexpression of VSH1/SVBP does not confer stability to MTs, as the majority of detyrosinated MTs was lost upon treatment with nocodazole (Sup.Fig.1G), in agreement with earlier work (Khawaja et al. 1988). " should be changed for example to: "In addition, we found that increasing MT detyrosination levels by overexpression of VSH1/SVBP does not protect detyrosinated MTs from depolymerization upon treatment with nocodazole (Sup.Fig.1G), in agreement with earlier work (Khawaja et al. 1988).", since the authors have not shown that detyrosination does not confer stability. What they show is that some unknown degree of detyrosination does not protect microtubules against depolymerization by nocodazole. A few pages later after stating that detyrosination did not confer stability against drug induced depolymerization, they then state starting on line 239: "These findings are also in line with earlier work that showed that detyrosinated MTs in extracted cells are not growth competent and are resistant to depolymerization (Infante et al. 2000). " Which one is it? This is completely confusing.

- As correctly noted by the reviewer, we and others have shown that detyrosination on its own (induced by overexpression of VSH1/SVBP) is not sufficient to confer protection against drug-induced depolymerization. Nonetheless, this does not exclude the possibility that in normal conditions the subset of microtubules that is resistant against depolymerization still turns out to be the subset of microtubule that is also detyrosinated. To avoid confusion, we have reformulated these sentences as follows:

"These findings are in line with earlier work that showed that, in TC7 cells, MTs that are detyrosinated and turn over slowly also do not serve as templates for MT growth (Webster et al. 1987, Infante et al. 2000)."

Following the reviewer's suggestion, we have checked our manuscript and added clarifications to better explain what we mean with stability in different places. We have also more explicitly cited a recent review that we co-authored and that has an extensive section describing different concepts and mechanisms of microtubule stabilization (Akhmanova and Kapitein, Nat. Rev. Mol. Cell. Biol. 2022).

2. The authors interpret the fact that only 48% of the MTs remained stable on both sides of their laser induced cut as an indication that the MTs could be stabilized starting from the microtubule ends and then eventually over the entire lattice. I am not sure I follow this argument: could it also not indicate that the process of stabilization is distributed randomly along the lattice (and not propagating from the end)?

- We favor our interpretation because the first thing needed to stabilize a microtubule is to prevent depolymerization, which starts from the end. We have now also added the possibility suggested by the reviewer.

3. There is almost complete overlap between tubulin detyrosination and their stable marker - ie all microtubules marked by their marker are also detyrosinated. Recent work has shown that detyrosination leads to loss of plus end proteins that leads to slower growing microtubules that catastrophe less (PMID: 34022132)., something the authors could add when discussing their proposal that the initial stabilization starts at the end creating a long-lived microtubule

- In our U2OS cells, detyrosinated MTs appear to form a subset within the subset of acetylated MTs. Because StableMARK largely overlaps with the subset of acetylated MTs, many StableMARK-positive MTs are not detyrosinated. Therefore, we feel it would be confusing to attribute the behavior of StableMARK-positive MTs to their tyrosination/detyrosination state and we thus chose to not discuss this interesting *in vitro* study.

Reviewer #2

The revised paper has been substantially improved. The authors have renamed their probe for stable microtubules (StableMARK) and have included new data that StableMark labeling does not increase in cells with elevated detyrosinated tubulin (Suppl Fig 1) showing that it does not detect detyrosinated tubulin. They have quantified previous data including microtubule growth rate (Fig. 4F) and Rab6 and lysosome vesicle speeds (Fig. 5H&I) showing that StableMARK does not perturb microtubule growth or kinesin-1 vesicle motility. And they have increased citations to previous work.

I still have some qualms about what StableMARK is detecting, but apparently, an editorial decision has been made that a forthcoming paper will resolve this in part. It is clear that StableMARK is detecting stable microtubules, but to me, the lack of information about how it detects stable microtubules is problematic for interpreting results with the probe. Just one example, whereas the authors data shows convincingly that it detects some stable microtubules, they are not able to comment whether it detects all or only some of them.

There are two remaining issues that should be addressed before publication.

- We thank the reviewer for the positive comments and for recommending publication, pending minor changes.

1. There is no evidence that StableMARK will be a useful probe for stable microtubules in cells other than U2OS cells. At least some of the simpler tests should be made in a distinct cell type.

- Supplemental Figure 6 of our revised manuscript already showed StableMARK in six different cell types, namely U2OS, COS-7, 3T3, Vero, HeLa and Caco-2. In the new version, this data is now shown in Supplemental Figure 7.

2. In the Discussion (lines 227-240) the assertion that the lack of incorporation of tubulin (or EB comets) at the ends of stable microtubules is a novel finding is simply incorrect. The authors should discuss their findings in light of previous studies that have revealed similar findings, e.g., (Schulze E and Kirschner M, JCB, 1986; Webster DR et al., PNAS, 1987; Infante AS et al, JCS,2000; Palazzo AF et al, NCB, 2001).

- We did not intend to suggest this was a new finding. We have re-worded this section to clarify that StableMARK allowed us to confirm many of the known properties of stable MTs and additionally provided new insights:

“Our experiments allowed us to confirm many of the known properties of stable MTs and additionally provided new insights. For example, while it has been previously reported that most stable MTs do not have a dynamic plus-end (Schulze & Kirschner 1986; Webster et al. 1987; Infante et al. 2000; Palazzo et al. 2001), our work reveals that long-lived MTs have varying degrees of lattice stabilization, with about half of them not depolymerizing upon laser-induced severing.”

Reviewer #3

In the revised version of their manuscript, the authors have carefully addressed my concerns. They have added a number of important experiments and textual changes that have strongly improved the manuscript, which I think could be considered for publication in its current form.

- We thank the reviewer for the positive comments and for recommending publication.

January 27, 2023

RE: JCB Manuscript #202106105RRR

Prof. Lukas Kapitein
Utrecht University
Padualaan 8
Utrecht 3533 CH
Netherlands

Dear Prof. Kapitein:

Thank you for submitting your revised manuscript entitled "A live-cell marker to visualize the dynamics of stable microtubules throughout the cell cycle." Thank you also for all your efforts in revising the paper throughout the review process. We would be happy to publish your paper in JCB pending final revisions necessary to meet our formatting guidelines (see details below). Please also clarify the statistical analysis of the data in Figure S2D, does the significance indicator apply to each replicate or the average?

A. MANUSCRIPT ORGANIZATION AND FORMATTING:

1) Text limits: Character count for Tools is < 40,000, not including spaces. Count includes title page, abstract, introduction, results, discussion, and acknowledgments. Count does not include materials and methods, figure legends, references, tables, or supplemental legends.

2) Figure formatting: Tools may have up to 10 main text figures. Scale bars must be present on all microscopy images, including inset magnifications. Please add a scale bar to Figure S6D.

Also, please avoid pairing red and green for images and graphs to ensure legibility for color-blind readers. If red and green are paired for images, please ensure that the particular red and green hues used in micrographs are distinctive with any of the colorblind types. If not, please modify colors accordingly or provide separate images of the individual channels.

3) Statistical analysis: Error bars on graphic representations of numerical data must be clearly described in the figure legend. The number of independent data points (n) represented in a graph must be indicated in the legend. Please, indicate whether 'n' refers to technical or biological replicates (i.e. number of analyzed cells, samples or animals, number of independent experiments). If independent experiments with multiple biological replicates have been performed, we recommend using distribution-reproducibility SuperPlots (please see Lord et al., JCB 2020) to better display the distribution of the entire dataset, and report statistics (such as means, error bars, and P values) that address the reproducibility of the findings.

Statistical methods should be explained in full in the materials and methods. For figures presenting pooled data the statistical measure should be defined in the figure legends. Please also be sure to indicate the statistical tests used in each of your experiments (both in the figure legend itself and in a separate methods section) as well as the parameters of the test (for example, if you ran a t-test, please indicate if it was one- or two-sided, etc.). Also, if you used parametric tests, please indicate if the data distribution was tested for normality (and if so, how). If not, you must state something to the effect that "Data distribution was assumed to be normal but this was not formally tested."

4) Materials and methods: Should be comprehensive and not simply reference a previous publication for details on how an experiment was performed. Please provide full descriptions (at least in brief) in the text for readers who may not have access to referenced manuscripts. The text should not refer to methods "...as previously described."

5) For all cell lines, vectors, constructs/cDNAs, etc. - all genetic material: please include database / vendor ID (e.g., Addgene, ATCC, etc.) or if unavailable, please briefly describe their basic genetic features, even if described in other published work or gifted to you by other investigators (and provide references where appropriate). Please be sure to provide the sequences for all of your oligos: primers, si/shRNA, RNAi, gRNAs, etc. in the materials and methods. You must also indicate in the methods the source, species, and catalog numbers/vendor identifiers (where appropriate) for all of your antibodies, including secondary. If antibodies are not commercial, please add a reference citation if possible.

6) Microscope image acquisition: The following information must be provided about the acquisition and processing of images:

- Make and model of microscope
- Type, magnification, and numerical aperture of the objective lenses
- Temperature
- Imaging medium
- Fluorochromes
- Camera make and model
- Acquisition software
- Any software used for image processing subsequent to data acquisition. Please include details and types of operations involved (e.g., type of deconvolution, 3D reconstitutions, surface or volume rendering, gamma adjustments, etc.).

7) References: There is no limit to the number of references cited in a manuscript. References should be cited parenthetically in the text by author and year of publication. Abbreviate the names of journals according to PubMed.

8) Supplemental materials: Tools papers are generally allowed up to 5 supplemental figures and 10 videos. You currently exceed this limit but, in this case, we will be able to give you the extra space. A summary of all supplemental material should appear at the end of the Materials and methods section. Please include one brief sentence per item.

9) Video legends: Should describe what is being shown, the cell type or tissue being viewed (including relevant cell treatments, concentration and duration, or transfection), the imaging method (e.g., time-lapse epifluorescence microscopy), what each color represents, how often frames were collected, the frames/second display rate, and the number of any figure that has related video stills or images.

10) eTOC summary: A ~40-50 word summary that describes the context and significance of the findings for a general readership should be included on the title page. The statement should be written in the present tense and refer to the work in the third person. It should begin with "First author name(s) et al..." to match our preferred style.

11) Conflict of interest statement: JCB requires inclusion of a statement in the acknowledgements regarding competing financial interests. If no competing financial interests exist, please include the following statement: "The authors declare no competing financial interests." If competing interests are declared, please follow your statement of these competing interests with the following statement: "The authors declare no further competing financial interests."

12) A separate author contribution section is required following the Acknowledgments in all research manuscripts. All authors should be mentioned and designated by their first and middle initials and full surnames. We encourage use of the CRediT nomenclature (<https://casrai.org/credit/>).

13) ORCID IDs: ORCID IDs are unique identifiers allowing researchers to create a record of their various scholarly contributions in a single place. At resubmission of your final files, please consider providing an ORCID ID for as many contributing authors as possible.

14) Materials and data sharing: As a condition of publication, authors must make protocols and unique materials (including, but not limited to, cloned DNAs; antibodies; bacterial, animal, or plant cells; and viruses) described in our published articles freely available upon request by researchers, who may use them in their own laboratory only. All materials must be made available on request and without undue delay. We strongly encourage to deposit all the cell lines/strains and reagents generated in this study in public repositories.

B. FINAL FILES:

****The license to publish form must be signed before your manuscript can be sent to production. A link to the electronic license to publish form will be sent to the corresponding author only. Please take a moment to check your funder requirements before choosing the appropriate license.****

Thank you for this interesting contribution, we look forward to publishing your paper in Journal of Cell Biology.

Sincerely,

Rebecca Heald, PhD
Monitoring Editor
Journal of Cell Biology

Dan Simon, PhD
Scientific Editor
Journal of Cell Biology